# Slight Corruption in Pre-training Data Makes Better Diffusion Models

Hao Chen[1]*, Yujin Han[2], Diganta Misra[1,3], Xiang Li[1], Kai Hu[1],
Difan Zou[2], Masashi Sugiyama[4,5], Jindong Wang[6]†, Bhiksha Raj[1,7]

[1]Carnegie Mellon University, [2]The University of Hong Kong, [3]Mila - Quebec AI Institute,
[4] RIKEN AIP, [5]The University of Tokyo, [6]William & Mary, [7]MBZUAI

## Abstract

Diffusion models (DMs) have shown remarkable capabilities in generating realistic high-quality images, audios, and videos. They benefit significantly from extensive pre-training on large-scale datasets, including web-crawled data with paired data and conditions, such as image-text and image-class pairs. Despite rigorous filtering, these pre-training datasets often inevitably contain *corrupted* pairs where *conditions* do not accurately describe the data. This paper presents the first comprehensive study on the impact of such condition corruption in pre-training data of DMs. We synthetically corrupt ImageNet-1K and CC3M to pre-train and evaluate over 50 conditional DMs. Our empirical findings reveal that various types of slight corruption in pre-training can significantly enhance the quality, diversity, and fidelity of the generated images across different DMs, both during pre-training and downstream adaptation stages. Theoretically, we consider a Gaussian mixture model and prove that slight corruption in the condition leads to higher entropy and a reduced 2-Wasserstein distance to the ground truth of the data distribution generated by the corruptly trained DMs. Inspired by our analysis, we propose a simple method to improve the training of DMs on practical datasets by adding condition embedding perturbations (CEP). CEP significantly improves the performance of various DMs in both pre-training and downstream tasks. We hope that our study provides new insights into understanding the data and pre-training processes of DMs and all models are released at `https://huggingface.co/DiffusionNoise`.

## 1 Introduction

Recently, diffusion models (DMs) have been demonstrating unprecedented capabilities in generating high-quality, realistic, and faithful images [1–5], audios [6, 7], and videos [8]. In addition, they exhibit impressive conditional generation results [9–11] when trained with classifier-free guidance [12]. The successes of DMs are often attributed to the massive pre-training on large-scale datasets consisting of paired data and conditions [13–17], which also empowered and facilitated numerous downstream applications and personalization of pre-trained models, such as subject-driven generation [18, 19], controllable conditional generation [20–22], and synthetic data training [23–25].

The large-scale pre-training datasets of paired data and conditions are usually web-crawled. For example, Stable Diffusion [26] was pre-trained on LAION-2B [17], which contains billion-scale image-text pairs collected from Common Crawl [27]. Despite the heavy filtering mechanisms used in collecting pre-training datasets [17, 28], they still inevitably contain corrupted pairs where conditions do not correctly describe or match the data, such as corrupted labels and texts [29–32]. While large-scale datasets are necessary for DMs to perform well, the corruption may lead to unexpected

---

*haoc3@andrew.cmu.edu

†Correspondence to: jwang80@wm.edu

38th Conference on Neural Information Processing Systems (NeurIPS 2024).

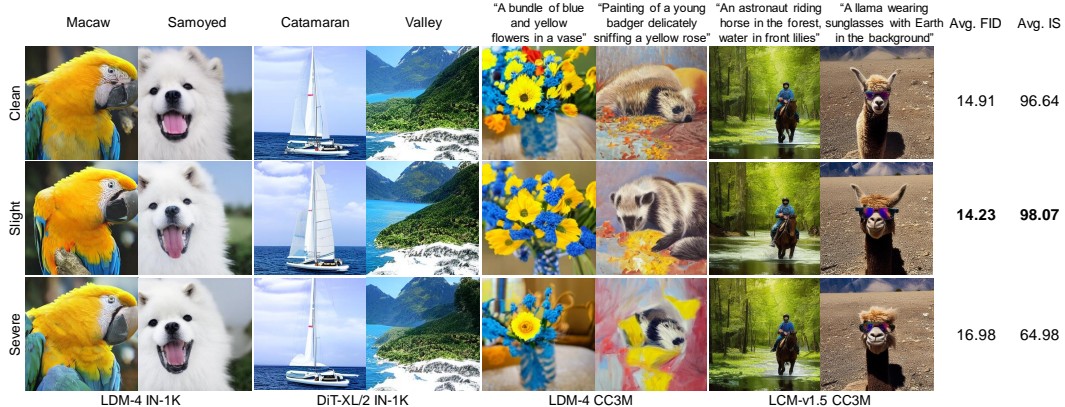

Figure 1: Visualization from class and text-conditional DMs pre-trained with clean, slight, and severe condition corruption. Slight corruption in pre-training improves the quality and diversity of images.

behavior or generalization performance of models [33–35] during both pre-training and adaptation stages, especially for safety-critical domains such as healthcare [36] and autonomous driving [37, 38].

Conventional wisdom may suggest that training under corrupted conditions could lead to deterioration in performance. For example, Noisy Label Learning [39–42, 29, 43, 44] aims to improve the generalization of models when training with corrupted labels. Label-noise robust conditional generative adversarial nets [45, 46] and DMs [47] have also been studied. However, these works are primarily concerned with supervised learning in downstream scenarios with assumptions of high noise ratios and the same training and testing data distributions. Due to the misalignment with large-scale self-supervised pre-training in practice on filtered datasets with relatively smaller noise ratios, the effects of corruption in pre-training can also differ from those in downstream [48, 49].Understanding the effects of pre-training with such corruption is challenging and still remains largely unexplored.

In this paper, we provide the first comprehensive and practical study on condition corruption in the pre-training of DMs. Through in-depth analysis, we empirically, theoretically, and methodologically verify that **slight condition corruption in pre-training makes better DMs**. We pre-train over 50 class-conditional and text-conditional DMs using classifier-free guidance (CFG) [12] on ImageNet-1K (IN-1K) [50] and CC3M [14] with synthetically corrupted conditions, i.e., classes and texts, of various levels. Our study covers a wide range of DM families, including Latent Diffusion Model (LDM) [9], Diffusion Transformer (DiT) [11], and Latent Consistency Model (LCM) [51, 52]. Due to the known obstacles of evaluating generative mod-

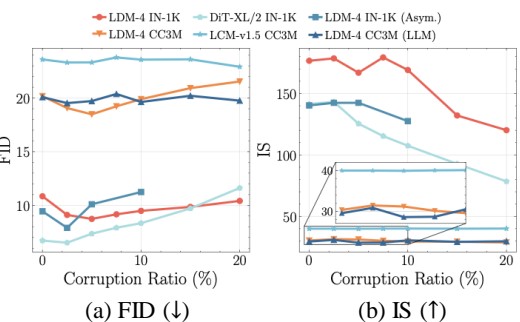

Figure 2: (a) FID and (b) IS of DMs pre-trained on IN-1K and CC3M with various corruption. Slight corruption of various types helps DMs achieve better performance, compared to the clean ones.

els [53–55], we conduct both pre-training and downstream evaluation from the perspectives of image quality, fidelity, diversity, complexity, and memorization, to comprehensively understand the effects of pre-training corruption of DMs. More specifically, for pre-training, we directly evaluate the images generated from the pre-trained models, and for downstream adaptation, we evaluate on the images generated using personalized models with ControlNet [20] and T2I-Adapter [21] from the pre-trained ones. In addition, we theoretically investigate how slight corruption in conditional embeddings benefits the training and generative processes of DMs. Our key findings include:

- Empirically, slight corruption in pre-training facilitates the DMs to generate images with higher quality and more diversity, both qualitatively (in Fig. 1) and quantitatively (in Fig. 2).
- Theoretically, we employ a Gaussian mixture model to show slight condition corruption improves the diversity and quality of generation by increasing entropy over clean condition generation and reducing the quadratic 2-Wasserstein distance to the true data distribution (in Section 4).

- Methodologically, based on our analysis, we propose a simple method to improve the pre-training of DMs by adding conditional embedding perturbations (CEP). We show that CEP can significantly boost the performance of various DMs in both pre-training and downstream tasks (in Section 5).

Going beyond images, we do see the potential of this study in other modalities. Our efforts may also inspire future investigation on other types of corruption and bias inside pre-training datasets. We hope that our work can shed light on the future research of diffusion models and responsible AI.

## 2 Preliminary

**Denoising Diffusion Models.** DMs are probabilistic models that learn the data distribution $\mathbb{P}(\mathbf{x})$, with $\mathbf{x}$ denoting the observed data[3], over a set of latent variables $\mathbf{z}_1, \ldots, \mathbf{z}_T$ with length $T$ [1, 57]. It assumes a forward diffusion process, gradually adding Gaussian noise to the data with a fixed Markov chain: $q(\mathbf{z}_t|\mathbf{x}) = \mathcal{N}(\sqrt{\bar{\alpha}_t}\mathbf{x}, (1 - \bar{\alpha}_t)\mathbf{I})$, which can be re-parameterized as $\mathbf{z}_t = \sqrt{\bar{\alpha}_t}\mathbf{x} + \sqrt{1 - \bar{\alpha}_t}\boldsymbol{\epsilon}$ with $\boldsymbol{\epsilon} \sim \mathcal{N}(\mathbf{0}, \mathbf{I})$ and $\bar{\alpha}_t$ as constants produced by a noise scheduler. DMs are trained via the reverse process, inverting the forward process as: $p_\theta(\mathbf{z}_{t-1}|\mathbf{z}_t) = \mathcal{N}(\boldsymbol{\mu}_\theta(\mathbf{z}_t), \boldsymbol{\Sigma}_\theta(\mathbf{z}_t))$, with a network that predicts the statistics of $p_\theta$. Setting $\boldsymbol{\Sigma}_\theta(\mathbf{z}_t) = (1 - \bar{\alpha}_t)\mathbf{I}$ to untrained constants, the reverse process is simplified as training equally weighted denoising autoencoders $\boldsymbol{\epsilon}(\mathbf{z}_t, t)$ with uniformly sampled $t$:

$$\mathcal{L}_{\text{DM}} = \mathbb{E}_{\mathbf{x}, \boldsymbol{\epsilon} \sim \mathcal{N}(0,\mathbf{I}), t \sim \mathcal{U}(1,T)} \left[ \|\boldsymbol{\epsilon} - \boldsymbol{\epsilon}_\theta(\mathbf{z}_t, t)\|_2^2 \right]. \tag{1}$$

After training, new images can be generated by sampling $\mathbf{z}_{t-1} \sim \mathbf{p}_\theta(\mathbf{z}_{t-1}|\mathbf{z}_t)$ starting with $\mathcal{N}(\mathbf{0}, \mathbf{I})$.

**Classifier-free Guidance (CFG).** Extra condition information $y$, such as class labels and text prompts, can be injected into DMs with conditional embeddings $\mathbf{c}_\theta(y)$ from modality-specific encoders [9] for conditional generation: $\mathbf{p}_\theta(\mathbf{z}_{t-1}|\mathbf{z}_t, \mathbf{c}_\theta(y))$. CFG [12] jointly learns a unconditional model $\boldsymbol{\epsilon}_\theta(\mathbf{z}_t, t, \mathbf{c}_\theta(\emptyset))$ with an empty condition $y = \emptyset$ and a conditional model $\boldsymbol{\epsilon}_\theta(\mathbf{z}_t, t, \mathbf{c}_\theta(y))$, and combines them linearly to control the trade-off of sample quality and diversity in generation:

$$\hat{\boldsymbol{\epsilon}}_\theta(\mathbf{z}_t, t, \mathbf{c}_\theta(y)) = \boldsymbol{\epsilon}_\theta(\mathbf{z}_t, t, \mathbf{c}_\theta(\emptyset)) + s\left(\boldsymbol{\epsilon}_\theta(\mathbf{z}_t, t, \mathbf{c}_\theta(y)) - \boldsymbol{\epsilon}_\theta(\mathbf{z}_t, t, \mathbf{c}_\theta(\emptyset))\right), \tag{2}$$

where $s > 1$ denotes the guidance scale. We adopt CFG by default with the training objective:

$$\mathcal{L}_{\text{DM}} = \mathbb{E}_{\mathbf{x}, y, \boldsymbol{\epsilon} \sim \mathcal{N}(0,\mathbf{I}), t \sim \mathcal{U}(1,T)} \left[ \|\boldsymbol{\epsilon} - \boldsymbol{\epsilon}_\theta(\mathbf{z}_t, t, \mathbf{c}_\theta(y))\|_2^2 \right]. \tag{3}$$

**Condition Corruption.** Ideally, each $y$ should accurately describe and match $\mathbf{x}$. However, in practice, due to errors from the collection of web-crawled datasets, conditions $y^c$ may un-match $\mathbf{x}$. We define $(\mathbf{x}, y^c)$ as pairs with condition corruption, and assume that $\mathbf{c}_\theta(y^c) = \mathbf{c}_\theta(y; \eta, \boldsymbol{\xi})$, where $\boldsymbol{\xi}$ denotes certain noise and $\eta$ denotes corruption ratio that implicitly controls the noise magnitude.

## 3 Understanding the Pre-training Corruption in Diffusion Models

In this section, we conduct the first comprehensive and practical study on pre-training DMs with condition corruption. Through holistic exploration with synthetically corrupted datasets, we reveal a surprising observation that slight pre-training corruption can be beneficial for DMs.

### 3.1 Pre-training Evaluation

**Pre-training Setup**. Here, we adopt Latent Diffusion Models (LDMs) [9] with the pre-trained VQ-VAE [58, 56] and a down-sampling factor of 4 for the latent space of observed data $\mathbf{x}$, denoted as LDM-4. More specifically, we train class-conditional and text-conditional LDM-4 from scratch on synthetically corrupted IN-1K [50] and CC3M [14], respectively, with a resolution of $256 \times 256$. We use a class embedding layer and a learnable pre-trained BERT [59] to compute the conditional embeddings of the IN-1K class labels and the CC3M text prompts. To introduce synthetic corruption into the conditions, we randomly flip the class label into a random class for IN-1K, and randomly swap the text of two sampled image-text pairs for CC3M, following [48, 49] (other corruption types studied in Section 3.3). We train models with different corruption ratios $\eta \in \{0, 2.5, 5, 7.5, 10, 15, 20\}\%$ More details on synthetic corruption and pre-training recipes are shown in Appendix B.1 and B.3.

---

[3]We use $\mathbf{x}$ for images in both the raw pixel space and the latent space of VQ-VAE [56].

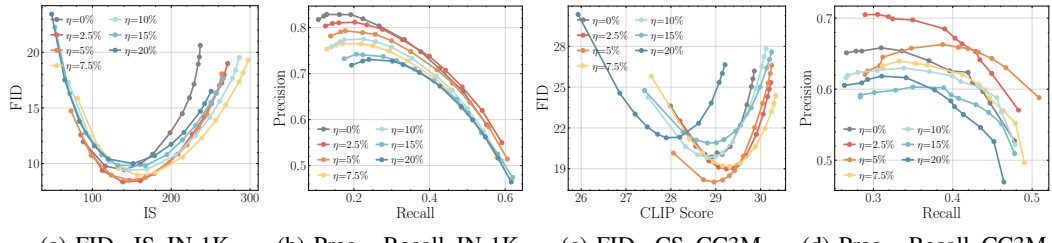

| (a) FID - IS, IN-1K | (b) Prec. - Recall, IN-1K | (c) FID - CS, CC3M | (d) Prec. - Recall, CC3M |

Figure 3: Quantitative evaluation of generated images from class and text-conditional LDMs pre-trained with condition corruption. All metrics are computed over $50K$ generated images and validation images of IN-1K and MS-COCO. We plot FID vs. IS or CS ((a) and (c)) , and Precision vs. Recall ((b) and (d)), where each point indicates the results computed from using a guidance scale. Models pre-trained with slight condition corruption achieve better FID, IS or CS, and PR trade-off.

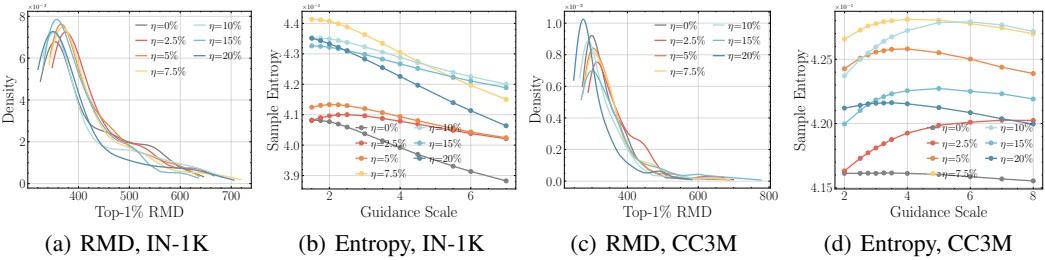

| (a) RMD, IN-1K | (b) Entropy, IN-1K | (c) RMD, CC3M | (d) Entropy, CC3M |

Figure 4: Quantitative evaluation of complexity and diversity of class and text-conditional LDMs. We plot the top-1% RMD score ((a) and (c)) which measures the complexity and diversity of samples (with $s = 2.0$ and $s = 3.0$ for IN-1K and CC3M LDMs), and the sample entropy ((b) and (d)) as a proxy measure of diversity, where each point indicates the result of a guidance scale. Models pre-trained with slight condition corruption generate samples of higher complexity and diversity.

**Evaluation of Pre-trained Models**. We directly use the pre-trained LDMs to generate images to study the effects of condition corruption in the pre-training stage. We use IN-1K class labels for class-conditional LDMs and MS-COCO text prompts [60] for text-conditional LDMs to generate 50K images and compare with the real validation images. The images are generated using a set of guidance scales $s \in \{1.5, 2.0, \ldots, 10.0\}$ and DPM [61] scheduler with 50 steps for faster inference speed[4]. We adopt Fréchet Inception Distance (FID) [63], Inception Score (IS) [64], Precision, and Recall [65] to evaluate the quality, fidelity, and coverage of the generated images. For CC3M models, we use the CLIP score (CS) [66] to measure the similarity of the generated images and conditional text prompts. From the perspectives of sample complexity and diversity, we compute the top-1% Relative Mahalanobis Distance (RMD) [67, 68], calculated from the estimated class-specific and class-agnostic distributions of generated data, and the sample entropy [69, 70], calculated from the VQ-VAE codebook. We also adopt other metrics, including sFID [71], TopPR F1 [72], average $L_2$ distance, and memorization ratio [73]. More details of the metrics used are shown in Appendix B.7.

**Results**. We present the main quantitative results of pre-training in Fig. 3 and 4, and the qualitative results in Fig. 5. More results are shown in Appendix C. In summary, we found that **slight pre-training corruption[5] can facilitate the quality, fidelity, and diversity of generated images**:

- Class and text-conditional models pre-trained with slight corruption achieve significantly lower FID and higher IS and CLIP score (Fig. 3(a) and 3(c)). They also present comparable and better Precision-Recall curves (Fig. 3(b) and 3(d)), compared to clean pre-trained models.

- Models pre-trained with slight corruption generate images with higher complexity and diversity, with a right-shifted density of RMD (Fig. 4(a) and 4(c)), and larger entropy (Fig. 4(b) and 4(d)).

- Qualitatively, models with slight corruption learn a more diverse distribution. Generated images present better variability in the circular walk around the latent space (Fig. 5(a) and 5(b)).

---

[4]In Appendix C, we also present the results of IN-1K LDM-4 using the DDIM [62] scheduler with 250 steps.
[5]Slight corruption corresponds to $\eta \leq 7.5\%$, which might be common in practical large-scale datasets.

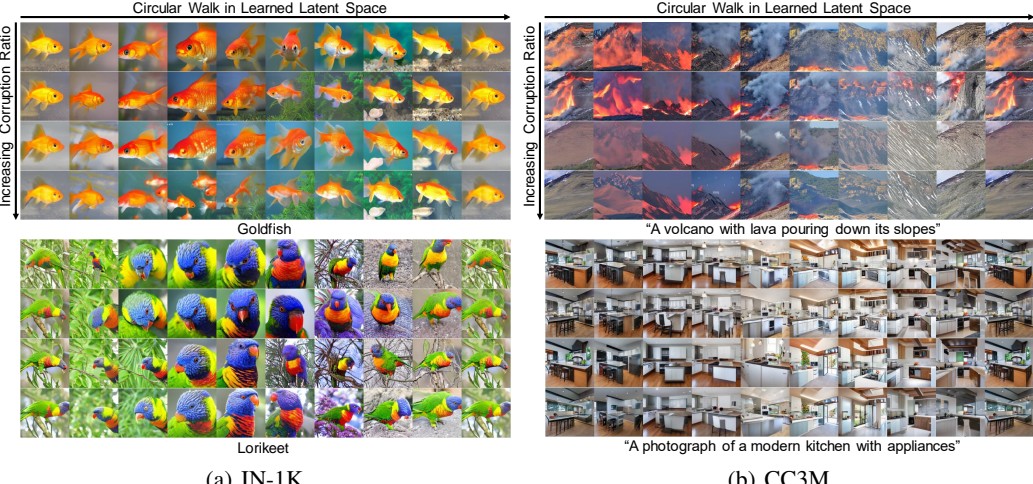

(a) IN-1K        (b) CC3M

Figure 5: Qualitative evaluation of images generated from circular walk around the learned latent space using (a) class-conditional IN-1K LDMs and (b) text-conditional CC3M LDMs. Models pre-trained with slight condition corruption present more diversity in the learned distribution.

- More corruption in pre-training can potentially lead to quality and diversity degradation. As $\eta$ increases, almost all metrics first improve and then degrade. However, the degraded measure with more corruption sometimes is still better than the clean ones (e.g. IS and Entropy).

## 3.2    Downstream Personalization Evaluation

A common scenario of DMs pre-trained on large-scale datasets is that they can be personalized and customized for more controllable generation [74, 75] after tuning on smaller datasets. Here, we also examine the effects of pre-training condition corruption of DMs at downstream personalization tasks.

**Downstream Personalization Setup**. We personalize the pre-trained LDMs with two common methods: ControlNet [20] and T2I-Adapter [21]. Both methods can enable the pre-trained DMs to generate more controllable images according to input spatial conditioning. For fair comparison, we automatically annotate the ImageNet-100 (IN-100) dataset to canny edges using the OpenCV canny detector [76] and segmentation masks using SegmentAnything (SAM) [77], similar to Zhang et al. [20]. For text-conditional LDMs, we additionally use BLIP [78] to generate MS-COCO-style text prompts for IN-100. We then fine-tune the LDMs on the annotated training set of IN-100. More details on the annotation and personalization setup are shown in Appendix B.2 and B.6, respectively.

**Evaluation of Personalized Models**. After tuning, we evaluate the personalized LDMs in the annotated validation set of IN-100. We mainly compute FID, IS, Precision, and Recall to compare the models. We similarly use a set of guidance scales to generate images, but only report results of the best guidance scale in this part due to space limit. The complete results are shown in Appendix D.

**Results**. Similarly, from the main results in Fig. 6 and Fig. 7, we found that **slight pre-training corruption can also benefit the quality of generated images at downstream personalization**:

- Slight pre-training corruption helps the personalized model generate images with lower FID (Fig. 6(a) and 6(c)) and higher IS score (Fig. 6(b) and 6(d)), whereas more corruption deteriorates.
- Qualitatively, the personalization images from slight corruption pre-trained models also present more diversity, better fidelity with the input spatial controls, and higher quality.

## 3.3    Discussion: Other Types of Pre-training Corruption and Diffusion Models

Our previous studies mainly involve LDMs and symmetric random corruption. Here, we additionally study other types of corruption and DMs to verify that the above observations universally hold.

**Condition Corruption**. We consider asymmetric (Asym.) label corruption for IN-1K and Large Language Model (LLM) corruption for CC3M. For IN-1K, we introduce corruption only within the

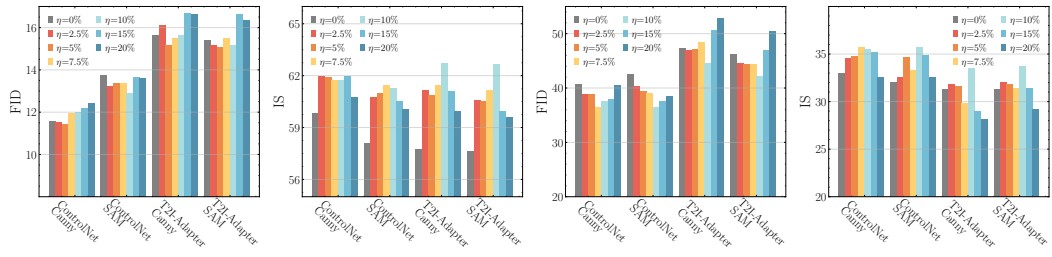

| (a) FID, IN-1K | (b) IS, IN-1K | (c) FID, CC3M | (d) IS, CC3M |

Figure 6: Quantitative evaluation of ControlNet and T2I-Adapter personalized class and text-conditional LDMs. FID ((a) and (c)) and IS ((b) and (d)) are computed using the 5K generated images. Slightly corrupted pre-trained models also present better performance in downstream personalization.

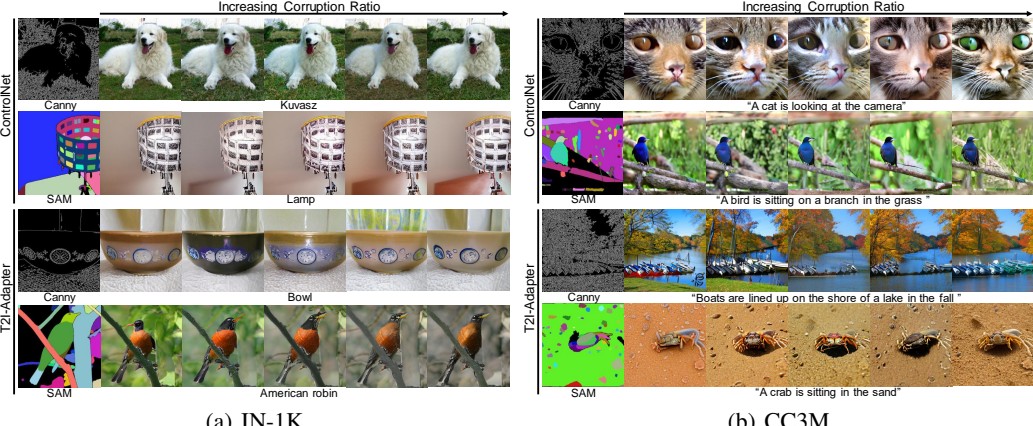

| (a) IN-1K | (b) CC3M |

Figure 7: Qualitative evaluation of ControlNet and T2I-Adapter (a) IN-1K and (b) CC3M LDMs.

overlapped classes with CIFAR-100 [79], while maintaining others as clean. For CC3M, we prompt GPT-4 [80] to corrupt the texts. More details of the corruption are shown in Appendix B.1.

**Diffusion Models**. LDMs utilize U-Net [81] as backbone and Cross-Attention for adding conditional information [9]. We pre-train class-conditional diffusion transformers on IN-1K for extra assessment, termed DiT-XL/2 [11], with Transformer [82] as backbone and adaptive LayerNorm [83–86] for conditional information. We also pre-train the recent text-conditional Latent Consistency Models (LCMs) [52, 51] on CC3M, which distill Stable Diffusion v1.5 [26] models to enable swift inference with minimal steps, noted as LCM-v1.5. Detailed setup is shown in Appendices B.4 and B.5.

**Results**. We present the main results in Fig. 2 due to the space limit. Full results are shown in Appendix C. We find that slight condition corruption of various types universally facilitates the performance of different DMs and consistently makes them outperform the clean pre-trained ones.

## 4 Theoretical Analysis

In this section, we theoretically analyze condition corruption and find that slight corruption prevents the generated distribution from collapsing to the empirical distribution of the training data and encourages coverage of the entire data space, thereby enhancing diversity and alignment with the ground truth. We present a concise overview here and provide a comprehensive analysis in Appendix A.

**Data Distribution.** We concentrate on the prototypical problem of sampling from Gaussian mixture models (GMMs). Specifically, we consider the distribution of data $\mathbf{x} \in \mathbb{R}^d$ that satisfies:

$$\mathbb{P}(\mathbf{x}) := \sum_{y \in \mathcal{Y}} w_y \mathcal{N}(\boldsymbol{\mu}_y, \mathbf{I}), \tag{4}$$

where $y$ denote class labels of a finite set $y \in \{1, \ldots, |\mathcal{Y}|\}$. Given any class, $\mathbf{x}|y$ follows a Gaussian $\mathcal{N}(\boldsymbol{\mu}_y, \mathbf{I})$, and $w_y$ represents the weight of the Gaussian components which satisfies $\sum_{y \in \mathcal{Y}} w_y = 1$.

**Denoising Networks and Condition Corruption.** Inspired by recent works that also target on GMMs [87, 88] of DMs, we parameterize the denoising network as a piece-wise linear function:

$$\boldsymbol{\epsilon}_\theta(\mathbf{x}_t, y^c) = \sum_{k=1}^{|\mathcal{Y}|} \mathbb{1}_{y^c=k}\left(\mathbf{W}_t^k \mathbf{x}_t + \mathbf{V}_t^k \mathbf{c}(y^c)\right), \tag{5}$$

where $\mathbf{c}(y^c)$ is the one-hot encoding of corrupted label $y^c$ and $\{\mathbf{W}_t^k, \mathbf{V}_t^k\}_{k=1}^{|\mathcal{Y}|}$ are trainable parameters. Specifically, following a line of existing work [89–91], we adopt a simpler label-noise model by adding Gaussian perturbation to the label embedding, perturbing the clean condition $\mathbf{c}(y)$ with standard Gaussian noise $\boldsymbol{\xi}$ to obtain $\mathbf{c}(y^c) = \mathbf{c}(y) + \gamma\boldsymbol{\xi}$. Here, the corruption control parameter $\gamma$ corresponds to the corruption ratio $\eta$ for a more direct noise magnitude control. While our theoretical framework focuses on Gaussian noise, it can also be extended to distributions such as uniform.

## 4.1 Generation Diversity: Clean vs. Corrupted Conditions

We employ entropy to evaluate the diversity of generated images, following Wu et al. [70]. Higher entropy suggests a wider spread of data, yielding greater diversity in generated images, while lower entropy implies a more concentrated distribution with less diversity. We present Theorem 1, showing the difference in entropy between generations with corrupted and clean conditions:

**Theorem 1.** *For any class $k \in \mathcal{Y}$ and sufficiently large length $T$, assuming the norm of corresponding expectation $\|\boldsymbol{\mu}_k\|_2^2$ is a constant and the empirical covariance of training data is full rank, let $\mathbf{z}_T$ and $\mathbf{z}_T^c$ be the generation with clean and corrupted conditions respectively, then it holds that*

$$H(\mathbf{z}_T^c|y=k) - H(\mathbf{z}_T|y=k) = \Theta(\gamma^2 d), \tag{6}$$

*where $\gamma$ is the corruption control parameter and $d$ is the data dimension.*

The proof is provided in Appendix A.4.1. Theorem 1 indicates that for any class $k$, corrupted conditions enhance image diversity by increasing generation entropy. Moreover, with suitable $\gamma$ values, image diversity can grow with noise, aligning with observations in Fig. 4.

## 4.2 Generation Quality: Clean vs. Corrupted Conditions

We then analyze why corrupted conditions benefit the quality of generated images, as also observed in Section 3.1. We employ the 2-Wasserstein distance as a metric to evaluate the sampling error between the true and the generated distributions, with clean and corrupted conditions. A distributed generated closer to the real data distribution indicates better image quality [63]. In Theorem 2, we analyze the difference in the quality of data generated by corrupted DMs and clean ones:

**Theorem 2.** *For any $k \in \mathcal{Y}$ and sufficiently large length $T$, assuming the norm of corresponding expectation $\|\boldsymbol{\mu}_k\|_2^2$ is constant, let $\mathbb{P}$, $\mathbb{Q}_\mathbf{X}$ and $\mathbb{Q}_\mathbf{X}^c$ be the ground truth, clean, and corrupted condition distributions over training data $\mathbf{X}$. Then if $\gamma = O(1/\sqrt{\max_k n_k})$, it holds that*

$$\mathbb{E}_\mathbf{X}\left[\mathcal{W}_2^2(\mathbb{P}, \mathbb{Q}_\mathbf{X}) - \mathcal{W}_2^2(\mathbb{P}, \mathbb{Q}_\mathbf{X}^c)|y=k\right] = \Omega\left(\frac{\gamma^2 d}{n_k}\right), \tag{7}$$

*where $\mathcal{W}_2(\cdot, \cdot)$ denotes the 2-Wasserstein distance between two distributions, $n_k$ is the sample size of $k$-labeled dataset, and $d$ is the data dimension.*

Here the expectation is taken over the random sample of the training dataset from the data distribution. Detailed proof is shown in Appendix A.4.2. Theorem 2 reveals that for any class $k$, small corruption yields generation distributions closer to the true distribution than clean ones. This partially verifies that the generation quality of the uncorrputly trained diffusion model can be improved by adding slight corruption to the training data. This is also well consistent with our empirical observation in Section 3.1, where the noise we used is approximately $0.04\epsilon$, close to the theoretical noise level of $0.03\epsilon$, showing that the FID of the generated images can be improved with a small corruption.

# 5 Improving Diffusion Models with Conditional Embedding Perturbation

## 5.1 Method

Our previous analysis demonstrates that slight condition corruption in the pre-training could potentially benefit both the image quality and diversity of DMs, which inspires us to improve the

Table 1: Pre-training results of IN-1K and MS-COCO using diffusion models pre-trained with perturbation. CEP achieves the best results (in bold).

| Model | Perturb. | FID ($\downarrow$) | IS ($\uparrow$) | Precision ($\uparrow$) | Recall ($\uparrow$) |
|---|---|---|---|---|---|
| LDM-4 [9] IN-1K ($s=2.0$) | - | 9.44 | 138.46 | 0.71 | 0.43 |
| | IP | 9.18 | 141.77 | 0.67 | 0.43 |
| | CEP-U | 7.00 | 170.73 | 0.73 | **0.45** |
| | CEP-G | **6.91** | **180.77** | **0.76** | 0.44 |
| DiT-XL/2 [11] IN-1K ($s=1.75$) | - | 6.76 | 179.67 | 0.74 | 0.46 |
| | IP | 6.75 | 182.28 | 0.75 | 0.45 |
| | CEP-U | **5.51** | **189.94** | **0.77** | 0.46 |
| | CEP-G | 5.92 | 185.21 | 0.75 | 0.45 |
| LDM-4 [9] CC3M ($s=3.0$) | - | 19.85 | 30.09 | 0.61 | 0.42 |
| | IP | 19.48 | 30.17 | 0.59 | 0.42 |
| | CEP-U | **17.93** | **30.77** | 0.65 | **0.41** |
| | CEP-G | 18.59 | 30.50 | **0.67** | 0.39 |
| LCM-v1.5 [52] CC3M ($s=4.5$) | - | 23.59 | 39.15 | 0.67 | 0.35 |
| | IP | 23.63 | 40.07 | 0.65 | 0.35 |
| | CEP-U | **22.91** | **40.31** | 0.67 | 0.35 |
| | CEP-G | 23.40 | 40.12 | **0.68** | **0.36** |

Table 2: ControlNet personalization results of IN-100 using LDMs pre-trained with perturbation. CEP achieves the best results (in bold).

| Control | Perturb. | FID ($\downarrow$) | IS ($\uparrow$) | Precision ($\uparrow$) | Recall ($\uparrow$) |
|---|---|---|---|---|---|
| IN-1K Canny ($s=2.25$) | - | 11.59 | 57.01 | 0.82 | 0.61 |
| | IP | 12.31 | 57.39 | 0.77 | 0.59 |
| | CEP-U | **11.46** | **59.29** | **0.84** | 0.58 |
| | CEP-G | 11.53 | 57.59 | 0.83 | **0.61** |
| IN-1K SAM ($s=2.25$) | - | 13.74 | 54.52 | 0.79 | 0.49 |
| | IP | 13.61 | 55.13 | 0.75 | 0.48 |
| | CEP-U | **12.95** | 56.68 | 0.79 | **0.50** |
| | CEP-G | 13.44 | 56.81 | **0.80** | 0.49 |
| CC3M Canny ($s=5.0$) | - | 40.65 | 32.56 | 0.63 | 0.51 |
| | IP | 40.12 | 32.43 | 0.62 | 0.52 |
| | CEP-U | 35.91 | 33.86 | 0.71 | 0.51 |
| | CEP-G | **34.57** | **36.59** | 0.68 | **0.53** |
| CC3M SAM ($s=4.0$) | - | 42.64 | 32.00 | 0.63 | 0.51 |
| | IP | 43.79 | 32.17 | 0.64 | 0.49 |
| | CEP-U | 38.00 | 32.98 | 0.67 | 0.51 |
| | CEP-G | **35.02** | **35.77** | 0.67 | **0.53** |

pre-training of DMs using this conclusion. In practice, it is usually infeasible to directly corrupt the conditions in the pre-training datasets either due to their large-scale nature or difficulties to select which conditions to corrupt. Instead, we propose to add the perturbation directly to the *conditional embeddings* of DMs, which is termed *conditional embedding perturbation (CEP)*. Compared to the fixed proportion of condition corruption in datasets we studied before, CEP adds perturbation to every data instance during training on the fly. Specifically, CEP slightly modifies the DM objective:

$$\mathcal{L}_{\mathrm{DM}} = \mathbb{E}_{\mathbf{x},y,\boldsymbol{\epsilon}\sim\mathcal{N}(0,\mathbf{I}),t\sim\mathcal{U}(1,T)} \left[ \|\boldsymbol{\epsilon} - \boldsymbol{\epsilon}_\theta\left(\mathbf{x}_t, t, \mathbf{c}_\theta(y) + \boldsymbol{\delta}\right)\|_2^2 \right], \tag{8}$$

where $\boldsymbol{\delta}$ denotes the perturbation added to conditional embeddings $\mathbf{c}_\theta(y)$. We simply set the perturbation to Uniform, i.e., $\boldsymbol{\delta} \sim \mathcal{U}\left(-\frac{\gamma}{\sqrt{d}}\mathbf{I}, \frac{\gamma}{\sqrt{d}}\mathbf{I}\right)$, or to Gaussian, i.e., $\boldsymbol{\delta} \sim \mathcal{N}\left(0, \frac{\gamma}{\sqrt{d}}\mathbf{I}\right)$, where the design of the factor $\frac{\gamma}{\sqrt{d}}$ mainly follows previous works [82, 92–95], $d$ denotes the dimension of $\mathbf{c}_\theta(y)$, and $\gamma$ controls the perturbation magnitude, mimicking the corruption ratio $\eta$. The main purpose of CEP is to learn better DMs with perturbation on relatively clean and heavily filtered datasets, such as CC3M and IN-1K studied in this paper, but it is also applicable to slightly corrupted datasets. Recently, Ning et al. [96] found that adding input perturbations (IP) to latent variables $\mathbf{z}_t$ during the forward process also helps diffusion training by mitigating exposure bias [97]. Compared to IP, CEP does not alter the marginal data distribution, but encourages the learned joint distribution to be more diverse.

## 5.2 Experiments

**Setup**. We pre-trained previous class-conditional LDM-4, text-conditional LDM-4, class-conditional DiT-XL/2, and text-conditional LCM-v1.5 with CEP, and compare with IP and clean pre-trained ones. We use both Uniform and Gaussian perturbation, denoted as CEP-U and CEP-G, respectively. We set $\gamma = 1$ for all models, with an ablation study with class-conditional LDM-4 with different $\gamma$s.

We evaluated the pre-trained class-conditional models on IN-1K and and text-conditional models on MS-COCO with FID, IS, Precision, and Recall. Additionally, we personalize the pre-trained LDMs with ControlNet on IN-100 to validate the effectiveness of CEP pre-training at downstream.

**Results**. We present the pre-training results of CEP in Table 1. CEP significantly and universally improves the performance for different class and text-conditional DMs, e.g., **2.53** and **1.25** FID improvement, and **42.31** and **10.27** IS improvement of LDM-4 and DiT-XL/2. CEP also improves precision and recall of DMs. In contrast, IP only achieves marginal improvement and yields slightly worse precision. Adopting CEP in pre-training also benefits the personalization tasks, especially for text-conditional LDMs, with FID improvement

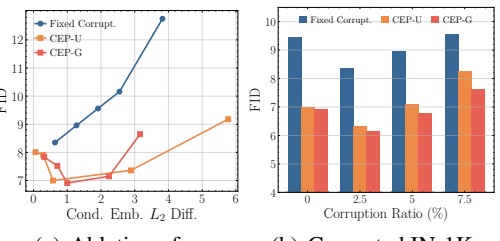

(a) Ablation of $\gamma$     (b) Corrupted IN-1K

Figure 8: Ablation with LDM-4 IN-1K. (a) FID and average $L_2$ distance of conditional embeddings against clean ones with $\gamma = \{0.1, 0.5, 1.0, 5.0, 10.0\}$, indicated by square points (left to right). We compare with fixed synthetic corruption $\eta = \{2.5, 5, 10, 15\}\%$, shown by circle points. (b) CEP on corrupted IN-1K.

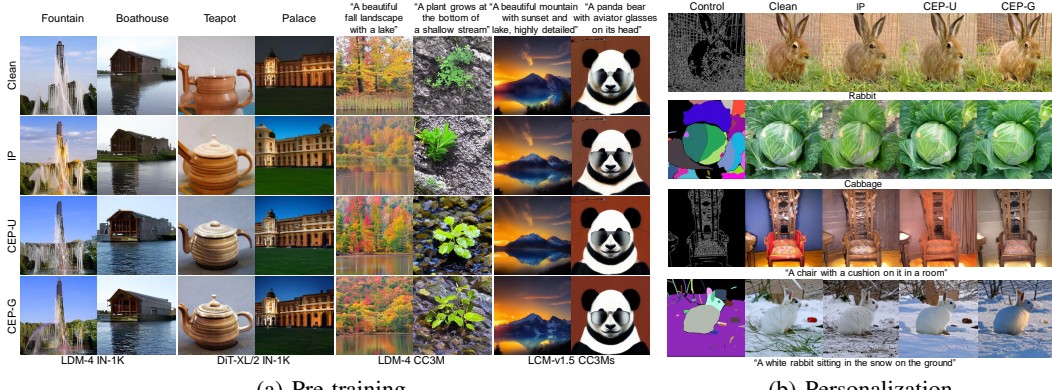

| (a) Pre-training | (b) Personalization |

Figure 9: Comparison of DMs pre-trained with CEP against IP and without perturbation.

of **6.08** and **7.02** for Canny and SAM spatial control, as shown in Table 2. Qualitatively, as shown in Fig. 9, images generated from DMs with CEP also look more visually appealing and realistic.

The ablation results of $\gamma$ are shown in Fig. 8(a). We also compare the average $L_2$ distance of CEP and fixed corruption against the clean condition embeddings. Interestingly, one can observe that CEP achieves a lower FID with more corruption in the embedding space (larger $L_2$ from the clean ones), demonstrating its effectiveness. CEP is applicable to slightly corrupted datasets that we may often encounter in practice, as shown in Fig. 8(b), where it also facilitates the performance significantly.

In addition, we also compare the proposed CEP with traditional regularization methods, such as Dropout [98] and Label Smoothing [99], and study the effects of fixed and random perturbation during training in Appendix E.3 and Appendix E.4. The results show that CEP is more effective.

# 6 Related Work

**Diffusion Models**. Inspired by thermodynamics, DMs were first proposed by Sohl-Dickstein et al. [100]. DMs have soon been developed into image generation with a fixed Gaussian noise diffusion process [1, 101]. Various techniques have then been proposed for more effective and efficient DMs [102, 4, 5]. One of the most well-known is modeling the diffusion process at the latent space of pre-trained image encoders as a strong prior [58, 56], instead of raw pixels spaces [3, 9, 11], which allows for high-quality image generation with affordable inference speed. Numerous foundational DMs that generate photorealistic images have thus been built [103–107, 10, 108, 26, 109]. These powerful models are generally pre-trained on web-crawled billion-scale data with conditions (usually text), which may inevitably contain corruption [31, 110, 111, 32, 35]. Recently, consistency models [51, 112, 52] were also developed from DMs, allowing generation with much fewer inference steps. These foundational DMs also enabled many downstream applications [20, 21, 113–122]. However, the effects of the pre-training corruption on downstream applications remain unknown.

**Learning with Noise**. Learning with noise is a long-standing challenge [123–126]. Noisy label learning has been widely studied in classification, from noise correction [127, 40, 128–131, 41, 132, 133, 43, 134, 44, 135] and noise-robust loss functions [39, 136–141, 29, 142]. Learning with noise has also been studied in the context of generative models [143–146]. Robust GANs and DMs [45–47] alleviated the quality degradation and condition misalignment of training generative models with label noise. In contrast, we study a more practical scenario, where the models are trained on corrupted pre-training data with a low noise ratio, and then adapted to downstream tasks.

In fact, more aligned with our work, there are several recent studies on exploring and exploiting the pre-training noise. Chen et al. [48, 49] found that slight label noise in supervised pre-training can be beneficial for in-domain downstream tasks, whereas detrimental for out-of-domain tasks. NoisyTune [147], NEFTune [95], and SymNoise [148] found that introducing noise to the weights and embedding of pre-trained language models can facilitate downstream performance. Ning et al. [96] also found that adding perturbation in the forward diffusion process helps reduce the exposure bias of DMs [97]. Similarly, Naderi et al. [149] introduced noise into the input of image translation

networks for better learning with limited data. Synthetic data (potentially with corruption) have also been found to be useful in pre-training [24, 150]. We demonstrate that slight corruption in conditions of the pre-training DMs can also be beneficial at both the pre-training and downstream.

# 7 Conclusion and Limitation

We presented the first comprehensive study on condition corruption in pre-training of DMs. Our empirical and theoretical analysis surprisingly demonstrate that slight condition corruption benefits DMs in both the pre-training and downstream adaptation, based on which we proposed CEP as a simple yet general technique that significantly improves the performance of DMs. We hope our findings could inspire more future work on understanding the pre-training data of foundation models.

This work has the following limitations. First, due to a lack of computing resources, we cannot study all types of DMs on larger datasets. Second, the theoretical analysis is based on several assumptions that might be further explored in the future. Third, the evaluation of image generation remains an open question, and we used most of the existing criteria for fair comparison.

# Disclaimer

While we study DMs for image generation in this paper, it is important to note that all generations have been selected and verified by human experts to ensure that they are responsible. Although we release all the pre-trained models under different corruption settings, it is possible that these models will generate inappropriate content due to the scale of pre-training and without alignment with human preferences. The main purpose of this research is to raise the awareness of the community on data cleaning and corruption in the research of diffusion models.

# Acknowledge

MS was supported by the Institute for AI and Beyond, UTokyo. DZ was supported by NSFC 62306252, Guangdong NSF 2024A1515012444, and Hong Kong ECS awards 27309624.

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

# Appendix

## Contents

# A Derivations and Proofs

In this section, we theoretically investigate the behavior of condition corruption on DMs. Our investigation encompasses the DDIM samplers with continuous-time processes. We focus on how slight conditional embedding corruption can affect the training process of DMs, as well as the consequences on the generative process. As our theoretical analysis indicates, slight conditional embedding corruption will benefit both the generation quality and diversity, which aligns with the experimental conclusions in Section 3.

## A.1 Preliminaries

We start by giving a concise overview of the problem setup.

**Data Distribution.** For precise theoretical characterizations, we concentrate on the prototypical problem of sampling from Gaussian mixture models (GMMs). Specifically, we consider that the distribution of the data $\mathbf{x} \in \mathbb{R}^d$ satisfies

$$\mathbb{P}(\mathbf{x}) := \sum_{y \in \mathcal{Y}} w_y \mathcal{N}(\boldsymbol{\mu}_y, \mathbf{I}). \tag{9}$$

Here, $y$ is denoted as the class labels with a finite set of values $y \in \{1, 2, \cdots, |\mathcal{Y}|\}$. Given any class label $y \in \mathcal{Y}$, the data distribution $\mathbf{x}|y$ is a Gaussian with the center and covariance as $(\boldsymbol{\mu}_y, \mathbf{I})$. And the positive $w_y$ represents the weights of the Gaussian components which satisfies $\sum_{y \in \mathcal{Y}} w_y = 1$.

The diffusion model is a two processes framework: a forward process that transforms the target distribution into Gaussian noise, and a reverse process that progressively denoises in order to reconstitute the original target distribution. In this paper, we consider the continuous-time processes and define the forward process as an Ornstein–Uhlenbeck (OU) process:

**Forward Process.** At time step $t \in [0, T]$, the forward process is

$$d\mathbf{x}_t = -\mathbf{x}_t d_t + \sqrt{2} d\mathbf{w}_t, \quad \mathbf{x}_0 = \mathbf{x} \sim \mathbb{P}(\cdot|y), \tag{10}$$

where $\mathbf{w}_t$ is a $d$-dimensional standard Brownian motion.

The advantage of considering the forward process as an OU process is that it enables us to directly derive the closed-form expression for the conditional sample distribution at any given time $t$.

$$\mathbf{x}_t|\mathbf{x} \sim \mathcal{N}(r_t \mathbf{x}, \sigma_t^2 \mathbf{I}), \tag{11}$$

where $r_t = e^{-t}$ and $\sigma_t = \sqrt{1 - e^{-2t}}$.

By the reparameterization trick, $\mathbf{x}_t$ can be represented as

$$\mathbf{x}_t = r_t \mathbf{x} + \sigma_t \boldsymbol{\epsilon}. \tag{12}$$

We explore the widely adopted sampling method, DDIM, augmented with classifier-free guidance. And the associated reverse process is stated as the following ODE implementation:

**Reverse Process.** We write the reverse process in a forward version by switching time direction $t \to T - t$ as

$$d\mathbf{z}_t = \Big(\mathbf{z}_t + (1+w)\nabla_{\mathbf{z}_t} \log \mathbb{P}(\mathbf{z}_t|y) - w\nabla_{\mathbf{z}_t} \log \mathbb{P}(\mathbf{z}_t)\Big)dt, \quad \mathbf{z}_0 \sim \mathcal{N}(\mathbf{0}, \mathbf{I}), \tag{13}$$

where $w \geq 0$ is a hyperparameter that controls the strength of the classifier guidance.

Given our primary focus on the impact of corrupted conditional embedding on generation, we simplify the reverse process by setting $w$ to 0 and concentrate solely on the conditional score network, i.e.,

$$d\mathbf{z}_t = \Big(\mathbf{z}_t + \nabla_{\mathbf{z}_t} \log \mathbb{P}(\mathbf{z}_t|y)\Big)dt. \tag{14}$$

The remaining task is to estimate the unknown conditional score function $\nabla_{\mathbf{z}_t} \log \mathbb{P}(\mathbf{z}_t|y)$ and unconditional score function $\nabla_{\mathbf{z}_t} \log \mathbb{P}(\mathbf{z}_t|y)$ via training. In the subsequent analysis, we will indicate that by minimizing Equation (3) and optimizing the denoising network $\boldsymbol{\epsilon}$, we can achieve an estimate of the conditional score.

**Denoising Networks.** Inspired by recent work that also target on GMMs [87, 88], we parameterize the denoising networks as the following piecewise linear function:

$$\epsilon_\theta(\mathbf{x}_t, y) = \sum_{k=1}^{|\mathcal{Y}|} \mathbf{1}_{y=k}\left(\mathbf{W}_t^k \mathbf{x}_t + \mathbf{V}_t^k \mathbf{c}(y)\right), \tag{15}$$

where $\mathbf{c}(y)$ is the one-hot encoding of label $y$, $\mathbf{1}_y$ is an indicator function and $\{\mathbf{W}_t^k, \mathbf{V}_t^k\}_{k=1}^{|\mathcal{Y}|}$ are trainable parameters.

**Conditional Embedding Corruption.** We examine the scenario of conditional embedding corruption, wherein the conditional embedding $\mathbf{c}(y)$ is no longer matched with the data $\mathbf{x}$, but instead is perturbed by Gaussian noise. Consequently, the corrupted conditional embedding causes the denoising networks to be as follows:

$$\epsilon_\theta^c(\mathbf{x}_t, y) = \sum_{k=1}^{|\mathcal{Y}|} \mathbb{1}_{y=k}\left(\mathbf{W}_t^k \mathbf{x}_t^k + \mathbf{V}_t^k(\mathbf{c}(y) + \gamma\boldsymbol{\xi})\right), \tag{16}$$

where the $d$-dimensional corrupted noise $\boldsymbol{\xi} \sim \mathcal{N}(\mathbf{0}, \mathbf{I})$ and $\gamma \geq 0$ serves as the corruption control parameter, mirroring the noise ratio $\eta$ for direct control of noise magnitude.

Based on the aforementioned setup, our ultimate goal is to obtain the closed form of the final generated data distribution by considering the impact of corrupted noise $\boldsymbol{\xi}$ on the training and generative processes. By comparing the differences in data distribution before and after the addition of corrupted noise, we aim to accurately characterize the impact of embedding corruption on image generation and explain the phenomena observed in experiments.

## A.2 The Estimation of Conditional Score

### A.2.1 Clean Conditions

We first analyze the case of clean conditional embedding. Note that the denoising networks $\epsilon_\theta$ is a piecewise linear function and the training objective can be represented as

$$\frac{1}{n}\sum_{i=1}^{n} \mathbb{E}_{\boldsymbol{\epsilon}}[\|\epsilon_\theta(\mathbf{x}_{t,i}, y) - \boldsymbol{\epsilon}\|_2^2] = \sum_{k=1}^{|\mathcal{Y}|} \frac{w_k}{n_k}\sum_{y_i=k} \mathbb{E}_{\boldsymbol{\epsilon}}[\|\epsilon_\theta(\mathbf{x}_{t,i}, y) - \boldsymbol{\epsilon}\|_2^2], \tag{17}$$

where the class sample size $n_k := n w_k$.

We observed that the optimization objective in Equation (17) can be divided into $|\mathcal{Y}|$ independent sub-problems based on the label $y$. This inspires us to analyze the training and generation processes according to different classes.

Given any class $k \in \mathcal{Y}$, we present the following lemma for determining the optimal parameters of the corresponding denoising network and the associated conditional score.

**Lemma 1.** *(Clean Conditional Embedding). Given any class $k \in \mathcal{Y}$, the optimal linear denoising network is*

$$\epsilon_\theta(\mathbf{x}_t, y = k) = \sigma_t\left(\sigma_t^2\mathbf{I} + r_t^2\boldsymbol{\Sigma}_k\right)^{-1}\mathbf{x}_t - r_t\sigma_t\left(\sigma_t^2\mathbf{I} + r_t^2\boldsymbol{\Sigma}_k\right)^{-1}\hat{\boldsymbol{\mu}}_k. \tag{18}$$

*And the corresponding optimal linear estimation of conditional score $\nabla_{\mathbf{x}_t} \log \mathbb{P}(\mathbf{x}_t|y = k)$ is*

$$\nabla_{\mathbf{x}_t} \log \mathbb{P}(\mathbf{x}_t|y = k) = -\frac{\epsilon_\theta(\mathbf{x}_t, y = k)}{\sigma_t}, \tag{19}$$

*where $\boldsymbol{\Sigma}_k := \frac{1}{n_k}\sum_{i=1}^{n_k} \mathbf{x}_i\mathbf{x}_i^\mathsf{T} - \frac{1}{n_k^2}\sum_{i=1}^{n_k} \mathbf{x}_i \sum_{i=1}^{n_k} \mathbf{x}_i^\mathsf{T}$ is the empirical covariance of $k$-labeled dataset and $\hat{\boldsymbol{\mu}}_k := \frac{1}{n_k}\sum_{i=1}^{n_k} \mathbf{x}_i$ is the empirical mean of $k$-labeled dataset, $n_k$ is the sample size of $k$-labeled dataset. And $r_t = e^{-t}$, $\sigma_t = \sqrt{1 - e^{-2t}}$.*

*Proof.* Given any class $k$, the training objective is

$$\frac{w_k}{n_k}\sum_{y_i=k} \mathbb{E}_{\boldsymbol{\epsilon}}[\|\epsilon(\mathbf{x}_{t,i}, y) - \boldsymbol{\epsilon}\|_2^2] = \frac{w_k}{n_k}\sum_{y_i=k} \mathbb{E}_{\boldsymbol{\epsilon}}[\|\mathbf{W}_t^k r_t\mathbf{x}_i + \mathbf{W}_t^k\sigma_t\boldsymbol{\epsilon} + \mathbf{V}_t^k\mathbf{c}(y) - \boldsymbol{\epsilon}\|_2^2]. \tag{20}$$

Since the weights $w_k$ is fixed in our analysis, we omit the notation $w_k$ without ambiguity in the following contents and for enhanced clarity, we initially omit the subscript/superscript $k$ in $n_k$, $\mathbf{W}_t^k$, $\mathbf{V}_t^k$, $\hat{\boldsymbol{\mu}}_k$, $\boldsymbol{\mu}_k$ and $\boldsymbol{\Sigma}_k$. Then Equation (20) can restated as as Equation (21) for simplicity.

$$\frac{1}{n} \sum_{i=1}^{n} \mathbb{E}_{\boldsymbol{\epsilon}}[\|\mathbf{W}_t r_t \mathbf{x}_i + \mathbf{W}_t \sigma_t \boldsymbol{\epsilon} + \mathbf{V}_t \mathbf{c}(y) - \boldsymbol{\epsilon}\|_2^2], \tag{21}$$

where $\mathbf{x}_i$ is labeled as $k$.

This training loss can be further simplified as

$$\frac{1}{n} \sum_{i=1}^{n} \mathbb{E}_{\boldsymbol{\epsilon}}[\|\mathbf{W}_t r_t \mathbf{x}_i + \mathbf{W}_t \sigma_t \boldsymbol{\epsilon} + \mathbf{V}_t \mathbf{c}(y) - \boldsymbol{\epsilon}\|_2^2]$$

$$= \frac{1}{n} \sum_{i=1}^{n} \|\mathbf{W}_t r_t \mathbf{x}_i + \mathbf{V}_t \mathbf{c}(y)\|_2^2 + \mathbb{E}_{\boldsymbol{\epsilon}}[\|(\mathbf{W}_t \sigma_t - \mathbf{I})\boldsymbol{\epsilon}\|_2^2]$$

$$= \frac{r_t^2}{n} \sum_{i=1}^{n} \mathbf{x}_i^\mathsf{T} \mathbf{W}_t^\mathsf{T} \mathbf{W}_t \mathbf{x}_i + \frac{2r_t}{n} \mathbf{e}^\mathsf{T}(y) \mathbf{V}_t^\mathsf{T} \mathbf{W}_t \sum_{i=1}^{n} \mathbf{x}_i + \mathbf{e}^\mathsf{T}(y) \mathbf{V}_t^\mathsf{T} \mathbf{V}_t \mathbf{c}(y) + \mathrm{Tr}\Big((\mathbf{W}_t \sigma_t - \mathbf{I})(\mathbf{W}_t^\mathsf{T} \sigma_t - \mathbf{I})\Big).$$

For simplicity, we denote $\mathbf{b}_t := \mathbf{V}_t \mathbf{c}(y)$ and we get

$$\frac{1}{n} \sum_{i=1}^{n} \mathbb{E}_{\boldsymbol{\epsilon}}[\|\mathbf{W}_t r_t \mathbf{x}_i + \mathbf{W}_t \sigma_t \boldsymbol{\epsilon} + \mathbf{V}_t \mathbf{c}(y) - \boldsymbol{\epsilon}\|_2^2]$$

$$= \frac{r_t^2}{n} \sum_{i=1}^{n} \mathbf{x}_i^\mathsf{T} \mathbf{W}_t^\mathsf{T} \mathbf{W}_t \mathbf{x}_i + \frac{2r_t}{n} \mathbf{b}_t^\mathsf{T} \mathbf{W}_t \sum_{i=1}^{n} \mathbf{x}_i + \mathbf{b}_t^\mathsf{T} \mathbf{b}_t + \mathrm{Tr}\Big((\mathbf{W}_t \sigma_t - \mathbf{I})(\mathbf{W}_t^\mathsf{T} \sigma_t - \mathbf{I})\Big).$$

Then, we define the loss function

$$J(\mathbf{W}_t, \mathbf{b}_t) = \frac{r_t^2}{n} \sum_{i=1}^{n} \mathbf{x}_i^\mathsf{T} \mathbf{W}_t^\mathsf{T} \mathbf{W}_t \mathbf{x}_i + \frac{2r_t}{n} \mathbf{b}_t^\mathsf{T} \mathbf{W}_t \sum_{i=1}^{n} \mathbf{x}_i + \mathbf{b}_t^\mathsf{T} \mathbf{b}_t + \mathrm{Tr}\Big((\mathbf{W}_t \sigma_t - \mathbf{I})(\mathbf{W}_t^\mathsf{T} \sigma_t - \mathbf{I})\Big).$$

The optimal $\mathbf{W}_t^*$ and $\mathbf{b}_t^*$ can be obtained by taking gradient to $J(\mathbf{W}_t, \mathbf{b}_t)$ such that,

$$\mathbf{0} = \nabla_{\mathbf{W}_t} J(\mathbf{W}_t, \mathbf{b}_t) = \frac{2r_t^2}{n_k} \mathbf{W}_t \sum_{i=1}^{n} \mathbf{x}_i \mathbf{x}_i^\mathsf{T} + \frac{2r_t}{n} \mathbf{b}_t \sum_{i=1}^{n} \mathbf{x}^\mathsf{T} + 2(\sigma_t^2 \mathbf{W}_t - \sigma_t \mathbf{I}),$$

$$\mathbf{0} = \nabla_{\mathbf{b}_t} J(\mathbf{W}_t, \mathbf{b}_t) = \frac{2r_t}{n} \mathbf{W}_t \sum_{i=1}^{n} \mathbf{x}_i + 2\mathbf{b}_t.$$

And the optimal $\mathbf{W}_t^*$ and $\mathbf{b}_t^*$ is,

$$\mathbf{W}_t^* = \sigma_t \Big(\sigma_t^2 \mathbf{I} + r_t^2 \boldsymbol{\Sigma}\Big)^{-1}, \quad \mathbf{b}_t^* = \mathbf{V}_t^* \mathbf{c}(y) = -r_t \sigma_t \Big(\sigma_t^2 \mathbf{I} + r_t^2 \boldsymbol{\Sigma}\Big)^{-1} \hat{\boldsymbol{\mu}}.$$

where $\boldsymbol{\Sigma} := \frac{1}{n} \sum_{i=1}^{n} \mathbf{x}_i \mathbf{x}_i^\mathsf{T} - \frac{1}{n_k^2} \sum_{i=1}^{n} \mathbf{x}_i \sum_{i=1}^{n} \mathbf{x}_i^\mathsf{T}$ is the empirical covariance of $k$-labeled dataset and $\hat{\boldsymbol{\mu}} := \frac{1}{n} \sum_{i=1}^{n} \mathbf{x}_i$ is the empirical mean of $k$-labeled dataset.

Based on above analyze, for class $k$, we then have the following optimal denoising network as the linear estimator of noise $\boldsymbol{\epsilon}$,

$$\boldsymbol{\epsilon}_\theta(\mathbf{x}_t, y = k) = \sigma_t \Big(\sigma_t^2 \mathbf{I} + r_t^2 \boldsymbol{\Sigma}\Big)^{-1} \mathbf{x}_t - r_t \sigma_t \Big(\sigma_t^2 \mathbf{I} + r_t^2 \boldsymbol{\Sigma}\Big)^{-1} \hat{\boldsymbol{\mu}}.$$

Given any class $k$, we derive

$$\nabla_{\mathbf{x}_t} \log \mathbb{P}(\mathbf{x}_t|\mathbf{x}, y = k) = -\frac{(\mathbf{x}_t - r_t \mathbf{x})}{\sigma_t^2} = -\frac{\boldsymbol{\epsilon}}{\sigma_t},$$

where data $\mathbf{x}$ and $\mathbf{x}_t$ are labeled with $k$.

Therefore, the linear estimator of $\nabla_{\mathbf{x}_t} \log \mathbb{P}(\mathbf{x}_t | \mathbf{x}, y = k)$ given $k$-labeled data $\mathbf{x}_t$ is

$$-\left(\sigma_t^2 \mathbf{I} + r_t^2 \mathbf{\Sigma}\right)^{-1} \mathbf{x}_t + r_t \left(\sigma_t^2 \mathbf{I} + r_t^2 \mathbf{\Sigma}\right)^{-1} \hat{\boldsymbol{\mu}}.$$

Since the estimation of $\nabla_{\mathbf{x}_t} \log \mathbb{P}(\mathbf{x}_t | \mathbf{x}, y = k)$ and $\nabla_{\mathbf{x}_t} \log \mathbb{P}(\mathbf{x}_t | y = k)$ are equivalent in optimization, the optimal linear estimator $\nabla_{\mathbf{x}_t} \log \mathbb{P}(\mathbf{x}_t | y = k)$ also can be

$$\nabla_{\mathbf{x}_t} \log \mathbb{P}(\mathbf{x}_t | y = k) = -\frac{\boldsymbol{\epsilon}_\theta(\mathbf{x}_t, y = k)}{\sigma_t}$$

$$= -\left(\sigma_t^2 \mathbf{I} + r_t^2 \mathbf{\Sigma}\right)^{-1} \mathbf{x}_t + r_t \left(\sigma_t^2 \mathbf{I} + r_t^2 \mathbf{\Sigma}\right)^{-1} \hat{\boldsymbol{\mu}}.$$

The proof is completed. $\qquad\square$

Note that the ground truth functions of conditional score given the label $k$ is

$$\nabla_{\mathbf{x}_t} \log \mathbb{P}(\mathbf{x}_t | y = k) = -\mathbf{x}_t + r_t \boldsymbol{\mu}_k. \tag{22}$$

We note the similarities in form and coefficients between the optimal linear estimator (19) and ground truth (22); they both are linear combinations of $\boldsymbol{\mu}_k / \hat{\boldsymbol{\mu}}_k$ and $\mathbf{x}_t$. Specifically, when the empirical covariance $\mathbf{\Sigma}_k$ equals the population covariance $\mathbf{I}$ and the empirical mean $\hat{\boldsymbol{\mu}}_k$ matches the expected value $\boldsymbol{\mu}_k$, the estimated conditional score in Equation (19) coincides with the true conditional score in Equation (22).

### A.2.2 Corrupted Conditions

The same analytical approach as in Section A.2.1 will be applied to the scenario where the conditional embedding is perturbed. Given the class $k$, we propose the following Lemma 2 to describe the optimal linear denoising network and the conditional score estimator with the corrupted conditional embedding $(\mathbf{c}(y) + \gamma \boldsymbol{\xi})$ where $\boldsymbol{\xi}$ is the standard Gaussian noise.

**Lemma 2.** *(Corrupted Conditional Embedding). Given any class $k \in \mathcal{Y}$, the optimal linear denoising network is*

$$\boldsymbol{\epsilon}_\theta^c(\mathbf{x}_t, y = k) = \sigma_t \left(\sigma_t^2 \mathbf{I} + r_t^2 \mathbf{\Sigma}_k + \frac{r_t^2 \gamma^2}{1 + \gamma^2} \|\hat{\boldsymbol{\mu}}_k\|_2^2 \mathbf{I}\right)^{-1} \mathbf{x}_t - \frac{r_t}{1 + \gamma^2} \sigma_t \left(\sigma_t^2 \mathbf{I} + r_t^2 \mathbf{\Sigma}_k + \frac{r_t^2 \gamma^2}{1 + \gamma^2} \|\hat{\boldsymbol{\mu}}_k\|_2^2 \mathbf{I}\right)^{-1} \hat{\boldsymbol{\mu}}_k. \tag{23}$$

*And the corresponding optimal linear estimation of conditional score $\nabla_{\mathbf{x}_t} \log \mathbb{P}^c(\mathbf{x}_t | y = k)$ is*

$$\nabla_{\mathbf{x}_t} \log \mathbb{P}^c(\mathbf{x}_t | y = k) = -\frac{\boldsymbol{\epsilon}_\theta^c(\mathbf{x}_t, y = k)}{\sigma_t} \tag{24}$$

*where $\mathbf{\Sigma}_k := \frac{1}{n_k} \sum_{i=1}^{n_k} \mathbf{x}_i \mathbf{x}_i^\mathsf{T} - \frac{1}{n_k^2} \sum_{i=1}^{n_k} \mathbf{x}_i \sum_{i=1}^{n_k} \mathbf{x}_i^\mathsf{T}$ is the empirical covariance of $k$-labeled dataset and $\hat{\boldsymbol{\mu}}_k := \frac{1}{n_k} \sum_{i=1}^{n_k} \mathbf{x}_i$ is the empirical mean of $k$-labeled dataset, $n_k$ is the sample size of $k$-labeled dataset, and $r_t = e^{-t}, \sigma_t = \sqrt{1 - e^{-2t}}$.*

*Proof.* For enhanced clarity, we initially omit the subscript/superscript $k$ in $n_k$, $\mathbf{W}_t^k$, $\mathbf{V}_t^k$, $\hat{\boldsymbol{\mu}}_k$, $\boldsymbol{\mu}_k$ and $\mathbf{\Sigma}_k$.

For any class $k$ and standard Gaussian noise $\boldsymbol{\xi}$, we consider the following training loss

$$\frac{1}{n} \sum_{i=1}^n \mathbb{E}_{\boldsymbol{\xi}} \mathbb{E}_{\boldsymbol{\epsilon}} [\|\mathbf{W}_t r_t \mathbf{x}_i + \mathbf{W}_t \sigma_t \boldsymbol{\epsilon} + \mathbf{V}_t (\mathbf{c}(y) + \gamma \boldsymbol{\xi}) - \boldsymbol{\epsilon}\|_2^2]. \tag{25}$$

And we further optimize the training loss as

$$\frac{1}{n} \sum_{i=1}^n \mathbb{E}_{\boldsymbol{\xi}} \mathbb{E}_{\boldsymbol{\epsilon}} [\|\mathbf{W}_t r_t \mathbf{x}_i + \mathbf{W}_t \sigma_t \boldsymbol{\epsilon} + \mathbf{V}_t (\mathbf{c}(y) + \gamma \boldsymbol{\xi}) - \boldsymbol{\epsilon}\|_2^2]$$

$$= \frac{1}{n} \sum_{i=1}^n \|\mathbf{W}_t r_t \mathbf{x}_i + \mathbf{V}_t \mathbf{c}(y) + \gamma \mathbf{V}_t \boldsymbol{\xi}\|_2^2 + \mathbb{E}_{\boldsymbol{\epsilon}} [\|(\mathbf{W}_t \sigma_t - \mathbf{I}) \boldsymbol{\epsilon}\|_2^2]$$

$$= \frac{r_t^2}{n} \sum_{i=1}^n \mathbf{x}_i^\mathsf{T} \mathbf{W}_t^\mathsf{T} \mathbf{W}_t \mathbf{x}_i + \frac{2 r_t}{n} \mathbf{e}^\mathsf{T}(y) \mathbf{V}_t^\mathsf{T} \mathbf{W}_t \sum_{i=1}^n \mathbf{x}_i + \mathbf{e}^\mathsf{T}(y) \mathbf{V}_t^\mathsf{T} \mathbf{V}_t \mathbf{c}(y)$$

$$+ \gamma^2 \mathrm{Tr}(\mathbf{V}_t \mathbf{V}_t^\mathsf{T}) + \mathrm{Tr}\left((\mathbf{W}_t \sigma_t - \mathbf{I})(\mathbf{W}_t^\mathsf{T} \sigma_t - \mathbf{I})\right).$$

Similarly, we get the optimal $\mathbf{W}_t^*$ and $\mathbf{b}_t^*$ is,

$$\mathbf{W}_t^* = \sigma_t \Big(\sigma_t^2 \mathbf{I} + r_t^2 \mathbf{\Sigma} + \frac{r_t^2 \gamma^2}{1+\gamma^2} \|\hat{\boldsymbol{\mu}}\|_2^2 \mathbf{I}\Big)^{-1},$$

$$\mathbf{b}_t^* = \mathbf{V}_t^* \mathbf{c}(y) = -\frac{r_t}{1+\gamma^2} \sigma_t \Big(\sigma_t^2 \mathbf{I} + r_t^2 \mathbf{\Sigma} + \frac{r_t^2 \gamma^2}{1+\gamma^2} \|\hat{\boldsymbol{\mu}}\|_2^2 \mathbf{I}\Big)^{-1} \hat{\boldsymbol{\mu}}.$$

where $\mathbf{\Sigma} := \frac{1}{n}\sum_{i=1}^n \mathbf{x}_i \mathbf{x}_i^\mathsf{T} - \frac{1}{n_k^2}\sum_{i=1}^n \mathbf{x}_i \sum_{i=1}^n \mathbf{x}_i^\mathsf{T}$ is the empirical covariance and $\hat{\boldsymbol{\mu}} := -\frac{1}{n}\sum_{i=1}^n \mathbf{x}_i$ is the empirical mean of $k$-labeled dataset.

So for any class $k$, we get the following optimal denoising network as the linear estimator of noise $\boldsymbol{\epsilon}$,

$$\boldsymbol{\epsilon}_\theta^c(\mathbf{x}_t, y=k) = \sigma_t \Big(\sigma_t^2 \mathbf{I} + r_t^2 \mathbf{\Sigma}_k + \frac{r_t^2 \gamma^2}{1+\gamma^2}\|\hat{\boldsymbol{\mu}}_k\|_2^2\mathbf{I}\Big)^{-1}\mathbf{x}_t - \frac{r_t}{1+\gamma^2}\sigma_t\Big(\sigma_t^2\mathbf{I} + r_t^2\mathbf{\Sigma}_k + \frac{r_t^2\gamma^2}{1+\gamma^2}\|\hat{\boldsymbol{\mu}}_k\|_2^2\mathbf{I}\Big)^{-1}\hat{\boldsymbol{\mu}}_k.$$

The corresponding optimal linear estimator $\nabla_{\mathbf{x}_t}\log\mathbb{P}^c(\mathbf{x}_t|y=k)$ is

$$\nabla_{\mathbf{x}_t}\log\mathbb{P}^c(\mathbf{x}_t|y=k) = -\frac{\boldsymbol{\epsilon}_\theta(\mathbf{x}_t, y=k)}{\sigma_t}$$

$$= -\Big(\sigma_t^2\mathbf{I} + r_t^2\mathbf{\Sigma} + \frac{r_t^2\gamma^2}{1+\gamma^2}\|\hat{\boldsymbol{\mu}}\|_2^2\mathbf{I}\Big)^{-1}\mathbf{x}_t + \frac{r_t}{1+\gamma^2}\Big(\sigma_t^2\mathbf{I} + r_t^2\mathbf{\Sigma} + \frac{r_t^2\gamma^2}{1+\gamma^2}\|\hat{\boldsymbol{\mu}}\|_2^2\mathbf{I}\Big)^{-1}\hat{\boldsymbol{\mu}}.$$

$\square$

## A.3 The Distribution of Generation

Given the class $k$, after getting the optimal linear estimator of conditional score $\nabla_{\mathbf{x}_t}\log\mathbb{P}(\mathbf{x}_t|y=k)$, we can accurately characterize the reverse process and derive the closed form of generation distribution.

### A.3.1 Clean Conditions

According to Lemma 1, we first replace the conditional score $\nabla_{\mathbf{x}_t}\log\mathbb{P}(\mathbf{x}_t|y=k)$ in Equation (14) with the optimal linear estimator. Given the class $k$, we get

$$d\mathbf{z}_t = \Big(\mathbf{z}_t - \big(\sigma_t^2\mathbf{I} + r_t^2\mathbf{\Sigma}_k\big)^{-1}\mathbf{z}_t + r_t\big(\sigma_t^2\mathbf{I} + r_t^2\mathbf{\Sigma}_k\big)^{-1}\hat{\boldsymbol{\mu}}_k\Big)dt. \tag{26}$$

Assume the empirical covariance $\mathbf{\Sigma}_k$ is full rank, since the empirical covariance is diagonalizable and semi-positive, we decompose $\mathbf{\Sigma}_k$ as

$$\mathbf{\Sigma}_k = \mathbf{U}\mathbf{\Lambda}\mathbf{U}^\mathsf{T}, \tag{27}$$

where $\mathbf{U} = [\mathbf{u}_1|\cdots|\mathbf{u}_d]$ is an orthogonal matrix whose columns are the real, orthonormal eigenvectors of $\mathbf{\Sigma}_k$ and $\mathbf{\Lambda} = \mathrm{diag}(\lambda_1,\ldots,\lambda_d)$ is a diagonal matrix whose entries are the eigenvalues arranged in descending order of $\mathbf{\Sigma}_k$, i.e., $1 \geq \lambda_1 \geq \cdots \geq \lambda_d > 0$.

Therefore, problem (26) can be decomposed into $d$ independent sub-problem as

$$\mathbf{u}_i^\mathsf{T} d\mathbf{z}_t = \Big(1 - \frac{1}{\sigma_t^2 + r_t^2\lambda_i}\Big)\mathbf{u}_i^\mathsf{T}\mathbf{z}_t dt + \frac{r_t}{\sigma_t^2 + r_t^2\lambda_i}\mathbf{u}_i^\mathsf{T}\hat{\boldsymbol{\mu}}_k dt, \tag{28}$$

where $\mathbf{u}_i$ is $i$-th eigenvector and $\lambda_i$ is its corresponding eigenvalue.

By analyzing Equation (28), we can obtain the expectation and variance of the final generation distribution separately, thereby inferring the distribution of the ultimately generated data as outlined in the subsequent lemma.

**Lemma 3.** *(Clean Conditional Embedding). For any class $k$, the distribution of the generated data $\mathbf{z}_T$ satisfies*

$$\lim_{T\to\infty}\mathbf{z}_T \sim \mathcal{N}(\hat{\boldsymbol{\mu}}_k, \mathbf{\Sigma}_k), \tag{29}$$

*where $\mathbf{\Sigma}_k := \frac{1}{n_k}\sum_{i=1}^{n_k}\mathbf{x}_i\mathbf{x}_i^\mathsf{T} - \frac{1}{n_k^2}\sum_{i=1}^{n_k}\mathbf{x}_i\sum_{i=1}^{n_k}\mathbf{x}_i^\mathsf{T}$ is the empirical covariance of $k$-labeled dataset and $\hat{\boldsymbol{\mu}}_k := \frac{1}{n_k}\sum_{i=1}^{n_k}\mathbf{x}_i$ is the empirical mean of $k$-labeled dataset, $n_k$ is the sample size of $k$-labeled dataset.*

*Proof.* For enhanced clarity, we initially remove the subscript $k$ in $\hat{\boldsymbol{\mu}}_k$, $\boldsymbol{\mu}_k$ and $\boldsymbol{\Sigma}_k$. In order to obtain the solution to Equation (28) and achieve the closed form of the generation distribution, we first examine the discrete solution of Equation (28).

Given any class $k$, we consider the discretization Euler-Maruyama scheme which is widely used in existing work [151–153] and discretize the interval $[0, T]$ into $N$ discretization points,

$$\mathbf{u}_i^\mathsf{T}\mathbf{z}_{(j+1)h} = \mathbf{u}_i^\mathsf{T}\mathbf{z}_{jh} + \left(1 - \frac{1}{\sigma_{t_j}^2 + r^2(t_j)\lambda_i}\right)\mathbf{u}_i^\mathsf{T}\mathbf{z}_jh + \frac{r(t_j)}{\sigma_{t_j}^2 + r^2(t_j)\lambda_i}\mathbf{u}_i^\mathsf{T}\hat{\boldsymbol{\mu}}h, \qquad (30)$$

where $h := \frac{T}{N}$ is the step size and $t_j = jh$.

According to Equation (30), $\mathbf{u}_i^\mathsf{T}\mathbf{z}_{(j+1)h}$ can be regarded as a linear transformation of the initial distribution, which is a standard Gaussian distribution. Therefore, $\mathbf{u}_i^\mathsf{T}\mathbf{z}_{(j+1)h}$ still satisfies a Gaussian distribution. The remaining task is analyze the mean and variance of the $\mathbf{u}_i^\mathsf{T}\mathbf{z}_{(j+1)h}$ separately.

**Expectation.** By Equation (30), we have

$$\mathbb{E}(\mathbf{u}_i^\mathsf{T}\mathbf{z}_{(j+1)h}) = \left(1 + \left(1 - \frac{1}{\sigma_{t_j}^2 + r^2(t_j)\lambda_i}\right)h\right)\mathbb{E}(\mathbf{u}_i^\mathsf{T}\mathbf{z}_{jh}) + \frac{r(t_j)}{\sigma_{t_j}^2 + r^2(t_j)\lambda_i}\mathbf{u}_i^\mathsf{T}\hat{\boldsymbol{\mu}}h.$$

By telescoping, we get the discretization solution

$$\mathbb{E}(\mathbf{u}_i^\mathsf{T}\mathbf{z}_{Nh}) = \prod_{k=0}^{N-1}\left(1 + \left(1 - \frac{1}{\sigma_{t_k}^2 + r^2(t_k)\lambda_i}\right)h\right)\mathbb{E}(\mathbf{u}_i^\mathsf{T}\mathbf{z}_0)$$

$$+ \sum_{k=0}^{N-1}\frac{r(t_k)}{\sigma_{t_k}^2 + r^2(t_k)\lambda_i}\mathbf{u}_i^\mathsf{T}\hat{\boldsymbol{\mu}}h\prod_{j=k+1}^{N-1}\left(1 + \left(1 - \frac{1}{\sigma_{t_j}^2 + r^2(t_j)\lambda_i}\right)h\right).$$

Since the initial distribution of reverse process is $\mathbf{z}_0 \sim \mathcal{N}(\mathbf{0}, \mathbf{I})$, thus $\mathbb{E}(\mathbf{u}_i^\mathsf{T}\mathbf{z}_0) = 0$. We further have

$$\mathbb{E}(\mathbf{u}_i^\mathsf{T}\mathbf{z}_{Nh}) = \sum_{k=0}^{N-1}\frac{r(t_k)}{\sigma_{t_k}^2 + r^2(t_k)\lambda_i}\mathbf{u}_i^\mathsf{T}\hat{\boldsymbol{\mu}}h\prod_{j=k+1}^{N-1}\left(1 + \left(1 - \frac{1}{\sigma_{t_j}^2 + r^2(t_j)\lambda_i}\right)h\right). \qquad (31)$$

By limiting $h \to 0$, we can use the identity $1 + hx \simeq \exp(hx)$. Then we get the continous version of Equation (31) that when $h \to 0$, we have

$$\mathbb{E}(\mathbf{u}_i^\mathsf{T}\mathbf{z}_T) = \int_0^T \frac{r_t}{\sigma_t^2 + r_t^2\lambda_i}\left(\exp\int_t^T 1 - \frac{1}{\sigma_r^2 + r^2(r)\lambda_i}dr\right)\mathbf{u}_i^\mathsf{T}\hat{\boldsymbol{\mu}}dt$$

$$= \int_0^T \frac{e^{t-T}}{1 + e^{2(t-T)}(\lambda_i - 1)}\left(\exp\int_t^T 1 - \frac{1}{1 + e^{2(r-T)}(\lambda_i - 1)}dr\right)\mathbf{u}_i^\mathsf{T}\hat{\boldsymbol{\mu}}dt$$

$$= \int_0^T \frac{e^{t-T}\sqrt{\lambda_i}}{[1 + e^{2(t-T)}(\lambda_i - 1)]^{\frac{3}{2}}}\mathbf{u}_i^\mathsf{T}\hat{\boldsymbol{\mu}}dt$$

$$= \sqrt{\lambda_i}\frac{e^{t-T}}{\sqrt{1 + e^{2(t-T)}(\lambda_i - 1)}}\Bigg|_0^T \mathbf{u}_i^\mathsf{T}\hat{\boldsymbol{\mu}}$$

$$= \left(1 - \frac{\sqrt{\lambda_i}e^{-T}}{\sqrt{1 + (\lambda_i - 1)e^{-2T}}}\right)\mathbf{u}_i^\mathsf{T}\hat{\boldsymbol{\mu}}.$$

Hence, we have

$$\mathbb{E}(\mathbf{z}_T) = \mathbf{U}\mathrm{diag}(e_1, e_2, \ldots, e_d)\mathbf{U}^T\hat{\mu},$$

where $e_i = 1 - \frac{\sqrt{\lambda_i}e^{-T}}{\sqrt{1+(\lambda_i-1)e^{-2T}}}$.

When $T \to \infty$, we get

$$e_i = 1 - \frac{\sqrt{\lambda_i}e^{-T}}{\sqrt{1 + (\lambda_i - 1)e^{-2T}}} \to 1.$$

And the expectation of generation distribution is the empirical mean, i.e.,

$$\mathbb{E}(\mathbf{z}_T) = \hat{\mu}.$$

**Variance.** Similarly, by Equation (30), we have the variance

$$\text{Var}(\mathbf{u}_i^\mathsf{T} \mathbf{z}_{(j+1)h}) = \left(1 + \left(1 - \frac{1}{\sigma_{t_j}^2 + r^2(t_j)\lambda_i}\right)h\right)^2 \text{Var}(\mathbf{u}_i^\mathsf{T} \mathbf{z}_{jh}).$$

By telescoping, we get the discretization solution

$$\text{Var}(\mathbf{u}_i^\mathsf{T} \mathbf{z}_{Nh}) = \prod_{k=0}^{N-1} \left(1 + \left(1 - \frac{1}{\sigma_{t_k}^2 + r^2(t_k)\lambda_i}\right)h\right)^2 \text{Var}(\mathbf{u}_i^\mathsf{T} \mathbf{z}_0).$$

Since $\text{Var}(\mathbf{u}_i^\mathsf{T} \mathbf{z}_0) = \mathbf{u}_i^\mathsf{T} \text{Var}(\mathbf{z}_0)\mathbf{u}_i = 1$, the variance at time $T$ is

$$\text{Var}(\mathbf{u}_i^\mathsf{T} \mathbf{z}_{Nh}) = \prod_{k=0}^{N-1} \left(1 + \left(1 - \frac{1}{\sigma_{t_k}^2 + r^2(t_k)\lambda_i}\right)h\right)^2.$$

When $h \to 0$ and use the identity $(1 + hx)^2 \simeq \exp(2hx)$, we get

$$\text{Var}(\mathbf{u}_i^\mathsf{T} \mathbf{z}_T) = \exp \int_0^T 2\left(1 - \frac{1}{\sigma_t^2 + r_t^2 \lambda_i}\right) dt$$

$$= \exp \int_0^T 2\left(1 - \frac{1}{1 + (\lambda_i - 1)e^{2(t-T)}}\right) dt$$

$$= \frac{\lambda_i}{1 + (\lambda_i - 1)e^{-2T}}.$$

Hence, we have

$$\text{Var}(\mathbf{z}_T) = \mathbf{U} \text{diag}(v_1, v_2, \ldots, v_d) \mathbf{U}^T.$$

where $v_i = \frac{\lambda_i}{1+(\lambda_i-1)e^{-2T}}$.

When $T \to \infty$, we get

$$v_i = \frac{\lambda_i}{1 + (\lambda_i - 1)e^{-2T}} \to \lambda_i.$$

And the expectation of generation distribution is the empirical mean, i.e.,

$$\text{Var}(\mathbf{z}_T) = \mathbf{\Sigma}.$$

We complete the proof. □

### A.3.2 Corrupted Conditions

We then discuss the distribution of generated data when the conditional embedding is perturbed by noise. Using the same method to derive the data distribution under the corrupted conditional embedding setting, we conclude the following Lemma 4.

**Lemma 4.** *(Corrupted Conditional Embedding). For any class $k \in \mathcal{Y}$, the distribution of generation $\mathbf{z}_T^c$ is*

$$\lim_{T \to \infty} \mathbf{z}_T^c \sim \mathcal{N}\left(\frac{\hat{\mu}_k}{1+\gamma^2}, \mathbf{\Sigma}_k + \frac{\gamma^2}{1+\gamma^2}\|\hat{\mu}_k\|_2^2 \mathbf{I}\right), \tag{32}$$

*where $\mathbf{\Sigma}_k := \frac{1}{n_k}\sum_{i=1}^{n_k} \mathbf{x}_i \mathbf{x}_i^\mathsf{T} - \frac{1}{n_k^2}\sum_{i=1}^{n_k} \mathbf{x}_i \sum_{i=1}^{n_k} \mathbf{x}_i^\mathsf{T}$ is the empirical covariance of $k$-labeled dataset and $\hat{\mu}_k := \frac{1}{n_k}\sum_{i=1}^{n_k} \mathbf{x}_i$ is the empirical mean of $k$-labeled dataset, $n_k$ is the sample size of $k$-labeled dataset. And $\gamma \geq 0$ is the corruption control parameter.*

*Proof.* To improve clarity, we initially disregard the subscript $k$ in $\hat{\boldsymbol{\mu}}_k$, $\boldsymbol{\mu}_k$ and $\boldsymbol{\Sigma}_k$. Building upon the proof details presented in Lemma 3 and the optimal conditional score estimation outlined in Lemma 2, we arrive at the subsequent findings:

**Expectation.** Similarly, we derive

$$
\begin{aligned}
\mathbb{E}(\mathbf{u}_i^\mathsf{T}\mathbf{z}_T^c) &= \frac{1}{1+\gamma^2}\int_0^T \frac{r_t}{\sigma_t^2 + r_t^2(\lambda_i + \frac{\gamma^2}{1+\gamma^2}\|\hat{\boldsymbol{\mu}}\|_2^2)}\left(\exp\int_t^T 1 - \frac{1}{\sigma_r^2 + r^2(r)(\lambda_i + \frac{\gamma^2}{1+\gamma^2}\|\hat{\boldsymbol{\mu}}\|_2^2)}dr\right)\mathbf{u}_i^\mathsf{T}\hat{\boldsymbol{\mu}}\,dt \\
&= \frac{1}{1+\gamma^2}\int_0^T \frac{e^{t-T}}{1 + e^{2(t-T)}(\lambda_i - 1 + \frac{\gamma^2}{1+\gamma^2}\|\hat{\boldsymbol{\mu}}\|_2^2))}\left(\exp\int_t^T 1 - \frac{1}{1 + e^{2(r-T)}(\lambda_i - 1 + \frac{\gamma^2}{1+\gamma^2}\|\hat{\boldsymbol{\mu}}\|_2^2))}dr\right)\mathbf{u}_i^\mathsf{T}\hat{\boldsymbol{\mu}}\,dt \\
&= \frac{1}{1+\gamma^2}\int_0^T \frac{e^{t-T}\sqrt{\lambda_i + \frac{\gamma^2}{1+\gamma^2}\|\hat{\boldsymbol{\mu}}\|_2^2})}{[1 + e^{2(t-T)}(\lambda_i - 1 + \frac{\gamma^2}{1+\gamma^2}\|\hat{\boldsymbol{\mu}}\|_2^2)]^{\frac{3}{2}}}\mathbf{u}_i^\mathsf{T}\hat{\boldsymbol{\mu}}\,dt \\
&= \frac{1}{1+\gamma^2}\sqrt{\lambda_i + \frac{\gamma^2}{1+\gamma^2}\|\hat{\boldsymbol{\mu}}\|_2^2}\frac{e^{t-T}}{\sqrt{1 + e^{2(t-T)}(\lambda_i - 1 + \frac{\gamma^2}{1+\gamma^2}\|\hat{\boldsymbol{\mu}}\|_2^2)}}\Bigg|_0^T\mathbf{u}_i^\mathsf{T}\hat{\boldsymbol{\mu}} \\
&= \frac{1}{1+\gamma^2}\left(1 - \frac{\sqrt{\lambda_i + \frac{\gamma^2}{1+\gamma^2}\|\hat{\boldsymbol{\mu}}\|_2^2}e^{-T}}{\sqrt{1 + (\lambda_i - 1 + \frac{\gamma^2}{1+\gamma^2}\|\hat{\boldsymbol{\mu}}\|_2^2)e^{-2T}}}\right)\mathbf{u}_i^\mathsf{T}\hat{\boldsymbol{\mu}}.
\end{aligned}
$$

Hence, we have

$$
\mathbb{E}(\mathbf{z}_T^c) = \mathbf{U}\mathrm{diag}(e_1^c, e_2^c, \ldots, e_d^c)\mathbf{U}^T\hat{\mu},
$$

where $e_i^c = \frac{1}{1+\gamma^2}\left(1 - \frac{\sqrt{\lambda_i + \frac{\gamma^2}{1+\gamma^2}\|\hat{\boldsymbol{\mu}}\|_2^2}e^{-T}}{\sqrt{1+(\lambda_i - 1 + \frac{\gamma^2}{1+\gamma^2}\|\hat{\boldsymbol{\mu}}\|_2^2)e^{-2T}}}\right)$.

When $T \to \infty$, we get

$$
e_i^c = \frac{1}{1+\gamma^2}\left(1 - \frac{\sqrt{\lambda_i + \frac{\gamma^2}{1+\gamma^2}\|\hat{\boldsymbol{\mu}}\|_2^2}e^{-T}}{\sqrt{1 + (\lambda_i - 1 + \frac{\gamma^2}{1+\gamma^2}\|\hat{\boldsymbol{\mu}}\|_2^2)e^{-2T}}}\right) \to \frac{1}{1+\gamma^2}.
$$

And the expectation of generation distribution is

$$
\mathbb{E}(\mathbf{z}_T^c) = \frac{1}{1+\gamma^2}\hat{\mu}
$$

**Variance.** We get

$$
\begin{aligned}
\mathrm{Var}(\mathbf{u}_i^\mathsf{T}\mathbf{z}_T^c) &= \exp\int_0^T 2\left(1 - \frac{1}{\sigma_t^2 + r_t^2(\lambda_i + \frac{\gamma^2}{1+\gamma^2}\|\hat{\boldsymbol{\mu}}\|_2^2}\right)dt \\
&= \exp\int_0^T 2\left(1 - \frac{1}{1 + (\lambda_i - 1 + \frac{\gamma^2}{1+\gamma^2}\|\hat{\boldsymbol{\mu}}\|_2^2)e^{2(t-T)}}\right)dt \\
&= \frac{\lambda_i + \frac{\gamma^2}{1+\gamma^2}\|\hat{\boldsymbol{\mu}}\|_2^2}{1 + (\lambda_i - 1 + \frac{\gamma^2}{1+\gamma^2}\|\hat{\boldsymbol{\mu}}\|_2^2)e^{-2T}}.
\end{aligned}
$$

Hence, we have

$$
\mathrm{Var}(\mathbf{z}_T^c) = \mathbf{U}\mathrm{diag}(v_1^c, v_2^c, \ldots, v_d^c)\mathbf{U}^T,
$$

where $v_i^c = \frac{\lambda_i + \frac{\gamma^2}{1+\gamma^2}\|\hat{\boldsymbol{\mu}}\|_2^2}{1+(\lambda_i - 1 + \frac{\gamma^2}{1+\gamma^2}\|\hat{\boldsymbol{\mu}}\|_2^2)e^{-2T}}$.

When $T \to \infty$, we get

$$
v_i^c = \frac{\lambda_i + \frac{\gamma^2}{1+\gamma^2}\|\hat{\boldsymbol{\mu}}\|_2^2}{1 + (\lambda_i - 1 + \frac{\gamma^2}{1+\gamma^2}\|\hat{\boldsymbol{\mu}}\|_2^2)e^{-2T}} \to \lambda_i + \frac{\gamma^2}{1+\gamma^2}\|\hat{\boldsymbol{\mu}}\|_2^2.
$$

And the expectation of generation distribution is the empirical mean, i.e.,

$$
\mathrm{Var}(\mathbf{z}_T^c) = \boldsymbol{\Sigma} + \frac{\gamma^2}{1+\gamma^2}\|\hat{\boldsymbol{\mu}}\|_2^2\mathbf{I}.
$$

$\square$

## A.4 Diversity and Quality: Clean vs. Corrupted Conditions

### A.4.1 Generation Diversity

Given any class $k$, building on previous work [70], we consider entropy to measure the diversity of generated images. In particular, let $\mathbb{P}$ and $\mathbb{P}^c$ be the probability densities for the generated data using clean and corrupted conditional embeddings respectively, we have the following for $\mathbf{z}_t$ and $\mathbf{z}_t^c$

$$H(\mathbf{z}_T|y=k) := -\int \mathbb{P}(\mathbf{z}|y=k)\log\mathbb{P}(\mathbf{z}|y=k)d\mathbf{z}, \tag{33}$$

$$H(\mathbf{z}_T^c|y=k) := -\int \mathbb{P}^c(\mathbf{z}|y=k)\log\mathbb{P}^c(\mathbf{z}|y=k)d\mathbf{z}. \tag{34}$$

We propose the following theorem to describe the difference between these two conditional differential entropy.

**Theorem 3.** *(Restatement of Theorem 1) For any class $k \in \mathcal{Y}$, assuming the norm of corresponding expectation $\|\boldsymbol{\mu}_k\|_2^2$ is a constant and the empirical covariance of training data is full rank, let $\mathbf{z}_T$ and $\mathbf{z}_T^c$ be the generation featuring clean and corrupted conditions respectively, then it holds that*

$$H(\mathbf{z}_T^c|y=k) - H(\mathbf{z}_T|y=k) = \Theta(\gamma^2 d), \tag{35}$$

*where $\gamma$ is the corruption control parameter and $d$ is the data dimension.*

*Proof.* For the sake of clarity, we begin by omitting the subscript $k$ from $\hat{\boldsymbol{\mu}}_k$, $\boldsymbol{\mu}_k$ and $\boldsymbol{\Sigma}_k$. Given any class $k$, since both the clean generation $\mathbf{z}_T$ and the corrupted generation $\mathbf{z}_T^c$ follow multivariate Gaussian distributions, we can derive the closed-form expression for the difference in their differential entropy by Lemma 3 and Lemma 4 as follows

$$H(\mathbf{z}_T^c|y=k) - H(\mathbf{z}_T|y=k) = \frac{1}{2}\log\left|\boldsymbol{\Sigma} + \frac{\gamma^2}{1+\gamma^2}\|\hat{\boldsymbol{\mu}}\|_2^2\mathbf{I}\right| - \frac{1}{2}\log|\boldsymbol{\Sigma}|$$

$$= \frac{1}{2}\sum_{i=1}^{d}\log\left(1 + \frac{\gamma^2}{(1+\gamma^2)\lambda_i}\|\hat{\boldsymbol{\mu}}\|_2^2\right),$$

where $\boldsymbol{\Sigma} := \frac{1}{n}\sum_{i=1}^{n}\mathbf{x}_i\mathbf{x}_i^\mathsf{T} - \frac{1}{n_k^2}\sum_{i=1}^{n}\mathbf{x}_i\sum_{i=1}^{n}\mathbf{x}_i^\mathsf{T}$ is the empirical covariance of $k$-labeled dataset and $\hat{\boldsymbol{\mu}} := \frac{1}{n}\sum_{i=1}^{n}\mathbf{x}_i$ is the empirical mean of $k$-labeled dataset.

When the noise ratio $\gamma$ is small and $\lambda_i = \omega(\gamma)$,

$$\log\left(1 + \frac{\gamma^2}{(1+\gamma^2)\lambda_i}\|\hat{\boldsymbol{\mu}}\|_2^2\right) = \frac{\gamma^2}{\lambda_i}\|\hat{\boldsymbol{\mu}}\|_2^2 + \mathcal{O}(\gamma^2).$$

Therefore,

$$H(\mathbf{z}_T^c|y=k) - H(\mathbf{z}_T|y=k) = \frac{1}{2}\sum_{i=1}^{d}\log\left(1 + \frac{\gamma^2}{(1+\gamma^2)\lambda_i}\|\hat{\boldsymbol{\mu}}\|_2^2\right) = \Theta(\gamma^2 d).$$

$\square$

### A.4.2 Generation Quality

Before starting proving that slight noise is beneficial to the quality of generation, we first introduce the lemmas required for the proof.

**Lemma 5.** *Given $\mathbf{x}_1, \cdots, \mathbf{x}_n$ independent and all distributed as a Gaussian $\mathcal{N}(\boldsymbol{\mu}, \mathbf{I})$. Then,*

$$\mathbb{E}(\mathrm{Var}(\mathbf{x}_i|\mathbf{x}_1 + \cdots + \mathbf{x}_n = \mathbf{z})) = \frac{n-1}{n}\mathbf{I}. \tag{36}$$

*Proof.* The expectation is

$$\mathbb{E}(\mathbf{x}_i|\mathbf{x}_1 + \cdots + \mathbf{x}_n = \mathbf{z})$$
$$= \mathbb{E}(\mathbf{z} - \mathbf{x}_1 - \cdots - \mathbf{x}_{i-1} - \mathbf{x}_{i+1} - \cdots, \mathbf{x}_n|\mathbf{x}_1 + \cdots + \mathbf{x}_n = \mathbf{z})$$
$$= \mathbf{z} - (n-1)\mathbb{E}(\mathbf{x}_i|\mathbf{x}_1 + \cdots + \mathbf{x}_n = \mathbf{z}).$$

Therefore,

$$\mathbb{E}(\mathbf{x}_i | \mathbf{x}_1 + \cdots + \mathbf{x}_n = \mathbf{z}) = \frac{\mathbf{z}}{n}.$$

By the law of total variance

$$\mathbb{E}(\mathrm{Var}(\mathbf{x}_i | \mathbf{x}_1 + \cdots + \mathbf{x}_n = \mathbf{z})) = \mathrm{Var}(\mathbf{x}_i) - \mathrm{Var}(\mathbb{E}(\mathbf{x}_i | \mathbf{x}_1 + \cdots + \mathbf{x}_n = \mathbf{z}))$$
$$= \frac{n-1}{n} \mathbf{I}.$$

$\square$

We consider the Wasserstein distance to measure the distance between the generation distribution and the true data distribution. A smaller Wasserstein distance implies a closer proximity between the generated data distribution and the true data distribution, thereby indicating better generation quality

Given class $k$, we define 2-Wasserstein distance between the true data distribution $\mathbf{x}|y = k$ and the clean generation $\mathbf{z}_T$ as $d := \mathcal{W}_2\Big( \mathcal{N}(\boldsymbol{\mu}_k, \mathbf{I}), \mathcal{N}(\hat{\boldsymbol{\mu}}_k, \boldsymbol{\Sigma}_k) \Big)$. Similarly, the Wasserstein distance between the true data distribution $\mathbf{x}|y = k$ and the corrupted generation $\mathbf{z}_t^c$ is denoted as $d_c := \mathcal{W}_2\Big( \mathcal{N}(\boldsymbol{\mu}_k, \mathbf{I}), \mathcal{N}(\frac{\hat{\boldsymbol{\mu}}_k}{1+\gamma^2}, \boldsymbol{\Sigma}_k + \frac{\gamma^2}{1+\gamma^2} \|\hat{\boldsymbol{\mu}}_k\|_2^2 \mathbf{I}) \Big)$.

**Theorem 4.** *(Restatement of Theorem 2) For any class $k \in \mathcal{Y}$, assume the norm of corresponding expectation $\|\boldsymbol{\mu}_k\|_2^2$ is a constant, let $\mathbb{P}$, $\mathbb{Q}_{\mathbf{X}}$ and $\mathbb{Q}_{\mathbf{X}}^c$ be the ground truth, clean, and corrupted condition distributions, respectively, where $\mathbf{X}$ represents the collection of training data points. If $\gamma = O(1/\sqrt{\max_k n_k})$, it holds that*

$$\mathbb{E}_{\mathbf{X}}\Big[ \mathcal{W}_2^2(\mathbb{P}, \mathbb{Q}_{\mathbf{X}}) - \mathcal{W}_2^2(\mathbb{P}, \mathbb{Q}_{\mathbf{X}}^c) | y = k \Big] = \Omega\Big( \frac{\gamma^2 d}{n_k} \Big), \tag{37}$$

*where $\mathcal{W}_2^2(\cdot, \cdot)$ denotes the quadratic 2-Wasserstein distance between two distributions, $n_k$ is the sample size of $k$-labeled dataset and $d$ is the data dimension.*

*Proof.* To express more clearly, we first omit the subscript $k$ of $n_k$, $\hat{\boldsymbol{\mu}}_k$, $\boldsymbol{\mu}_k$ and $\boldsymbol{\Sigma}_k$. According to Lemma 3 and Lemma 4, we then can directly derive the closed form of Wasserstein distance between two Gaussian as

- The squared Wasserstein distance between true data distribution and clean generation distribution

$$d^2 = \|\hat{\boldsymbol{\mu}} - \boldsymbol{\mu}\|_2^2 + \mathrm{Tr}(\mathbf{I}) + \mathrm{Tr}(\boldsymbol{\Sigma}) - 2\mathrm{Tr}(\boldsymbol{\Sigma}^{\frac{1}{2}})$$
$$= \|\hat{\boldsymbol{\mu}} - \boldsymbol{\mu}\|_2^2 + d + \sum_{i=1}^{d} \lambda_i - 2\sum_{i=1}^{d} \sqrt{\lambda_i}.$$

- The squared Wasserstein distance between true data distribution and corrupted generation distribution

$$d_c^2 = \Big\| \frac{\hat{\boldsymbol{\mu}}}{1+\gamma^2} - \boldsymbol{\mu} \Big\|_2^2 + \mathrm{Tr}(\mathbf{I}) + \mathrm{Tr}\Big( \boldsymbol{\Sigma} + \frac{\gamma^2}{1+\gamma^2} \|\hat{\boldsymbol{\mu}}\|_2^2 \mathbf{I} \Big) - 2\mathrm{Tr}\Big( (\boldsymbol{\Sigma} + \frac{\gamma^2}{1+\gamma^2} \|\hat{\boldsymbol{\mu}}\|_2^2 \mathbf{I})^{\frac{1}{2}} \Big)$$
$$= \Big\| \frac{\hat{\boldsymbol{\mu}}}{1+\gamma^2} - \boldsymbol{\mu} \Big\|_2^2 + d + \sum_{i=1}^{d} \Big( \lambda_i + \frac{\gamma^2}{1+\gamma^2} \|\hat{\boldsymbol{\mu}}\|_2^2 \Big) - 2\sum_{i=1}^{d} \sqrt{\lambda_i + \frac{\gamma^2}{1+\gamma^2} \|\hat{\boldsymbol{\mu}}\|_2^2}.$$

The difference in squared Wasserstein distance is

$$d^2 - d_c^2 = -\Big\| \frac{\hat{\boldsymbol{\mu}}}{1+\gamma^2} - \boldsymbol{\mu} \Big\|_2^2 + \|\hat{\boldsymbol{\mu}} - \boldsymbol{\mu}\|_2^2 - \sum_{i=1}^{d} \frac{\gamma^2}{1+\gamma^2} \|\hat{\boldsymbol{\mu}}\|_2^2 + 2\sum_{i=1}^{d} \Big( \sqrt{\lambda_i + \frac{\gamma^2}{1+\gamma^2} \|\hat{\boldsymbol{\mu}}\|_2^2} - \sqrt{\lambda_i} \Big).$$

We consider the expectation error to reduce the randomness of $\hat{\boldsymbol{\mu}}$ as

$$\mathbb{E}[d^2 - d_c^2] = \underbrace{-\mathbb{E}\Big[ \Big\| \frac{\hat{\boldsymbol{\mu}}}{1+\gamma^2} - \boldsymbol{\mu} \Big\|_2^2 \Big] + \mathbb{E}\Big[ \|\hat{\boldsymbol{\mu}} - \boldsymbol{\mu}\|_2^2 \Big]}_{\text{errors caused by the mean}} + \underbrace{\sum_{i=1}^{d} \mathbb{E}\Big[ 2\Big( \sqrt{\lambda_i + \frac{\gamma^2}{1+\gamma^2} \|\hat{\boldsymbol{\mu}}\|_2^2} - \sqrt{\lambda_i} \Big) - \frac{\gamma^2}{1+\gamma^2} \|\hat{\boldsymbol{\mu}}\|_2^2 \Big]}_{\text{errors caused by the variance}}.$$

We notice that the expectation error can be decomposed into two parts. The error of the first part being caused by the difference in means between the generated distribution and the true distribution. Since the distribution of empirical mean is $\hat{\boldsymbol{\mu}} \sim \mathcal{N}(\boldsymbol{\mu}, \frac{1}{n}\mathbf{I})$ where $n_k$ is the sample size of $k$-labeled dataset, we get

$$-\mathbb{E}\left[\left\|\frac{\hat{\boldsymbol{\mu}}}{1+\gamma^2} - \boldsymbol{\mu}\right\|_2^2\right] + \mathbb{E}\left[\|\hat{\boldsymbol{\mu}} - \boldsymbol{\mu}\|_2^2\right] = -\frac{d}{n(1+\gamma^2)^2} - (\frac{\gamma^2}{1+\gamma^2})^2\|\boldsymbol{\mu}\|_2^2 + \frac{d}{n}$$

$$\geq -\frac{d}{n} + \frac{2d\gamma^2}{n} - o(\gamma^4) + \frac{d}{n}$$

$$= \frac{2d\gamma^2}{n} - o(\gamma^4).$$

The second part is attributable to the difference in covariance.

$$\sum_{i=1}^{d} \mathbb{E}\left[2\Big(\sqrt{\lambda_i + \frac{\gamma^2}{1+\gamma^2}\|\hat{\boldsymbol{\mu}}\|_2^2} - \sqrt{\lambda_i}\Big) - \frac{\gamma^2}{1+\gamma^2}\|\hat{\boldsymbol{\mu}}\|_2^2\right]$$

$$\geq \sum_{i=1}^{d} \mathbb{E}\left[\gamma^2 \frac{\|\hat{\boldsymbol{\mu}}\|_2^2}{\sqrt{\lambda_i}} - \gamma^2\|\hat{\boldsymbol{\mu}}\|_2^2 - o(\gamma^4)\right]$$

$$= \gamma^2 \sum_{i=1}^{d} \mathbb{E}\left[(\frac{1}{\sqrt{\lambda_i}} - 1)\|\hat{\boldsymbol{\mu}}\|_2^2\right] - o(\gamma^4 d).$$

By the law of total expectation

$$\sum_{i=1}^{d} \mathbb{E}\left[(\frac{1}{\sqrt{\lambda_i}} - 1)\|\hat{\boldsymbol{\mu}}\|_2^2\right] = \sum_{i=1}^{d} \mathbb{E}\left[\|\hat{\boldsymbol{\mu}}\|_2^2 \mathbb{E}\Big(\frac{1}{\sqrt{\lambda_i}}\Big|\hat{\boldsymbol{\mu}}\Big)\right] - d$$

$$\overset{(a)}{\geq} \mathbb{E}\left[\sum_{i=1}^{d} \frac{\|\hat{\boldsymbol{\mu}}\|_2^2}{\sqrt{\mathbb{E}(\lambda_i|\hat{\boldsymbol{\mu}})}}\right] - d$$

$$\overset{(b)}{\geq} \mathbb{E}\left[\frac{\|\hat{\boldsymbol{\mu}}\|_2^2}{\sqrt{\sum_{i=1}^{d} \mathbb{E}(\lambda_i|\hat{\boldsymbol{\mu}})}}\right] - d.$$

(a) and (b) achieve by the Jensen's inequality given $\frac{1}{\sqrt{\lambda_i}}$ is a convex function

By Lemma 5, we derive

$$\mathrm{Tr}\Big(\mathbb{E}\Big[\mathrm{Var}(\mathbf{x}|\hat{\boldsymbol{\mu}})\Big]\Big) = \mathrm{Tr}\Big(\mathbb{E}\Big[\boldsymbol{\Sigma}|\hat{\boldsymbol{\mu}}\Big]\Big) = \mathrm{Tr}\Big(\mathbf{U}\mathbb{E}\Big[\boldsymbol{\Lambda}|\hat{\boldsymbol{\mu}}\Big]\mathbf{U}^{\mathsf{T}}\Big) = \sum_{i=1}^{d} \mathbb{E}(\lambda_i|\hat{\boldsymbol{\mu}}) = \frac{n-1}{n}d.$$

Hence, we get

$$\sum_{i=1}^{d} \mathbb{E}\left[(\frac{1}{\sqrt{\lambda_i}} - 1)\|\hat{\boldsymbol{\mu}}\|_2^2\right] \geq \mathbb{E}\left[\frac{\|\hat{\boldsymbol{\mu}}\|_2^2}{\sqrt{\sum_{i=1}^{d} \mathbb{E}(\lambda_i|\hat{\boldsymbol{\mu}})}}\right] - d$$

$$= \Big(\sqrt{\frac{n}{n-1}} - 1\Big)d\mathbb{E}[\|\hat{\boldsymbol{\mu}}\|_2^2]$$

$$= \Big(\sqrt{\frac{n}{n-1}} - 1\Big)\Big(\frac{d^2}{n} + d\|\boldsymbol{\mu}\|_2^2\Big).$$

The second part is

$$\sum_{i=1}^{d} \mathbb{E}\left[2\Big(\sqrt{\lambda_i + \frac{\gamma^2}{1+\gamma^2}\|\hat{\boldsymbol{\mu}}\|_2^2} - \sqrt{\lambda_i}\Big) - \frac{\gamma^2}{1+\gamma^2}\|\hat{\boldsymbol{\mu}}\|_2^2\right]$$

$$\geq \gamma^2\Big(\sqrt{\frac{n}{n-1}} - 1\Big)(\frac{d^2}{n} + d\|\boldsymbol{\mu}\|_2^2) - o(\gamma^4 d).$$

Therefore, with small noise ratio $\gamma$, we then can conclude that

$$\mathbb{E}[d^2 - d_c^2] = -\mathbb{E}\left[\left\|\frac{\hat{\boldsymbol{\mu}}}{1+\gamma^2} - \boldsymbol{\mu}\right\|_2^2\right] + \mathbb{E}\left[\|\hat{\boldsymbol{\mu}} - \boldsymbol{\mu}\|_2^2\right] + \sum_{i=1}^{d}\mathbb{E}\left[2\Big(\sqrt{\lambda_i + \frac{\gamma^2}{1+\gamma^2}\|\hat{\boldsymbol{\mu}}\|_2^2} - \sqrt{\lambda_i}\Big) - \frac{\gamma^2}{1+\gamma^2}\|\hat{\boldsymbol{\mu}}\|_2^2\right]$$

$$\geq \frac{2d\gamma^2}{n} + \gamma^2\Big(\sqrt{\frac{n}{n-1}} - 1\Big)\Big(\frac{d^2}{n} + d\|\boldsymbol{\mu}\|_2^2\Big) - o(\gamma^4 d).$$

Noting that $\sqrt{n/(n-1)} - 1 = -\Theta(1/n)$ when $n$ is large, then if the corruption level $\gamma$ satisfies $\gamma = O(1/\sqrt{n})$, for any class $k$, we have

$$\mathbb{E}[d^2 - d_c^2] = \Omega\Big(\frac{\gamma^2 d}{n}\Big).$$

This completes the proof. $\qquad\square$

# B  Details of Condition Corruption, Model Training and Evaluation

In this section, we provide detailed training setup of each diffusion models we studied in the main paper, the synthetic corruption for IN-1K and CC3M, the annotation process of IN-100, and the evaluation metrics we adopted.

## B.1  Synthetic Condition Corruption

We mainly studied four types of condition corruption in this paper, with two datasets. For IN-1K, we used random symmetric and asymmetric condition corruption. For CC3M, we adopted text swapping and LLM re-writing corruption. We used several levels of corruption $\eta = \{0, 2.5, 7.5, 10, 15, 20\}\%$.

**Symmetric Condition Corruption for IN-1K**. To introduce symmetric condition corruption in IN-1K according to a corruption ratio $\eta$, we randomly sample a $(\mathbf{x}, y)$ pair from the dataset, and flip $y$ to another class according to the class prior in IN-1K to obtain $y^c$, until the ratio of $y^c$ satisfies $\eta$.

**Asymmetric Condition Corruption for IN-1K**. For asymmetric condition corruption of IN-1K, we first find the class overlap between IN-1K and CIFAR-100 using WordNet [154], denoted as $\mathcal{Y}_{\text{IN}-1\text{K}}^{\text{C}-100}$. Then, we randomly sample $(\mathbf{x}, y)$ from the data subset whose $y$ satisfies $y \in \mathcal{Y}_{\text{IN}-1\text{K}}^{\text{C}-100}$ and flip $y$ into the remaining classes of the overlapped set $\mathcal{Y}_{\text{IN}-1\text{K}}^{\text{C}-100}/y$.

**Text Swapping Condition Corruption for CC3M**. For CC3M, where $y$ is text captions for the images, we randomly sample two pairs and swap the text of these two pairs to introduce condition corruptions. This mainly follows Chen et al. [48], where very disruptive corruption is introduced.

**LLM Text Condition Corruption for CC3M**. Text swapping corruption may not be common in practice for image-text datasets. Instead, we may encounter captions that have unmatched entities or partially unmatched sentences with the images. To study the text corruption in a more realistic scenarios, we use GPT-4 and prompt it to re-write the captions to introduce corruptions. We pre-define 5 levels of corruption in the prompt, and randomly sample a level as input to GPT-4.

## B.2  Automatic ImageNet-100 Annotation

Here, we present the details of annotate ImageNet-100 for personalization of LDMs using ControlNet and T2I-Adapters. A few examples of the annotated images and captions are shown in Fig. 10.

**Canny Edge**. For canny edge, we directly use the Canny detector from OpenCV to annotate the images. We set the low threshold and high threshold of canny detector to 100 and 200 respectively.

**Segmentation Mask from SAM**. We use SAM to annotate segmentation masks from IN-100 images. We directly use the colormap of the segmentation masks as input control to ControlNet and T2I-Adapters.

**Captions from BLIP**. We use BLIP captioning model to generate captions for IN-100 for adapting text-conditional LDMs.

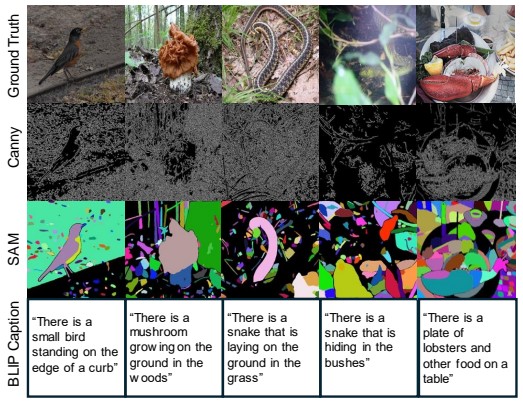

Figure 10: Annotation examples of IN-100.

## B.3 LDM Pre-training Setup

The pre-training setup of LDM-4 mainly follows Rombach et al. [9], as shown in Table 3. For LDM-4 models, we use a VQ-VAE [56] with a down-sampling factor of $4$ and a latent space with shape $64 \times 64 \times 3$. It also has a vocabulary size of 8196.

Table 3: Hyper-parameters of IN-1K class-conditional and CC3M text-conditional LDMs.

|  | IN-1K $256 \times 256$ | CC3M $256 \times 256$ |
|---|---|---|
| Down-sampling Factor | 4 | 4 |
| Latent Shape | $64 \times 64 \times 3$ | $64 \times 64 \times 3$ |
| Vocabulary Size | 8192 | 8192 |
| Diffusion Steps | 1000 | 1000 |
| Noise Schedule | Linear | Linear |
| U-Net Param. Size | 400M | 400M |
| Condition Net | Class Embedder | BERT |
| Channels | 192 | 192 |
| Channel Multipler | 1,2,3,5 | 1,2,3,5 |
| Number of Heads | 1 | 1 |
| Batch Size | 64 | 64 |
| Training Iter. | 178K | 396K |
| Learning Rate | 1$e$-4 | 1$e$-4 |

**IN-1K**. The hyper-parameters of training IN-1K class-conditional LDMs are summarized as follows. We use a U-Net with channels of 192 and channel multipliers of $1, 2, 3, 5$ as the denoising network backbone. We use class embedding, i.e., embedding layer, for computing the embeddings of class labels. The conditional embedding is injected to the U-Net with cross-attention. We use DDPM with linear schedule of 1000 steps. The batch size is set to 64 per GPU, and the learning rate is set to $1e$-4 Training IN-1K LDMs for 178K iterations takes about 2.5 days on 8 NVIDIA A100.

**CC3M**. We use a same U-Net as denoising network backbone. We adjust the training iterations to 396K iterations for CC3M, which takes 7.5 days to train on 8 NVIDIA A100. We use a pre-trained BERT model (bert-base-uncased) for the conditional embeddings, and it is fully trainable.

## B.4 DiT Pre-training Setup

We pre-train DiT-XL/2 on IN-1K follows Peebles et al. [11]. The hyper-parameters are shown in Table 4. We train DiT-XL/2 for 400K training iterations using a per GPU batch size of 32 on 8 NVIDIA A100, which takes around 2.5 days. Compared to LDM-4, DiT-XL/2 used a fine-tuned VQ-VAE with a down-sampling factor of $8$, a latent space of shape $32 \times 32 \times 4$, and a vocabulary size of $16384$. DiT-XL/2 has a denoising network backbone based on Transformer architecture and uses Adaptive LayerNorm, initialized with zeros, for injecting the conditional information.

Table 4: Hyper-parameters of IN-1K class-conditional DiT-XL/2.

| | DiT-XL/2 IN-1K $256 \times 256$ |
|---|---|
| Down-sampling Factor | 8 |
| Latent Shape | $32 \times 32 \times 4$ |
| Vocabulary Size | 16384 |
| Params. | 675M |
| Training Iters. | 400K |
| Batch Size | 32 |
| Learning Rate | 1e-4 |

## B.5 LCM Pre-training Setup

LCM distills the pre-trained Stable Diffusion models to enable faster inference with fewer steps. We choose Stable Diffusion v1.5 as the teacher model and conduct distillation for 35K iterations, which takes 1.5 days on 8 NVIDIA A100 GPUs. We use a learning rate of $1e$-5.

## B.6 ControlNet and T2I-Adapter Personalization Setup

We use the implementation of ControlNet and T2I-adapter of Diffusers [155] for downstream personalization tasks. Default learning rate and batch size from Diffusers are used for these two methods, and we set the training epochs for IN-100 as 10. On 4 NVIDIA V100 GPUs, training ControlNet and T2I-Adapter with LDM-4 takes about 6 hours.

## B.7 Evaluation Metrics

We introduce the details of metrics we used to evaluate the diffusion models here. Due to the known difficulties of evaluating generative models, we adopt most of the existing criteria to evaluate the models we have trained.

**Fréchet Inception Distance (FID)** [63]. FID measures the distance between real and generated images in the feature space of an ImageNet-1K pre-trained classifier [156], indicating the similarity and fidelity of the generated images to real images.

**sFID** [71]. sFID utilizes the mid-level features of the inception network [156], which are more sensitive to spatial variability.

**Inception Score (IS)** [64]. IS also measures the fidelity and diversity of generated images. It consists of two parts: the first part measures whether each image belongs confidently to a single class of an ImageNet-1K pre-trained image classifier [156] and the second part measures how well the generated images capture diverse classes.

**Precision and Recall** [65]. The real and generated images are first converted to non-parametric representations of the manifolds using k-nearest neighbors, on which the Precision and Recall can be computed. Precision is the probability that a random generated image from estimated generated data manifolds falls within the support of the manifolds of estimated real data distribution. Recall is the probability that a random real image falls within the support of generated data manifolds. Thus, precision measures the general quality and fidelity of the generated images, and the recall measures the coverage and diversity of the generated images.

**Top-1% Relative Mahalanobis Distance (RMD) Score** [67]. RMD score measures the sample complexity and difficulty. It is defined as the difference between the Mahalanobis distances of a sample induced by the class-specific and class-agnostic Gaussian distributed estimated from the generated data. Given the dataset $\{(\mathbf{x}_i, y_i)\}_{i \in [N]}$, we first compute the features using the CLIP ViT-B-16 encoder from the images as $G(\mathbf{x})$. The class-specific Gaussian distribution is then estimated:

$$\mathbb{P}(G(\mathbf{x}) \mid y = k) = \mathcal{N}(G(\mathbf{x}) \mid \boldsymbol{\mu}_k, \boldsymbol{\Sigma})$$
$$\boldsymbol{\mu}_k = \frac{1}{N_k} \sum_{i:y_i=k} G(\mathbf{x}_i)$$
$$\boldsymbol{\Sigma} = \frac{1}{N} \sum_k \sum_{i:y_i=k} (G(\mathbf{x}_i) - \boldsymbol{\mu}_k)(G(\mathbf{x}_i) - \boldsymbol{\mu}_k)^\top .$$

(38)

The class-agnostic Gaussian distribution is estimated over all data as;

$$\mathbb{P}(G(\mathbf{x})) = \mathcal{N}\left(G(\mathbf{x}) \mid \boldsymbol{\mu}_{\mathrm{agn}}, \boldsymbol{\Sigma}_{\mathrm{agn}}\right),$$

$$\boldsymbol{\mu}_{\mathrm{agn}} = \frac{1}{N} \sum_{i}^{N} G\left(\mathbf{x}_i\right),$$

$$\boldsymbol{\Sigma}_{\mathrm{agn}} = \frac{1}{N} \sum_{i}^{N} \left(G\left(\mathbf{x}_i\right) - \boldsymbol{\mu}_{\mathrm{agn}}\right)\left(G\left(\mathbf{x}_i\right) - \boldsymbol{\mu}_{\mathrm{agn}}\right)^{\top}. \tag{39}$$

The RMD is defined as:

$$\mathcal{RMD}\left(\mathbf{x}_i, y_i\right) = \mathcal{M}\left(\mathbf{x}_i, y_i\right) - \mathcal{M}_{\mathrm{agn}}\left(x_i\right)$$

$$\mathcal{M}\left(\mathbf{x}_i, y_i\right) = -\left(G\left(\mathbf{x}_i\right) - \boldsymbol{\mu}_{y_i}\right)^{\top} \boldsymbol{\Sigma}^{-1}\left(G\left(\mathbf{x}_i\right) - \boldsymbol{\mu}_{y_i}\right) \tag{40}$$

$$\mathcal{M}_{\mathrm{agn}}\left(\mathbf{x}_i\right) = -\left(G\left(\mathbf{x}_i\right) - \boldsymbol{\mu}_{\mathrm{agn}}\right)^{\top} \boldsymbol{\Sigma}_{\mathrm{agn}}^{-1}\left(G\left(\mathbf{x}_i\right) - \boldsymbol{\mu}_{\mathrm{agn}}\right)$$

We compute the RMD score for all generated images, and report only the top-1% of them.

**Average Top-5 $L_2$ Distances**. As an additional metric of sample diversity, we compute the $L_2$ distance of each generated image with the top-5 nearest neighbor training images. To reduce computation requirement of searching over the raw pixel space, we use the CLIP ViT-B-16 image encoder [157] to transform images into the feature space before calculating the $L_2$ distance. This metric measures the distance of generated samples with training images, as a proxy evaluation of diversity and memorization.

**TopPR F1** [72]. TopPR is a set of reliable evaluation metrics with statistically consistent estimates of generated data and real data. We use the F1 score, computed from the TopPR Precision and Recall as an additional metric to evaluate the general quality and diversity of generated images.

**CLIP Score** [66]. CLIP score measures the cosine similarity between the CLIP embedding of an image-text pair. It is widely used as a metric to evaluate the fidelity and alignment of the generated images and the conditional text prompts [9].

**Memorization Ratio** [73]. We compute the memorization ratio as the percentage of generated images whose $L_2$ distances with their nearest neighbor training images are below a pre-defined threshold. We compute the distances in the feature space of CLIP ViT-B-16 image encoder [157] due to the massive size and resolution of the training images and set the threshold as $0.12$. Although there are several studies using the distance comparison between the first and second nearest neighbor as a reflection of memorization [158, 159], we found that this metric is not effective for large-scale datasets.

**Entropy**. We compute the entropy metric within the latent space of the pre-trained VQ-VAE [58, 56]. Since LDMs (and DiT) learn the data distribution from the latent space of VQ-VAE, we can compute the sample entropy $\mathbb{E}[H(\mathbf{x})]$ using the generated and flatten latent vector $\mathbf{x} \in \mathbb{R}^{HW \times D}$ and the codebook $\mathcal{C} \in \mathbb{R}^{C \times D}$ of VQ-VAE, where $H$ and $W$ are the height and weights of the original latent vectors, $D$ indicates the dimension of the latent space, and $C$ denotes the number of embeddings of the codebook. We compute the probability of each latent vector as $\mathrm{Softmax}\left(\|\mathbf{x} - \mathcal{C}\|_2^2 / \tau\right)$, where $\tau$ is a temperature parameter controlling the sharpness of the probability. We compute entropy as:

$$\mathbb{E}[H(\mathbf{x})] = \frac{1}{NHW} \sum_{i}^{N} \sum_{j}^{HW} \sum_{k}^{C} \mathrm{Softmax}\left(\|\mathbf{x}_{(i,j)} - \mathcal{C}\|_2^2 / \tau\right) \tag{41}$$

## C   Full Results of Pre-training Evaluation

In this section, we present all results of our pre-training evaluation, over different diffusion models, including LDM-4, DiT-XL/2, and LCM-v1.5, and various types of condition corruption.

### C.1   Quantitative Results

We present the full evaluation results of IN-1K class-conditional LDMs and CC3M text-conditional LDMs in Fig. 11 and Fig. 12 respectively. All the results are computed from using a set of guidance scales. For IN-1K LDMs, we use $s \in$

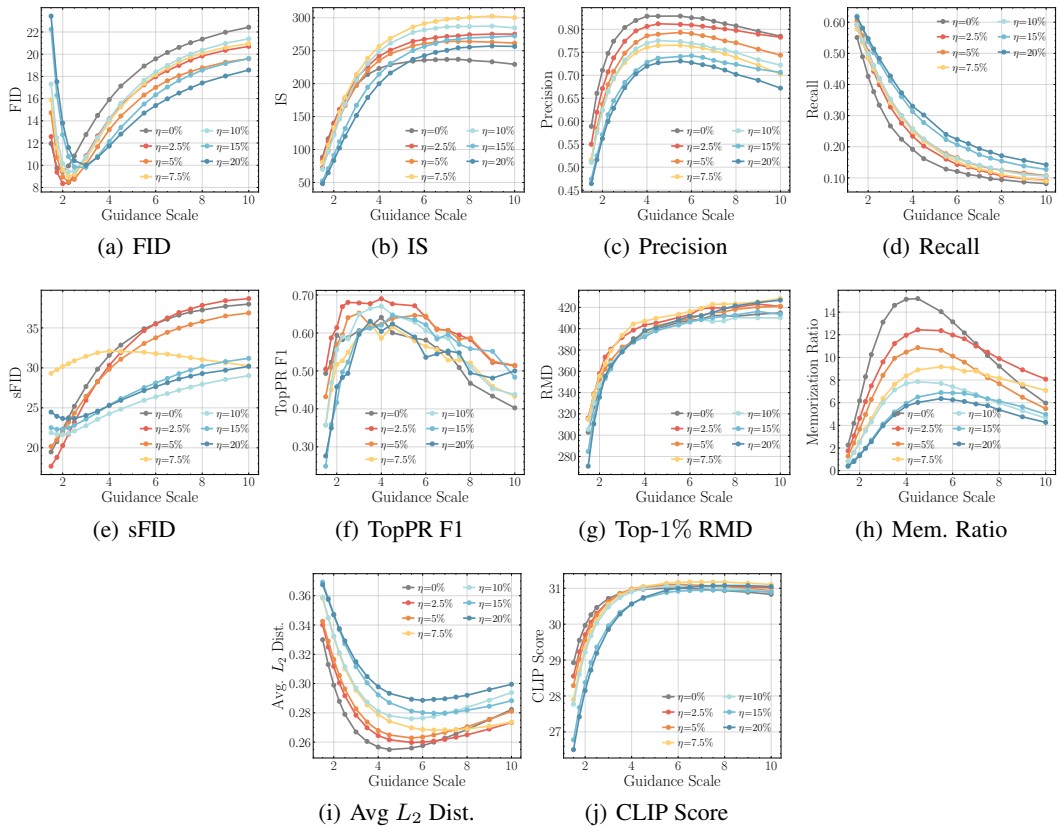

Figure 11: Qualitative evaluation results of 50K images generated by class-conditional LDMs pre-trained on ImageNet-1K with synthetic corruptions. The images are generated with various guidance scales using 1K class conditions and compared with 50K validation images of ImageNet-1K.

$\{1.5, 1.75, 2.0, 2.25, 2.5, 3.0, 3.5, 4.0, 4.5, 5.5, 6.0, 6.5, 7.0, 7.5, 8.0, 9.0, 10.0\}$. For CC3M LDMs, we use $s \in \{1.5, 2.0, 2.5, 2.75, 3.0, 3.25, 3.5, 4.0, 5.0, 6.0, 7.0, 7.5, 8.0, 10.0\}$ For all the metrics, including FID, IS, Precision, Recall, sFID, TopPR F1, and CLIP score, we can all observe that slight condition corruption makes LDMs perform better, with improved image quality and diversity. We also observe that, when there is condition corruption in the dataset, the memorization ratio based on $L_2$ distances actually decreases, in line with observations as in Gu et al. [158].

By default we use DPM scheduler for generating the images with 50 inference steps. But we also study the generation of DDIM scheduler with 250 inference steps, as adopted in [9]. Due to the computation cost of running DDIM scheduler for 250 steps, we only study it with IN-1K LDM-4. The results are shown in Fig. 13. One can observe the same trends from the metrics using DPM and DDIM, demonstrating our findings are scheduler agnostic.

We then show the pre-training results of DiT-XL/2 and LCM-v1.5 in Fig. 19, where we primarily compute the FID, IS, Precision, and Recall. For DiT-XL/2, we use $s \in \{1.5, 1.75, 2.0, 2.25, 2.5, 3.0, 3.5, 4.0, 4.5, 5.0, 5.5\}$. For LCM-v1.5, we use $s \in \{1.5, 2.0, 2.5, 3.0, 3.5, 4.0, 4.5, 5.0, 5.5, 6.0, 6.5, 7.0, 7.5, 8.0, 9.0, 10.0\}$. Slight condition corruption also facilitates the performance by using the most suitable guidance scale.

We additionally include the FID and IS trend along training for LDM IN-1K model with no corruption and 2.5% corruption, using a guidance scale of 2.5, as shown in Table 5. Slight corruption begins to be effective at the very early stage of training.

Finally, we present the results of LDMs pre-trained on CC3M with LLM corruption and IN-1K with asymmetric corruption, where similar observations still hold.

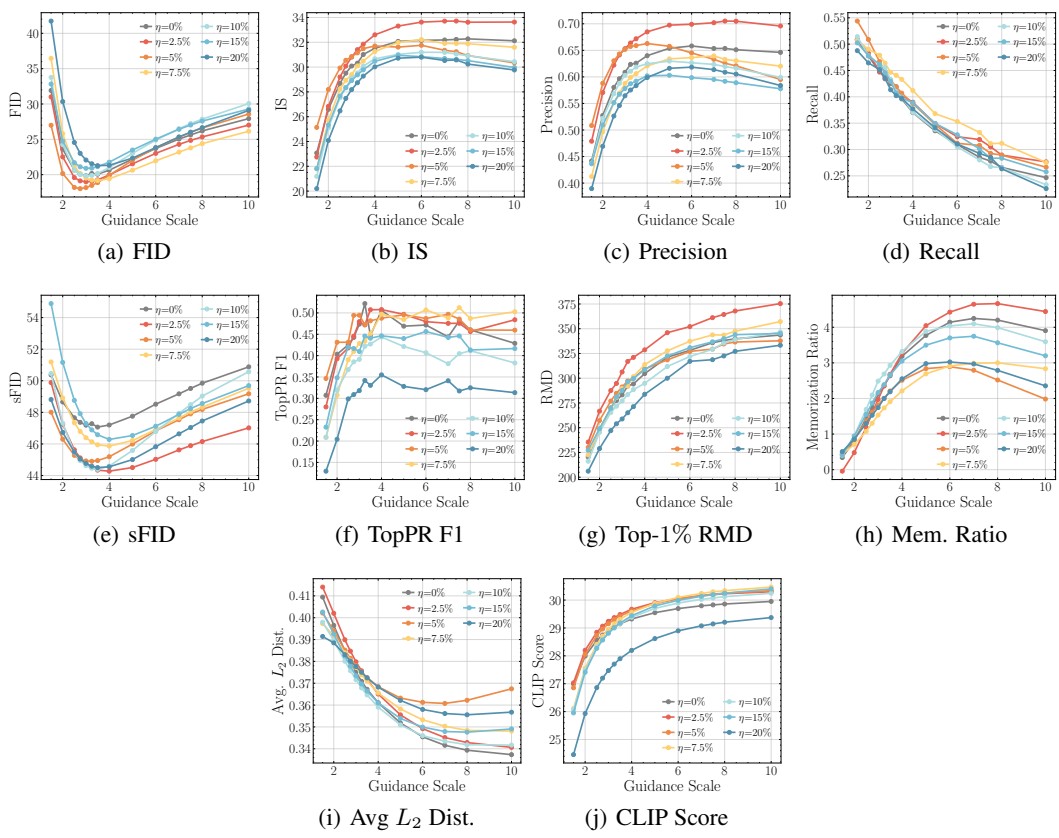

Figure 12: Qualitative evaluation results of 50K images generated by class-conditional LDMs pre-trained on CC3M with synthetic corruptions. The images are generated with various guidance scales using 5K text conditions from MS-COCO and compared with validation images of MS-COCO.

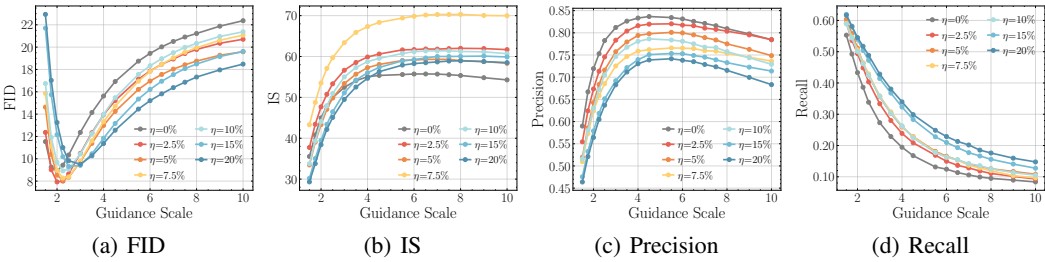

Figure 13: Qualitative evaluation results of 50K images generated by class-conditional LDMs pre-trained on ImageNet-1K with synthetic corruptions. The images are generated with various guidance scales using 1K class conditions and compared with 50K validation images of ImageNet-1K. We use DDIM scheduler with 250 inference steps for these results.

## C.2 Qualitative Results

We present more visualization results of class-conditional LDM-4 in Fig. 18, class-conditional DiT-XL/2 in Fig. 19, text-conditional LDM-4 in Fig. 20, and text-conditional LCM-v1.5 in Fig. 21. One can observe that DMs pre-trained with slight condition corruption in general more visually appealing images.

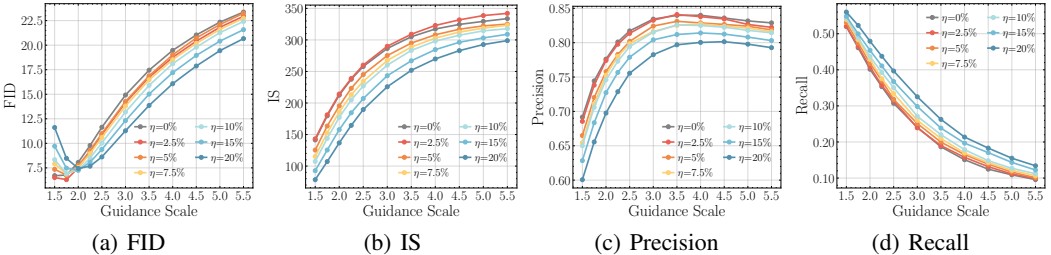

| (a) FID | (b) IS | (c) Precision | (d) Recall |

Figure 14: Qualitative evaluation results of 50K images generated by class-conditional DiT-XL/2 pre-trained on IN-1K with synthetic corruptions. The images are generated with various guidance scales using 1K class conditions from IN-1K and compared with validation images of IN-1K.

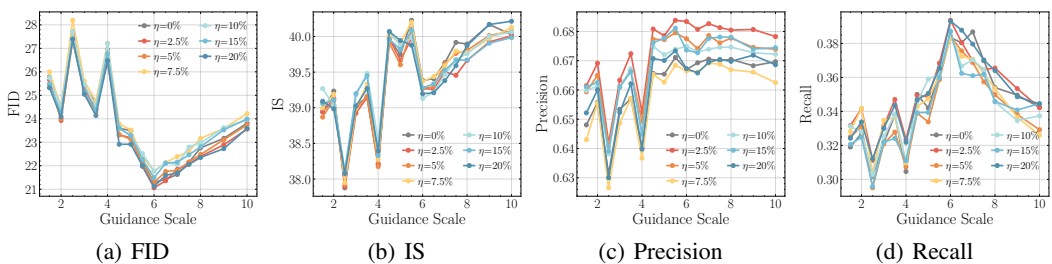

| (a) FID | (b) IS | (c) Precision | (d) Recall |

Figure 15: Qualitative evaluation results of 50K images generated by text-conditional LCM-v1.5 pre-trained on CC3M with synthetic corruptions. The images are generated with various guidance scales using 5K text conditions from MS-COCO and compared with validation images of MS-COCO.

## D  Full Results of Downstream Personalization Evaluation

We present complete results of downstream personalization here.

### D.1  Quantitative Results

We show the results of ControlNet IN-1K LDM-4 in Fig. 22, T2I-Adapter IN-1K LDM-4 in Fig. 23, ControlNet CC3M LDM-4 in Fig. 24, and T2I-Adapter CC3M LDM-4 in Fig. 25. For all personalization experiments, we compute the results for both Canny and SAM spatial controls. Guidance scales of $\{1.25, 1.5, 2.0, 2.25, 2.5, 3.0, 4.0, 5.0, 6.0, 7.0\}$ and $\{2.0, 3.0, 4.0, 5.0, 6.0, 6.5, 7.0, 7.5, 8.0, 8.5, 9.0, 10.0\}$ are used for IN-1K models and CC3M models, respectively, for all experiments here.

From the results, one can observe that models pre-trained with slight condition corruption also present the best performance in downstream personalization tasks.

### D.2  Qualitative Results

We present the qualitative comparison of ControlNet personalization results here. Since T2I-Adapter personalization results are similar but visually worse (quantitatively worse too), we skip their results. The visualizations of ControlNet IN-1K LDM-4 with Canny and SAM conditions are shown in Fig. 26 and Fig. 27, respectively. The visualizations of ControlNet CC3M LDM-4 with Canny and SAM conditions are shown in Fig. 28 and Fig. 29, respectively. Similarly, models pre-trained with slight condition corruption present the best image quality.

Table 5: FID and IS along training of LDM IN-1K with guidance scale 2.5.

| $\eta$ | 10K | 25K | 50K | 75K | 100K | 125K | 150K |
|---|---|---|---|---|---|---|---|
| | | | | FID | | | |
| 0 | 71.48 | 52.02 | 20.88 | 14.49 | 12.66 | 10.44 | 10.12 |
| 2.5 | 77.94 | 51.59 | 21.16 | 13.08 | 12.24 | 9.25 | 8.98 |
| | | | | IS | | | |
| 0 | 4.86 | 23.49 | 71.26 | 93.85 | 103.27 | 164.41 | 170.2 |
| 2.5 | 13.66 | 24.40 | 64.27 | 97.11 | 109.39 | 167.21 | 175.83 |

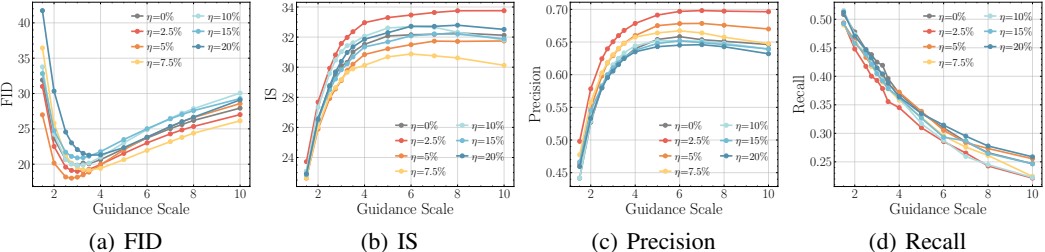

|  (a) FID  |  (b) IS  |  (c) Precision  |  (d) Recall  |
|---|---|---|---|

Figure 16: Qualitative evaluation results of 50K images generated by text-conditional LDMs pre-trained on CC3M with LLM re-writing corruptions. The images are generated with various guidance scales using 5K text conditions from MS-COCO and compared with validation images of MS-COCO.

# E Full Results of Conditional Embedding Perturbation

## E.1 Qualitative Results

More visualizations of CEP, compared with clean and IP pre-trained are shown here. We present the more results of IN-1K LDM-4 and DiT-XL/2 in Fig. 30(a) and Fig. 30(b), respectively. We also present more results of CC3M LDM-4 and LCM-v1.5 in Fig. 31(a) and Fig. 31(b), respectively. CEP generally helps DMs generate more visually appealing and realistic images. We also show the more personalization visualization in Fig. 32 and Fig. 33.

## E.2 Ablation Study

In Fig. 8(a), we compute the $L_2$ distance of perturbed condition embeddings and the clean ones, as a measurement for the corruption levels (of CEP). Here, we elaborate how we compute the $L_2$ distances. For fixed corruption, we calculate the distances as:

$$\frac{1}{N}\sum_{i=1}^{N}\|\mathbf{c}_{\theta^*}(y_i^c) - \mathbf{c}_{\theta^*}(y_i)\|_2^2, \tag{42}$$

where $\theta^*$ is learned from clean data. For CEP, we directly calculate the $L_2$ norm of sampled noise:

$$\sum_{i=1}^{N}\|\boldsymbol{\sigma}_i\|_2^2 \tag{43}$$

## E.3 Comparison with Dropout and Label Smoothing

Here, we additionally compare CEP with dropout and label smoothing on LDM IN-1K models, which are two alternatives that also introduce perturbations in class embeddings. The results are shown in Table 6. One can observe that, both dropout and label smoothing have similar regularization effects on training diffusion models, whereas CEP-U and CEP-G is more effective.

## E.4 Comparison with Fixed and Random Corruption

We further compare with fixed CEP corruption, and random data corruption, to study the effects of fixed and random perturbation to train diffusion models. For fixed CEP-U, we first select the samples

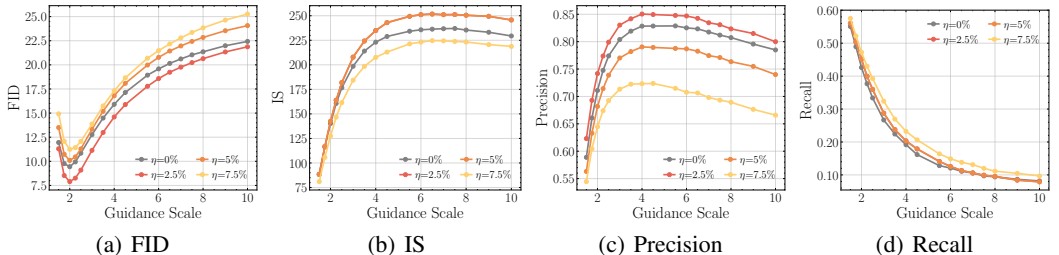

| (a) FID | (b) IS | (c) Precision | (d) Recall |

Figure 17: Qualitative evaluation results of 50K images generated by class-conditional LDMs pre-trained on IN-1K with asymmetric corruptions. The images are generated with various guidance scales using 1K class conditions from IN-1K and compared with validation images of IN-1K.

Table 6: Comparison of CEP with dropout and label smoothing on LDM IN-1K.

| Corruption | FID | IS |
|---|---|---|
| Clean | 9.44 | 138.46 |
| + Dropout 0.1 | 8.67 | 145.80 |
| + Label Smoothing 0.1 | 8.49 | 146.27 |
| + CEP-U | 7.00 | 170.73 |
| + CEP-G | 6.91 | 180.77 |

to add perturbation, and then fix them during training. For random data corruption, we randomly choose samples during training to make their label noisy by flipping to other classes. From the results in Table 7, we show that CEP works the best among all corruption methods. Also fixed CEP is more effective than adding data corruption (fixed and random). Random data corruption can be viewed as a CEP-variant with embeddings from flipping label instead of adding noise, and thus is also more effective than fixed data corruption.

Table 7: Comparison of fixed and random corruption on LDM IN-1K.

| Corruption | FID | IS |
|---|---|---|
| Clean | 9.44 | 138.46 |
| + CEP-U | 7.00 | 170.33 |
| + Fixed CEP-U | 7.94 | 154.48 |
| + Random Data Corruption | 8.13 | 143.07 |
| + Fixed Data Corruption | 8.44 | 140.27 |

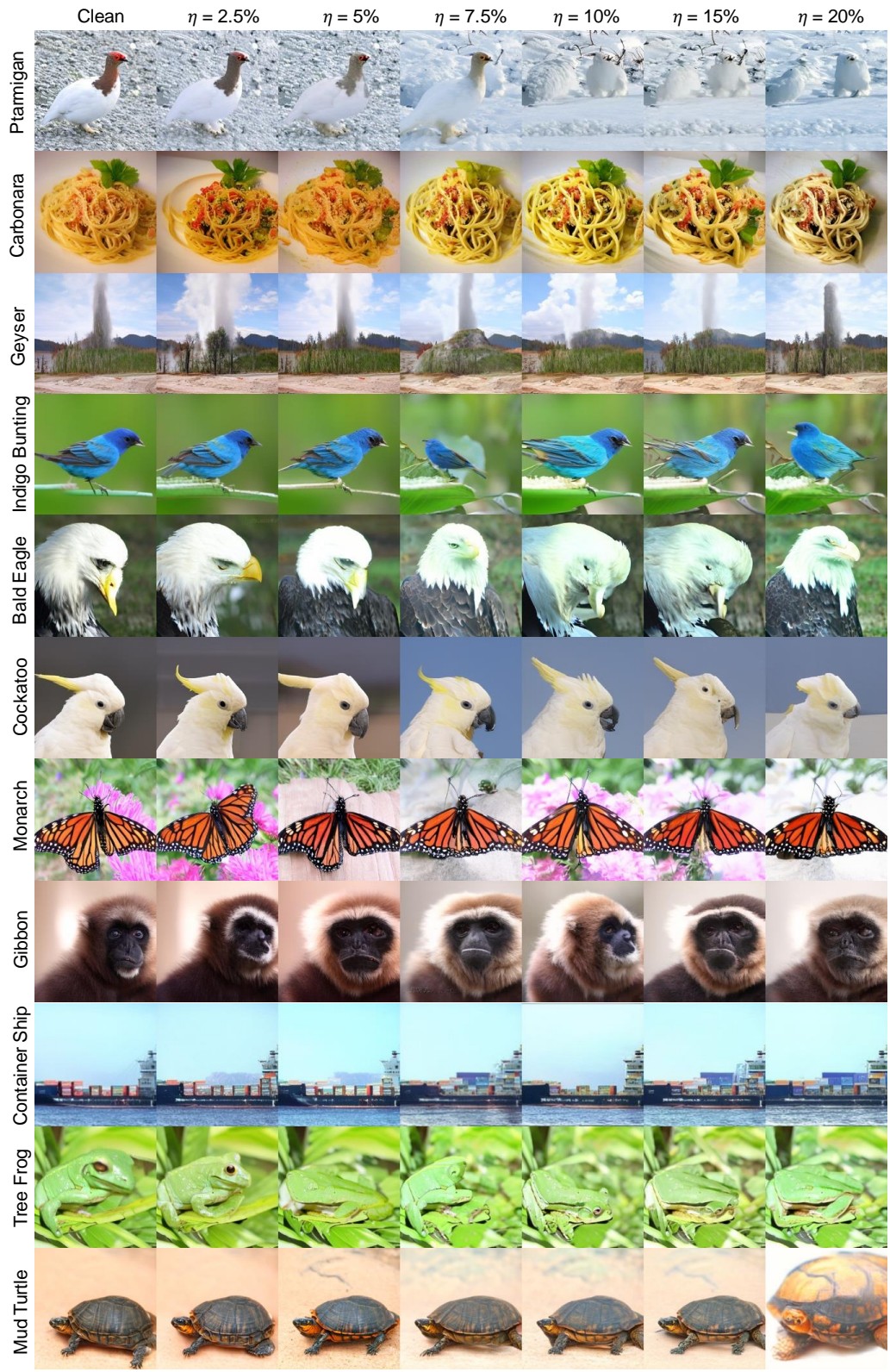

Figure 18: Visualization of LDMs IN-1K pre-training results.

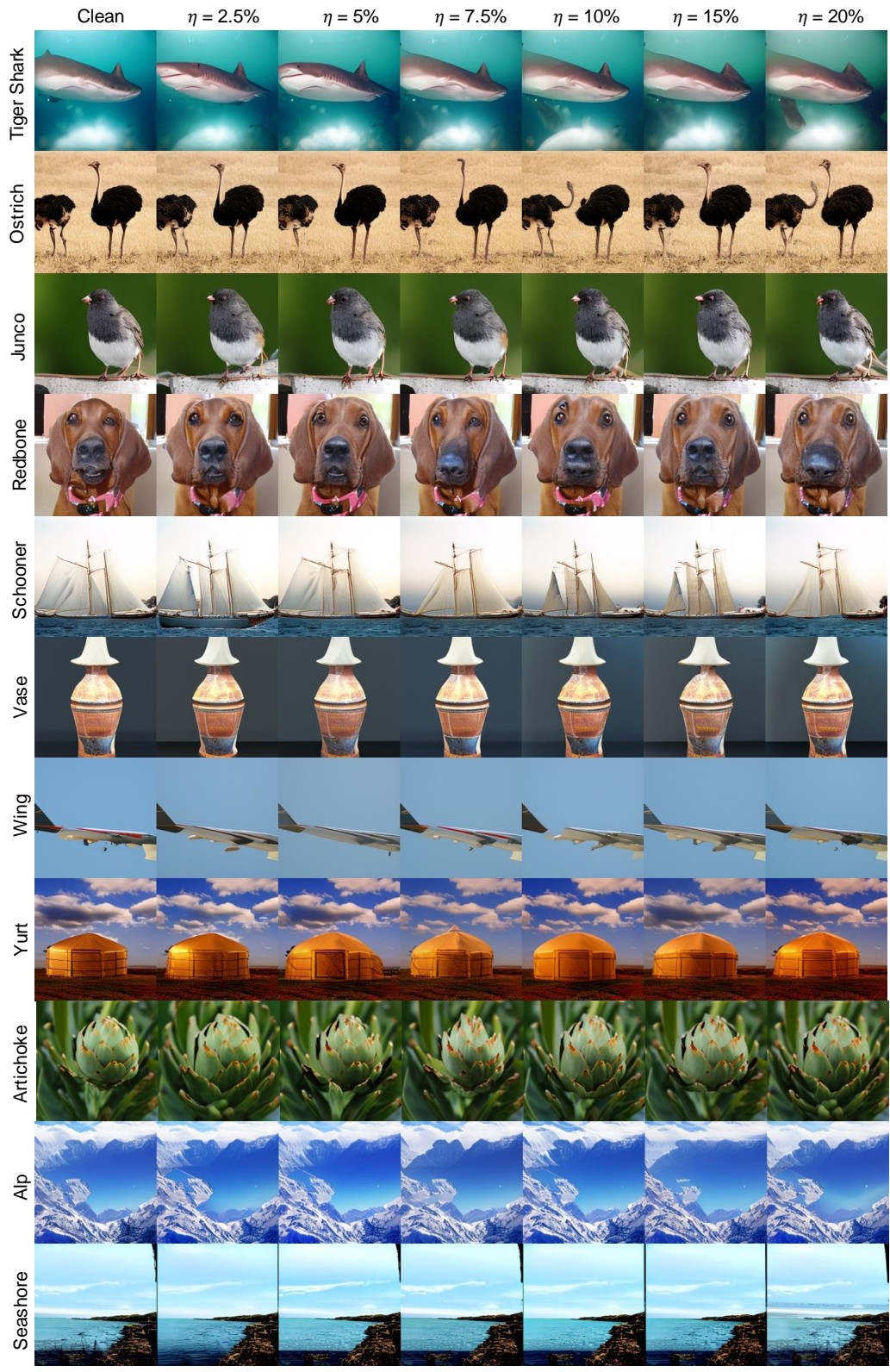

Figure 19: Visualization of DiT-XL/2 IN-1K pre-training results.

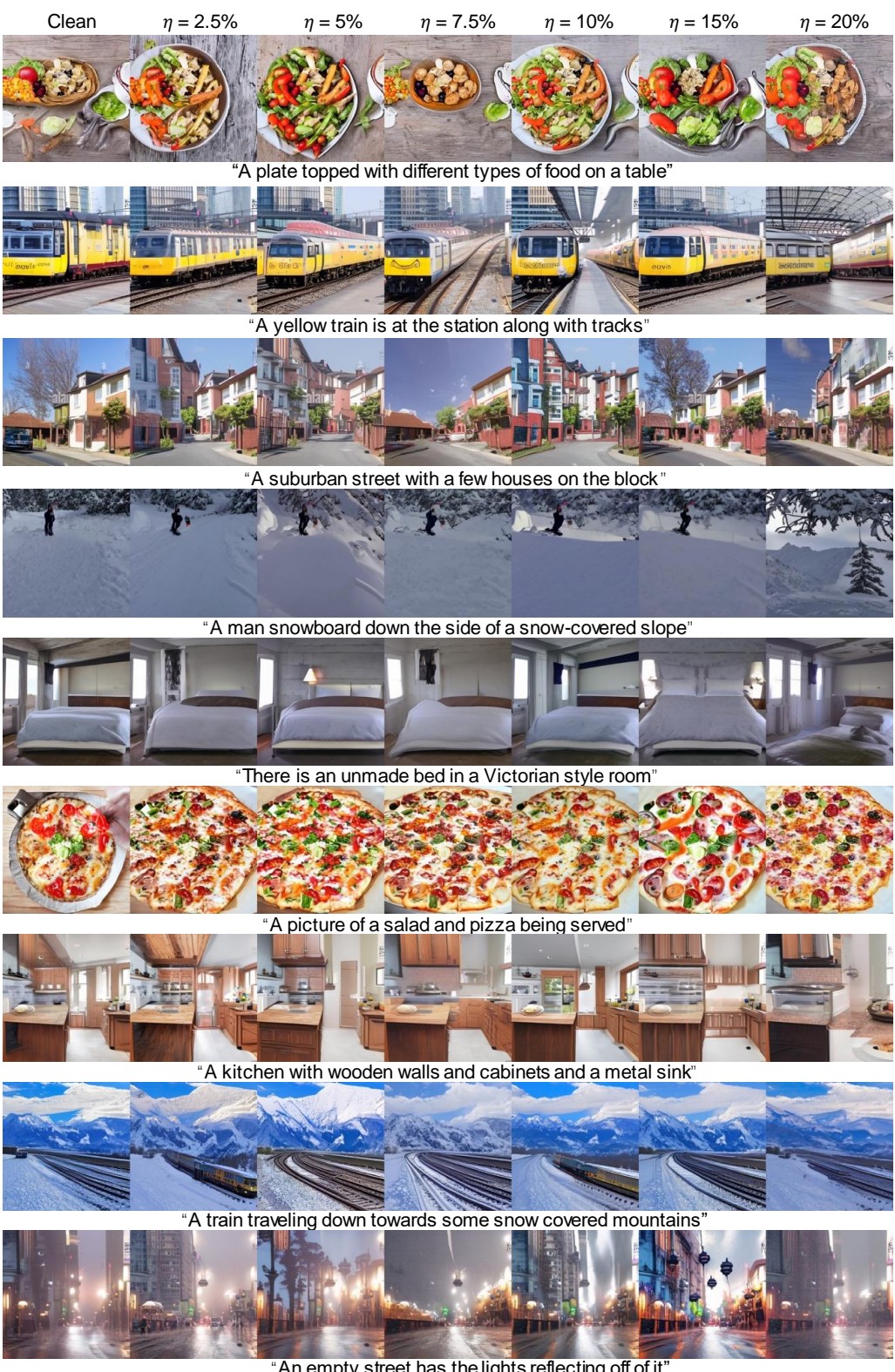

Figure 20: Visualization of LDMs CC3M pre-training results.

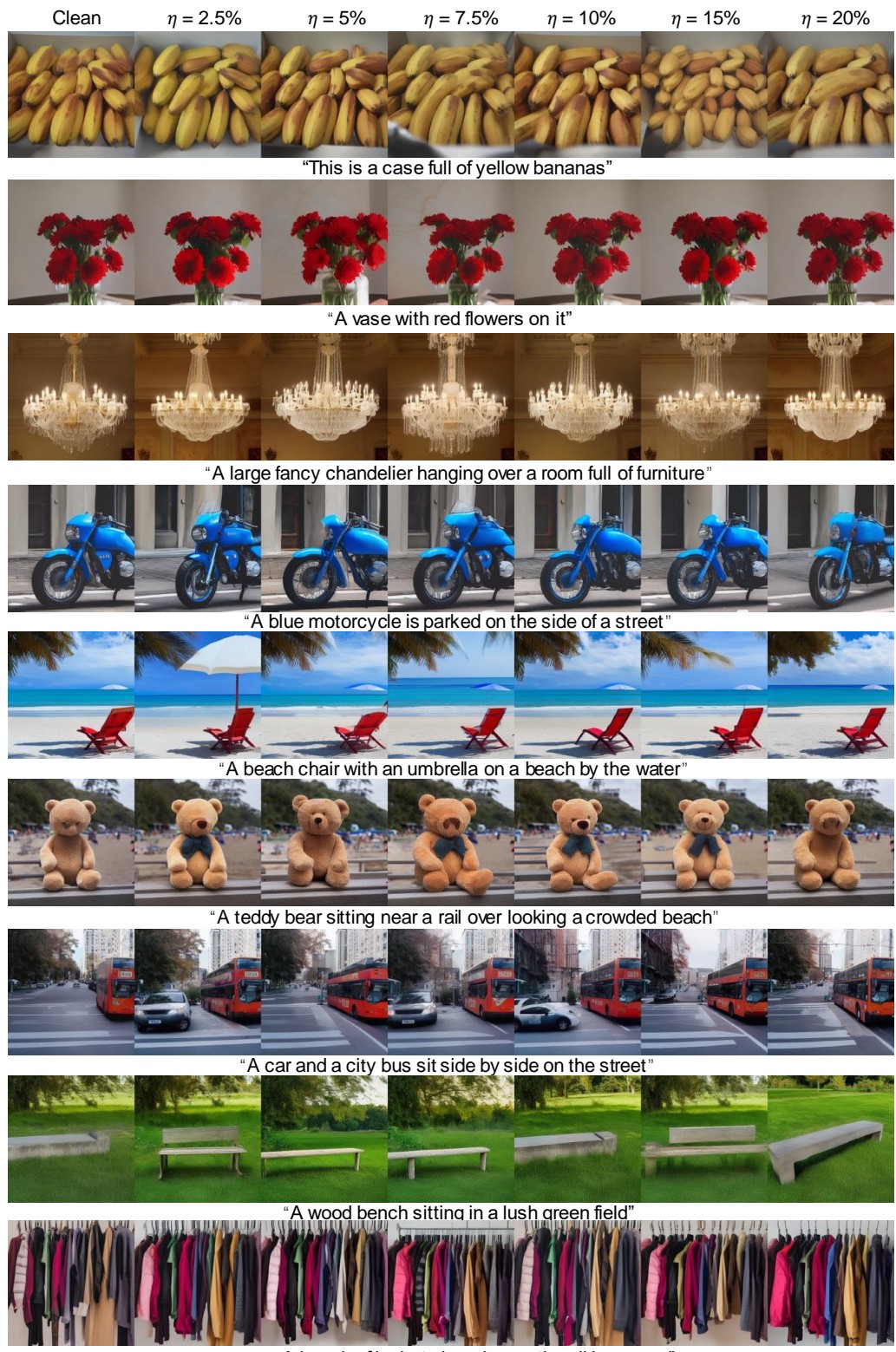

Figure 21: Visualization of LCM CC3M pre-training results.

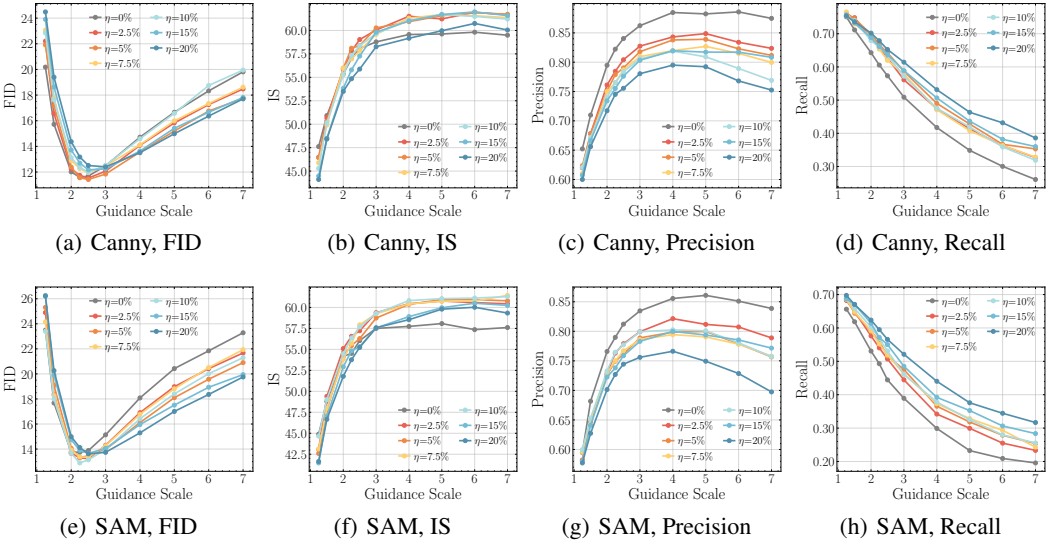

Figure 22: Qualitative evaluation results of 5K images generated by class-conditional LDMs pretrained on ImageNet-1K and personalized on ImageNet-100 using ControlNet. We personalized the models with different control styles, including canny ((a) - (d)), segmentation mask from SAM ((e) - (h)), and lineart ((i) - (l)). The images are generated using 100 class conditions with various guidance scales, compared with 5K validation images of ImageNet-100.

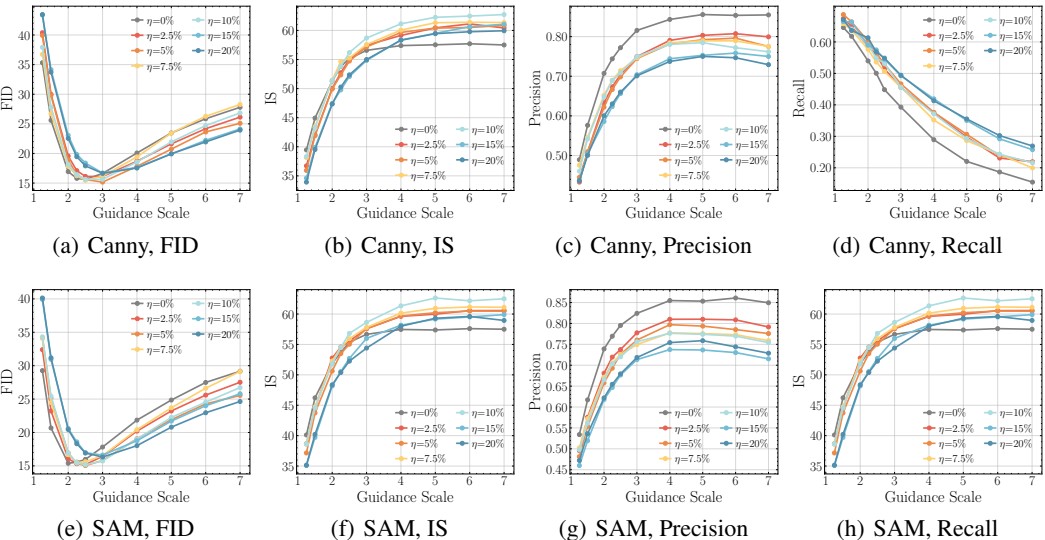

Figure 23: Qualitative evaluation results of 5K images generated by class-conditional LDMs pretrained on ImageNet-1K and personalized on ImageNet-100 using T2I-Adapter. We personalized the models with different control styles, including canny ((a) - (d)), segmentation mask from SAM ((e) - (h)), and lineart ((i) - (l)). The images are generated using 100 class conditions with various guidance scales, compared with 5K validation images of ImageNet-100.

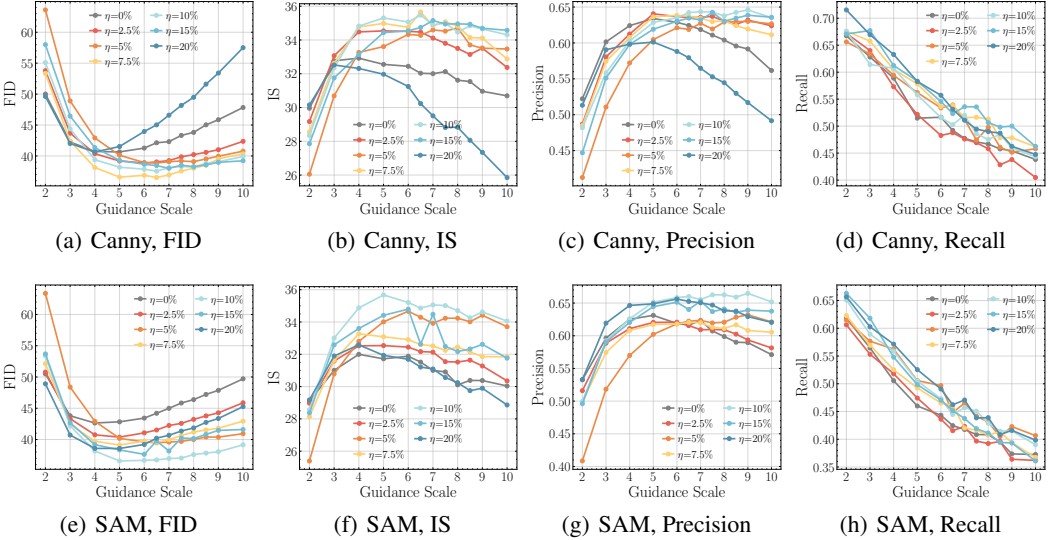

Figure 24: Qualitative evaluation results of 5K images generated by text-conditional LDMs pre-trained on CC3M and personalized on ImageNet-100 using ControlNet. We personalized the models with different control styles, including canny ((a) - (d))and segmentation mask from SAM ((e) - (h)). The images are generated using text captions annotated from BLIP with various guidance scales, compared with 5K validation images of ImageNet-100.

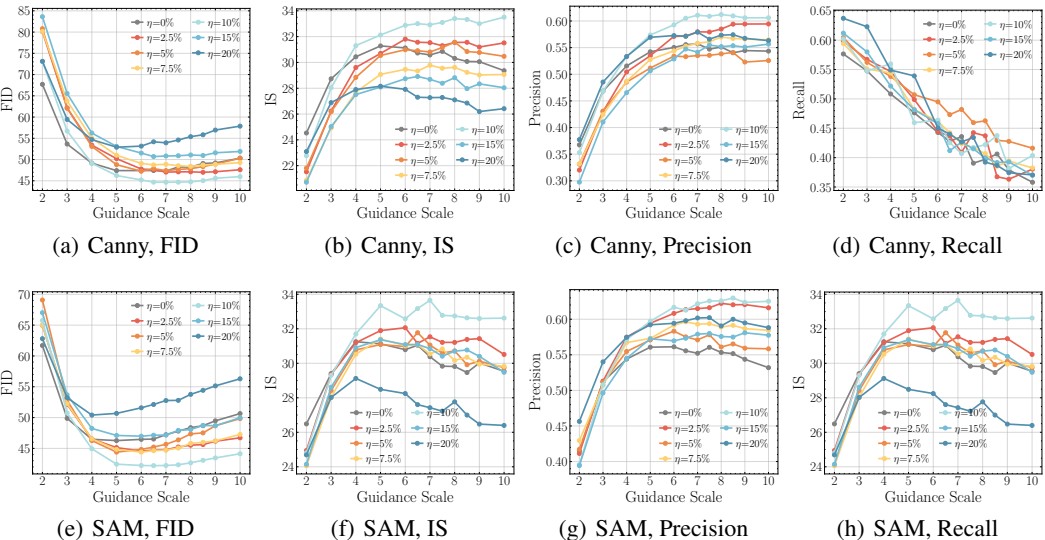

Figure 25: Qualitative evaluation results of 5K images generated by text-conditional LDMs pre-trained on CC3M and personalized on ImageNet-100 using T2I-Adapter. We personalized the models with different control styles, including canny ((a) - (d)) and segmentation mask from SAM ((e) - (h)). The images are generated using text captions annotated from BLIP with various guidance scales, compared with 5K validation images of ImageNet-100.

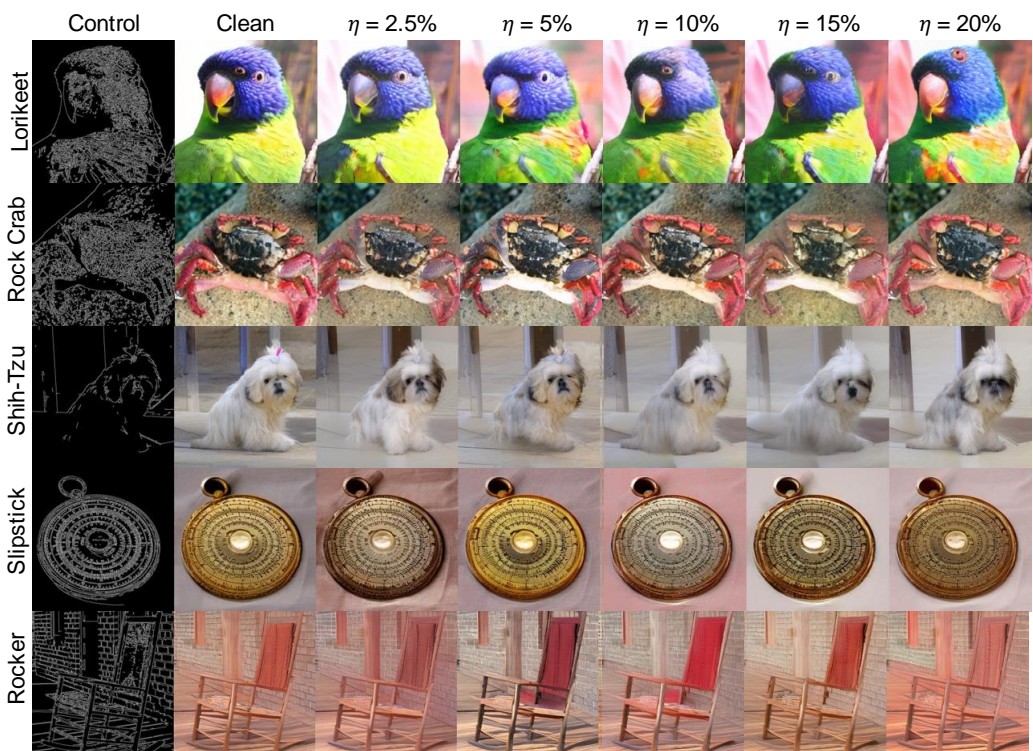

Figure 26: Visualization of LDMs IN-1K ControlNet Canny personalization results.

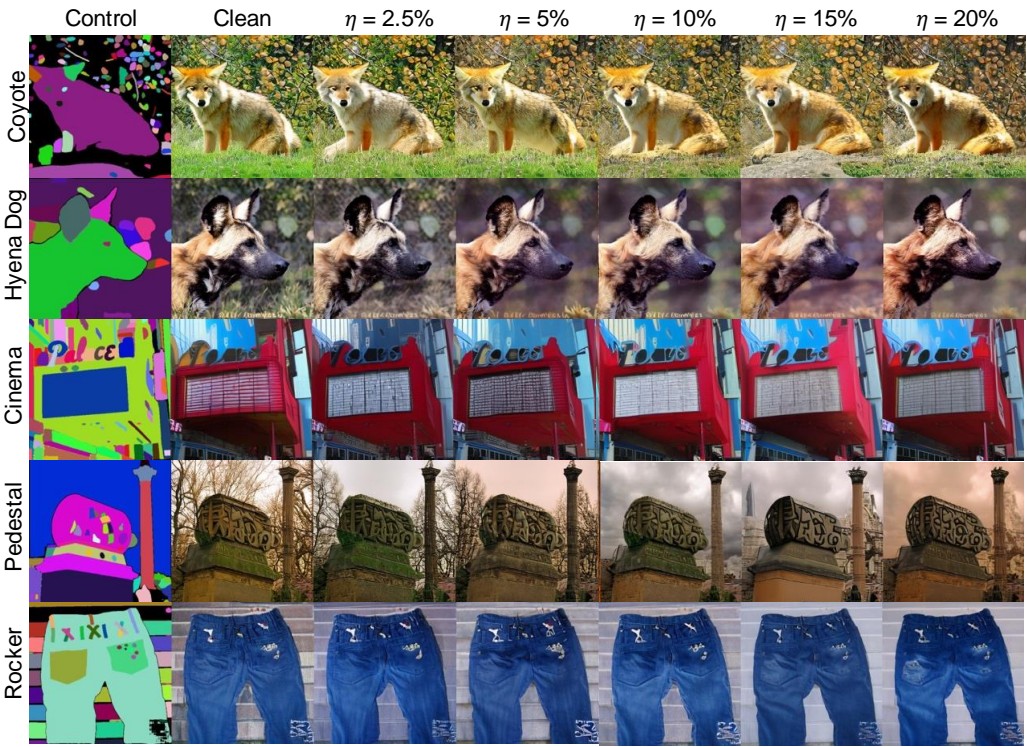

Figure 27: Visualization of LDMs IN-1K ControlNet SAM personalization results

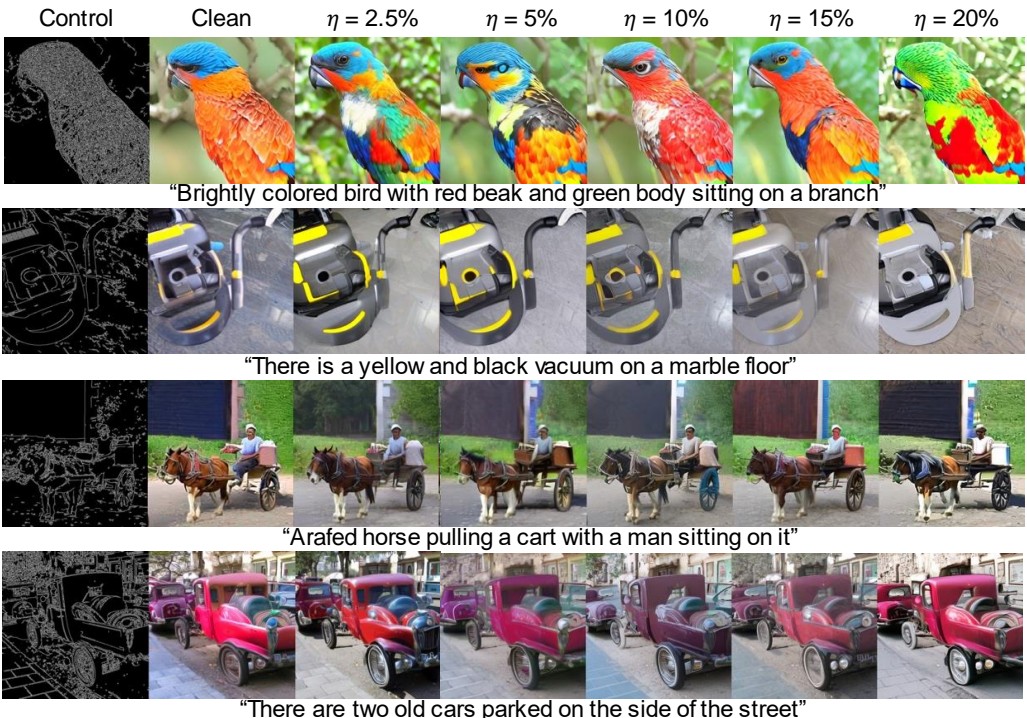

Figure 28: Visualization of LDMs CC3M ControlNet Canny personalization results.

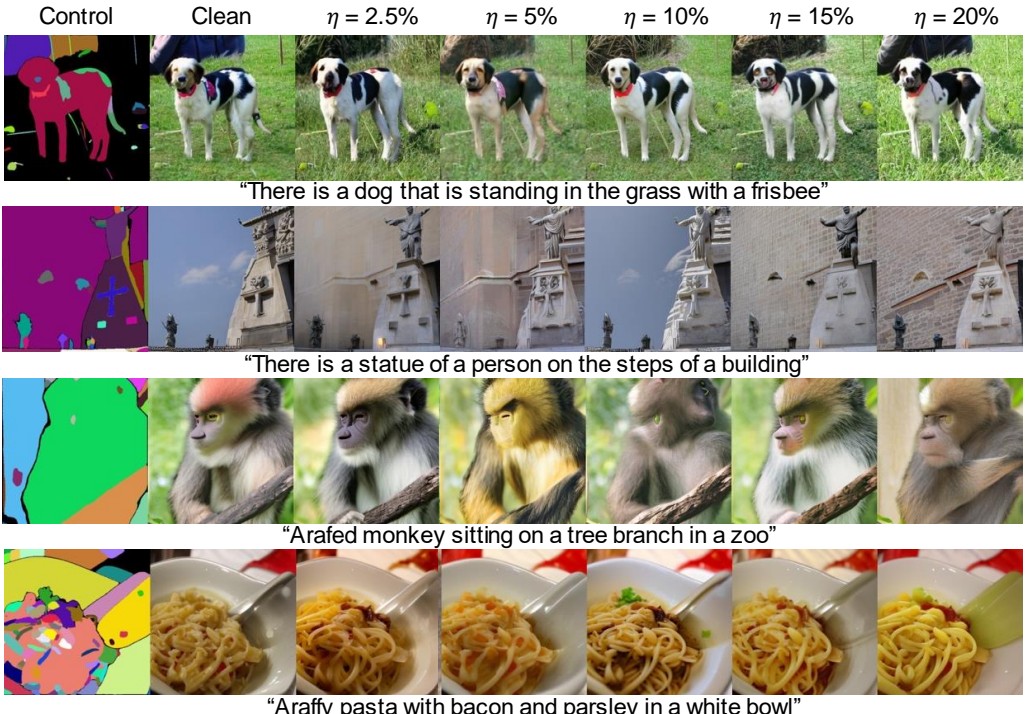

Figure 29: Visualization of LDMs CC3M ControlNet SAM personalization results

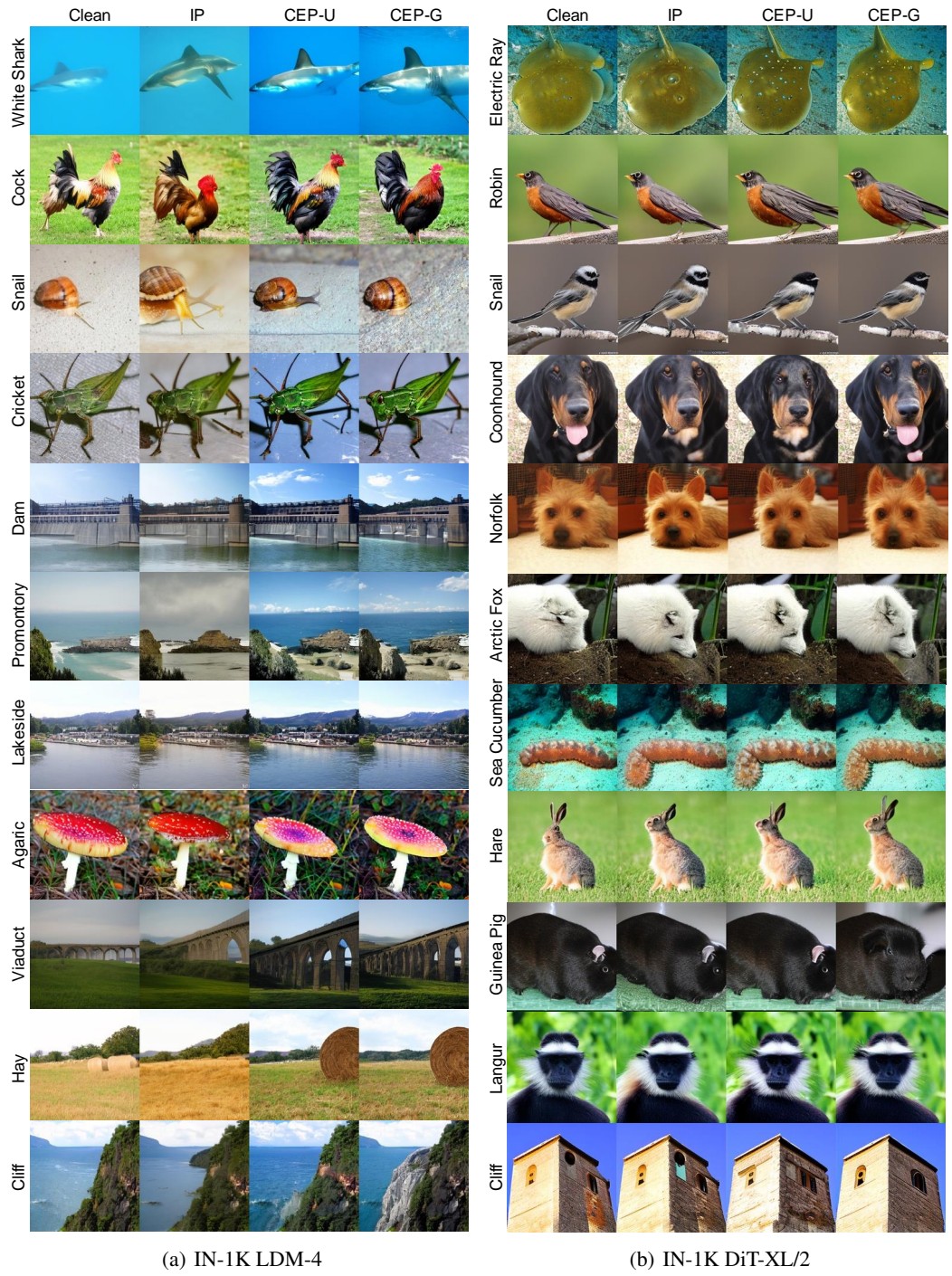

(a) IN-1K LDM-4        (b) IN-1K DiT-XL/2

Figure 30: Visualization of CEP on IN-1K pre-trained LDM-4 and DiT-XL/2

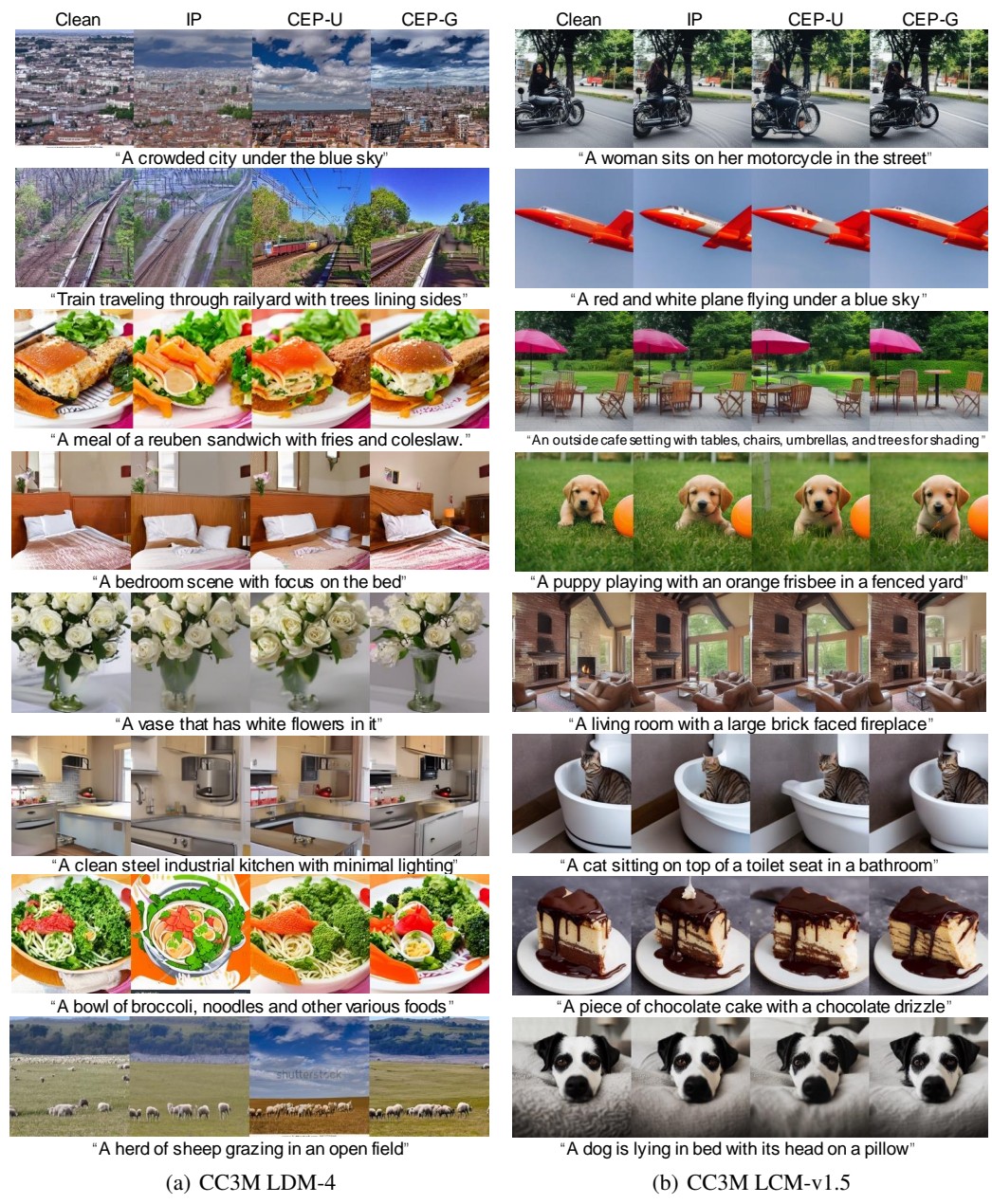

(a) CC3M LDM-4           (b) CC3M LCM-v1.5

Figure 31: Visualization of CEP on CC3M pre-trained LDM-4 and LCM-v1.5

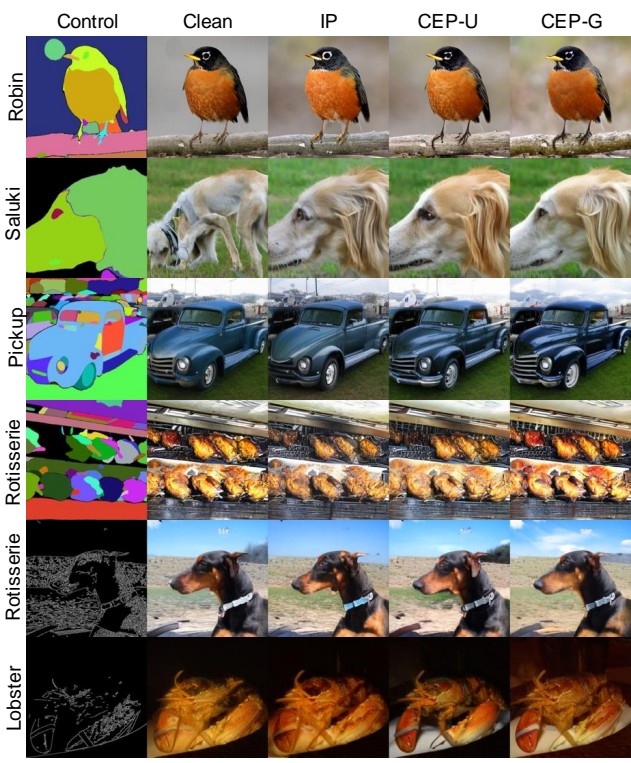

Figure 32: Visualization of CEP on ControlNet adapted IN-1K pre-trained LDM-4

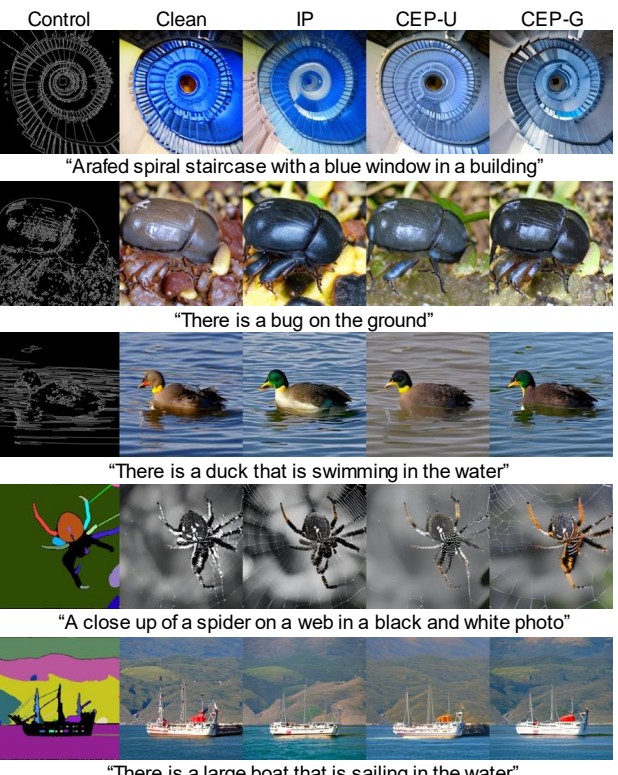

Figure 33: Visualization of CEP on ControlNet adapted CC3M pre-trained LDM-4

