# OpenReview forum: "Slight Corruption in Pre-training Data Makes Better Diffusion Models"
_NeurIPS.cc/2024/Conference — NeurIPS 2024 spotlight_

### Official Review · Reviewer_XQzf · 2024-07-08

**Soundness:** 4
**Presentation:** 3
**Contribution:** 3
**Rating:** 7
**Confidence:** 4

**Summary:**

The authors find that slight corruption in the conditioning of text-to-image diffusion models improves the performance. The authors theoretically analyze this empirical finding in a toy model where the goal is to learn to sample from a Gaussian Mixture and the network is piece-wise linear function. Inspired by their experimental and theoretical findings, the authors propose a technique to improve the performance of text-conditional diffusion models by adding noise to the embedding of the conditioning vector.

**Strengths:**

* The research topic is timely. State-of-the-art diffusion models are trained on billions of image-text pairs. Many of those are potentially noisy.
* The authors propose a very simple modification to the training of text-to-image diffusion models that leads to improved performance.
* The finding that slight corruption in the conditioning helps is interesting.
* The authors validate their findings through an extensive experimental evaluation.

**Weaknesses:**

* I believe that the authors should emphasize in the title of their paper and in the main body that they study corruption in the conditioning. Numerous works study corruption in the images themselves, e.g. see:

1) Ambient Diffusion: Learning Clean Distributions from Corrupted Data
2) Consistent Diffusion Meets Tweedie: Training Exact Ambient Diffusion Models with Noisy Data
3) GSURE-Based Diffusion Model Training with Corrupted Data
4) Solving inverse problems with score-based generative priors learned from noisy data

In these works, it has been observed that data corruption decreases the performance of diffusion models. I believe the authors should acknowledge these relevant works and clarify that their paper analyzes corruption in the conditioning.

* The theoretical model is not very relevant, at least not very relevant to Section 3 of the paper. In the experiments of Section 3, the authors mislabeled some of the training examples.  In the theoretical model, it is assumed that the true class is given to the model since the model is using the parameters that correspond to the true class. In a sense, the theoretical model of Section 3 is more related to the proposed algorithm in Section 5. Also, the existence of multiple centers in Section 4 does not seem relevant, unless I am missing something. This is because the model has separate parameters for each center and the centers are given for each example the model sees. I went over the proof and it looks like this is exploited in the proof as well, to obtain a closed-form solution for the optimal network.
* For the reasons mentioned above, it seems to me that the reason this works is some sort of regularization. I wonder if a similar effect could be obtained by increased dropout in the conditioning or some other form of regularization. Overall, I believe that the reader of the paper doesn't develop much intuition about the origins of the experimental finding.

**Questions:**

Apart from the weaknesses mentioned above, I have one more important question regarding this paper. Could the authors please clarify whether the corruption in Sections 3 and Sections 5 happens once for each data sample before the training or whether a new corruption takes place whenever we encounter the same data point across different epochs? To clarify, I want to understand which one of the two is happening:

1) We take a dataset, we corrupt it once, we train with the corrupted dataset
2) We use the clean dataset and every time we see a sample we corrupt it (by adding a different noise each time to its embedding or by changing each label/text to something else each time we see the same example across epochs).

If 2) is happening, it seems to me that the proposed method is closely related to regularization. I think this needs to be clarified and the effect of doing 1) or 2) should be studied as an ablation.

**Limitations:**

The authors have adequately discussed the limitations of their work.

---

> ### Author Rebuttal · Authors · 2024-08-06
>
> We thank the reviewer's efforts on this paper and his constructional suggestions on the ablation study of CEP.
> We now address the concerns as the follows.
>
> ---
>
> > I believe that the authors should emphasize...
>
> Thanks for this suggestion.
> We will make condition perturbation more clear in our title and main paper.
> We will also include these relevant works into discussion in our revised paper.
>
> > The theoretical model is not very relevant...more related to the proposed algorithm in Section 5.
>
> In the theoretical part, we considered a more general setting [1,2], where the label embedding is perturbed by Gaussian noise. We proved that slight corruption in pre-training data can lead to more diverse and higher quality generation in diffusion models. Additionally, the results from the theoretical part also inspire and justify the algorithm proposed in Section 5.
>
> The theoretical framework in Section 4 can also be applied to other types of noise. For example, in Section 3, label flipping can be regarded as a special corrupted label embeddings, i.e., $\mathbf{c}(y^c) = \mathbf{c}(y)+ \boldsymbol{\xi}$, where the noise $\boldsymbol{\xi}$ are vectors with entries 0, 1, and -1. Assuming there are $K$ classes, and each class has an equal sample size (approximately 1300 of each class in the experiment), each label has a probability $p$ of being flipped.Then the noise $\boldsymbol{\xi}$ follows a distribution satisfies $\mathcal{F}(p\frac{K-2}{K}\mathbf{1},(p-p^2(\frac{K-2}{K})^2)\mathbf{I})$. After obtaining the mean and variance information of the noise  $\boldsymbol{\xi}$ , the corresponding optimal linear denoising network and the generative distribution can still be solved using the techniques in Section 4.
>
> [1] Hu W, et al. Simple and Effective Regularization Methods for Training on Noisily Labeled Data with Generalization Guarantee. ICLR.
>
> [2] Tang Y H, et al. Detecting label noise via leave-one-out cross-validation. arXiv preprint 2021.
>
> > Also, the existence of multiple centers in Section 4 does not seem relevant, unless I am missing something.
>
> The multiple centers in Section 4 are derived from the assumption that the data follows a mixture of Gaussian distributions, which is a common assumption in the theoretical analysis of diffusion models [1-4]. The multiple centers correspond to the multiple classes of the real data. We consider modeling each center as a simplification of the real problem, and this simplification is sufficient to illustrate how noisy labels affect the training process of the denoising function (Section A.2), thereby influencing the final generative distribution (Section A.3), which shows improvements in both quality and diversity compared to the distribution generated with clean labels (Section A.4).
>
> Although the theoretical part considers a simplified single-center version, no other theoretical work simultaneously consider both the training and generative processes of diffusion models, and their interaction, while precisely solving the generative distribution. Our theoretical contributions should not be overlooked. We also agree that employing a more practical theoretical model could improve the theoretical part, which is left for our future study direction.
>
> [1] Shah, Kulin, et al.. "Learning mixtures of gaussians using the DDPM objective." NeurIPS 2023.
>
> [2] Chen, Sitan, et al. "Learning general gaussian mixtures with efficient score matching." *arXiv preprint.
>
> [3] Li, Puheng, et al. "On the generalization properties of diffusion models." NeurIPS 2024.
>
> [4] Li, Yangming, et al. "Soft Mixture Denoising: Beyond the Expressive Bottleneck of Diffusion Models." ICLR.
>
> > For the reasons mentioned above...finding.
>
> Thanks for pointing out this point. We also think CEP can be viewed as a regularization method for diffusion training.  By adding small noise during the training process, CEP prevents the trained model from overly collapsing onto the training data, thereby improving the diversity and quality of the generation distribution.
>
> Through ablation study of other methods, we showed that CEP is superior to other methods (dropout on conditional embedding and label smoothing of class labels for computing conditional embedding) and may function beyond regularization.
>
> | $\eta$ (%) 	| FID 	| IS 	|
> |:---:|:---:|---:|
> | LDM-4 IN-1K 	| 9.44 	| 138.46 	|
> | + Dropout 0.1 	| 8.67 	| 145.80 	|
> | + Label Smoothing 0.1	| 8.49 	|  146.27	|
> | + CEP-U 	|  7.00	| 170.73 	|
> | + CEP-G 	|  6.91	| 180.77 	|
>
> While the regularization methods like Dropout and Label Smoothing can also improve the performance of diffusion, CEP is significantly more effective than these methods.
> We will include this ablation study into the Appendix of our paper.
>
> > Could the authors please clarify whether the corruption in Sections 3 and Sections 5 happens...
>
> In section 3, for our empirical study, we adopt fixed corruption, i.e., take a dataset, introduce corruption, and train with the corrupted dataset.
> In section 5, the proposed CEP is introduced randomly at each training iteration.
> However, for results shown in Figure 8 with noisy dataset, we still used the fixed corrupted dataset.
>
> | Corruption 	| FID 	| IS 	|
> |:---:|:---:|:---:|
> | None 	| 9.44 	| 138.46 	|
> | CEP-U	| 7.00 	| 170.33 	|
> | Fixed CEP-U 	|  7.94	| 154.48 	|
> | Random Data Corruption (2.5%) 	|  8.13	| 143.07 	|
> | Fixed Data Corruption (2.5%)	| 8.44 	|140.27  	|
>
> Here we compare these two types of corruption.
> We showed that CEP works the best among all corruption methods.
> Also fixed CEP is more effective than adding data corruption (fixed and random).
> Random data corruption can be viewed as a CEP-variant with embeddings from flipping label instead of adding noise, and thus is also more effective than fixed data corruption.
> We will include this analysis in our Appendix.
>
> ---
> If you find our revise and response helpful, please consider raising the score for better support of our work.
> We are open to discuss more if any question still hold.

---

> > ### Comment · Reviewer_XQzf · 2024-08-09
> > **Increased my rating**
> >
> > Thanks for your rebuttal. Most of my concerns are now addressed. Please make the appropriate changes in the camera-ready version of your work, as promised.
> >
> > I increased my rating to 7.

---

> > > ### Author Response · Authors · 2024-08-09
> > >
> > > We thank the reviewer's feedback and efforts reviewing this work again.
> > > The changes and additional ablation studies will be included our revised work.

---

### Official Review · Reviewer_45qi · 2024-07-12

**Soundness:** 3
**Presentation:** 4
**Contribution:** 4
**Rating:** 7
**Confidence:** 3

**Summary:**

The paper investigates the impact of slight corruption of conditioning
information in pre-training data on the performance of diffusion models (DMs).
By introducing synthetic corruption to ImageNet-1K and CC3M datasets, the study
evaluates over 50 conditional DMs. Empirical and theoretical analyses reveal
that slight corruption enhances the quality, diversity, and fidelity of
generated images. Based on these insights, the work proposes Conditional
Embedding Perturbations (CEP) as a method to improve DM training which shows
significant improvements in both pre-training and downstream tasks.

**Strengths:**

- **Clarity, very well written:** The paper is clearly structured and builds up
   the motivation for the proposed method step by step from experiments and
   theoretical analysis.
 - **Extensive experiments:** The study examines many conditions training
   multiple diffusion models with different noise conditions from scratch.
   This gives a detailed perspective on the performance benefits of using
   slight conditioning corruption during training and is a valuable
   addition to the field.  These experiments are evaluated both qualitatively and quantitatively.
 - **Unexpected insight:** The study well describes a generally unexpected
   phenomenon and shows how it can be leveraged to develop an new method.

**Weaknesses:**

- **Notion of significance:** The paper claims that slight pre-training
   corruption yield significantly better performance in terms of FID and IS
   metrics. Nevertheless, this notion of significance is never formally tested.
   For this at least a experiment with multiple repetitions should be performed
   to determine statistical significance.
 - **Typos:**, Page 8, lines 257 and 259 rrecision -> precision, Page 9, lines 299 dataset -> data
 - **Link between theoretical analysis and metrics** should be made explicit.
   I.e. pointing out that the FID is essentially a Gaussian approximation
   based estimated for the 2-Wasserstein distance of inception features (preferably at page 7, line 223).

**Questions:**

None

**Limitations:**

The authors correctly reflect the limitations of the study. In particular, the
theoretical analysis is limited due to the assumptions made, but can still shed
some light on the underlying mechanism.  Further, the general issue of
evaluating image generative models quantitatively is pointed out and
a limitation of the work.

---

> ### Author Rebuttal · Authors · 2024-08-06
>
> We thank the reviewer's acknowledgment on this paper.
> We now address the weakness raised as follows.
>
> ---
>
> > Notion of significance: The paper claims that slight pre-training corruption yield significantly better performance in terms of FID and IS metrics. Nevertheless, this notion of significance is never formally tested. For this at least a experiment with multiple repetitions should be performed to determine statistical significance.
>
> Thanks for this great suggestion.
> Please understand that we will not be able to perform pre-training experiments for multiple repetitions, simply due to the un-affordable cost for that.
> This is also common in most study of generative models where only a single run is conducted in pre-training.
> Although we are not able to run multiple rounds of pre-training, for IN-1K models, we conducted an experiment for generating images with guidance scale of 2.25 of 5 random seeds, and reported average FID and IS with their standard deviation.
> Also, we conducted a t-test between the results from the clean model and noisy models and reported the p-value here.
> The results demonstrate that the improvement is indeed significant (extremely significant in terms of the p-value).
>
> | $\eta$ (%) 	| FID 	| FID p-value 	| IS 	| IS p-value 	|
> |:---:|:---:|:---:|:---:|:---:|
> | 0 	| 9.79 (0.11) 	|  	|  138.42 (0.35)	|  	|
> | 2.5 	| 8.49 (0.06) 	| < 0.0001 	|  155.73 (0.34)	| < 0.0001 	|
> | 5 	| 8.60 (0.07) 	| < 0.0001 	|  146.77 (0.29)	| < 0.0001 	|
> | 10 	| 9.11 (0.19)	| = 0.0001 	|  144.13 (0.86)	|  < 0.0001	|
>
>
> Due to the limited discussion time, we will try to include more significant test of other models and downstream applications for the final version of our paper.
>
>
> > Typos:, Page 8, lines 257 and 259 rrecision -> precision, Page 9, lines 299 dataset -> data
>
> Thanks for pointing out the typos!
> We have fixed all of these in the paper.
>
> > Link between theoretical analysis and metrics should be made explicit. I.e. pointing out that the FID is essentially a Gaussian approximation based estimated for the 2-Wasserstein distance of inception features (preferably at page 7, line 223).
>
> Thank you for your suggestion. We will emphasize on page 7, line 223 of the manuscript that the FID metric used in the experiments is the 2-Wasserstein distance under the assumption that the inception features follow a multivariate Gaussian distribution, to highlight their connection.
>
> ---
>
> If you find the above response help resolve your concerns, please consider raising score for better support of our work.
> Thanks!

---

> > ### Comment · Reviewer_45qi · 2024-08-12
> >
> > Thank you for the comments and additional points. While they do clarify some of my concerns, I think my previous score is still appropriate.

---

> > > ### Author Response · Authors · 2024-08-13
> > >
> > > Thank you for your comments and additional points. We appreciate your thoughtful consideration and suggestions.

---

### Official Review · Reviewer_6StB · 2024-07-16

**Soundness:** 3
**Presentation:** 3
**Contribution:** 2
**Rating:** 6
**Confidence:** 4

**Summary:**

In this paper, the authors study the effect of slight corruptions to the training data during pretraining of conditioned diffusion models. They introduce the perturbations to the "condition" in the diffusion models and show that this leads to a better model (via FID and other metrics).  They also provide an intuition on why corruptions help by theoretically studying this case on a GMM.

**Strengths:**

- Well-written paper
- All the experiments are well thought out, especially the downstream application experiments.
- Multiple metrics are evaluated and shown that perturbation helps, which makes the claim more sound.

**Weaknesses:**

- While authors showed extensively that adding noise can help, they missed to show "when does it help"? For example, if I'm a practitioner who wants to train a large model from scratch, I usually will not have the budget to train multiple models to determine the amount of noise that needs to be added. That insight is missing in this paper.

**Questions:**

- Does the optimal noise perturbation level transfer between the models? If 10% is the best noise level for LDM on one dataset, would it be the best noise for the same-size LDM on another?
- How do metrics look like throughout training for models trained with different levels of noise?
- CC3M corruption details are not clear. I went to Appendix B1, you mentioned 5 levels of corruption, but it is still vague! What are these 5 levels?
- How is the number of training iterations determined for the models? I looked at the appendix and made some rough calculations, it seems like you trained the IN model for 91 epochs and CC3M for 67 epochs (Pls correct me if I'm wrong). On what basis are these numbers decided? This again ties back to the question, when is "adding noise" helpful? Are we in an overtraining regime? Is that the reason why adding corruption acts as "augmentation" and hence the metrics improve?

- (Minor) A few relevant citations are missing. For example, adding perturbations to the conditioning both in text space and latent space is explored in the context of memorization in last year's Neurips paper [1].

[1] - "Understanding and mitigating copying in diffusion models." Advances in Neural Information Processing Systems 36 (2023): 47783-47803.

---

> ### Author Rebuttal · Authors · 2024-08-06
>
> Thanks for the reviewer's constructional feedback on this paper.
> We now address the weaknesses and questions as follows.
>
> ---
>
> > While authors showed extensively that adding noise can help, they missed to show "when does it help"? For example, if I'm a practitioner who wants to train a large model from scratch...That insight is missing in this paper.
>
> Thanks for mentioning this important question.
> However, we would like to highlight that core insight from our empirical and theoretical study is *not* to find the "optimal" noise ratio in the pre-training dataset that can help.
> Instead, we aim to present insights that, to pre-train diffusion models (and self-supervised models), it is important to ensure the diversity in the pre-training and slight corruption in the pre-training dataset may benefit diversity.
> No over-filtering on corruption is needed for pre-training data.
>
> Insights for practitioners: the proposed CEP can be simply adapted into pre-training of diffusion models to improve performance, no matter the datasets and models. In Figure 8 (b), we also presented that, on noisy datasets (of different levels) as in practical settings, CEP can also facilitate both the pre-training and downstream transferring performance.
>
>
> > Does the optimal noise perturbation level transfer between the models? If 10% is the best noise level for LDM on one dataset, would it be the best noise for the same-size LDM on another?
>
> This is indeed an intriguing question. While our empirical study shows that slight corruption universally improves the performance across models and datasets, we would like to mention that it also depends on the pre-training dataset size.
> As the dataset size scales larger, the slight corruption ratio may also decrease, since we may not expect 10% of our pre-training data of size 2B is corruption.
>
>
> > How do metrics look like throughout training for models trained with different levels of noise?
>
> Thanks for mentioning this important question.
> We use the pre-trained LDM-4 IN-1K clean and noise $2.5$ to generate images at different checkpoints along training, using a guidance scale of $2.5$.
> We present the FID and IS results as follows:
>
> FID:
> | $\eta$ (%) 	| 10K 	| 25K 	| 50K 	| 75K 	| 100K 	| 125K 	| 150K |
> |:---:|:---:|:---:|:---:|:---:|:---:|:---:| :---:|
> | 0 	| 71.48 	|  52.02	| 20.88 	| 14.49 	|  12.66	| 10.44 | 10.12 |
> | 2.5 	| 77.94 	| 51.59  	| 21.16 	| 13.08 	| 12.24 	| 9.25  	| 8.98 |
>
> IS:
> | $\eta$ (%) 	| 10K 	| 25K 	| 50K 	| 75K 	| 100K 	| 125K 	| 150K |
> |:---:|:---:|:---:|:---:|:---:|:---:|:---:| :---:|
> | 0 	| 14.86 	| 23.49 	| 71.26 	| 93.85 	| 103.27 	| 164.41 	| 170.22 |
> | 2.5 	| 13.66 	| 24.40  	| 64.27 	|  97.11	| 109.39 	| 167.21 	| 175.83 |
>
> From the results, we can observe that, with slight corruption, the model performs slightly worse at the beginning of the training (before 50K iter.), while becomes outperforming the clean one towards the end of the training.
>
> We will include these results in our Appendix.
>
>
> > CC3M corruption details are not clear. I went to Appendix B1, you mentioned 5 levels of corruption, but it is still vague! What are these 5 levels?
>
> Sorry for the confusion caused. For CC3M, to introduce corruptions, we randomly sample two image-text pairs, i.e. $(I_1, T_1)$ and $(I_2, T_2)$, and swap the text of these two pairs to make it $(I_1, T_2)$ and $(I_2, T_1)$.
> For corruption levels, we use the same levels, i.e. {2.5, 5, 7.5, 10, 15, 20}%, as in ImageNet.
> These are shown in the legend of CC3M figures and mentioned at line 112.
> We will add more details in Appendix B1 and highlight in the main text to make this more clear.
>
>
> > How is the number of training iterations determined for the models?...On what basis are these numbers decided?
>
> The number of training iterations mainly follows settings in the previous papers.
> For example, in LDM paper [1], the main settings of LDM-4 on ImageNet and CC3M is 178K and 390K steps.
> To replicate this, we used a total batch size of 512 in our experiments, resulting in roughly 2.5K and 5.4k training steps per epoch. Thus we use 71 and 70 epochs of IN-1K (1.28M samples) and CC3M (2.8M samples due to expired url) to make the total training iteration as 178K and 390K.
> This is similar for DiT and LCM, where we follow their paper to setup the training steps.
>
> > "this again ties back to the question, when is "adding noise" helpful? Are we in an overtraining regime? Is that...and hence the metrics improve?"
>
> From the above results, one can see that, adding noise starts being helpful once the model converges to generate reasonable figures and consistently improves along training after that.
> From our paper and ablation results for Reviewer XQzf, you can also see that CEP performs better than other regularization methods, including input perturbation (augmentation), dropout on embedding, label smoothing, etc.
> This demonstrates the better effectiveness of CEP than traditional regularization.
>
> > A few relevant citations are missing...last year's Neurips paper [1].
>
> Thanks for mentioning this relevant work. We will add this work into discussion in our revised paper. Theie differences are:
>
> This work mainly focuses on memorization and copying of diffusion models.
> The authors found adding perturbations to condition or condition embedding help relieve the copying issue (but FID remains similar as clean, possibly due to the small-scale dataset and focus on fine-tuning instead of pre-training), which aligns also with our findings as shown in Figure 11 (i) and Figure 13 (i), where large L2 distance of generated images and ground truth are present with pre-training corruption.
>
> ---
>
> We hope the above response can resolve your concern.
> If you find our revise and response helpful, please consider raising the score to better support of our work.
> Also please let us know if there is any more question.

---

> > ### Comment · Reviewer_6StB · 2024-08-13
> > **Thank you for the response**
> >
> > While I still disagree with the justification for the paper's insights, I appreciate the authors' effort in addressing most of my other questions. I will increase my score.

---

> > > ### Author Response · Authors · 2024-08-13
> > >
> > > Thank you for your feedback and for acknowledging our efforts to address your questions.
> > > We appreciate for the increased score.

---

### Official Review · Reviewer_mxHS · 2024-07-21

**Soundness:** 3
**Presentation:** 3
**Contribution:** 3
**Rating:** 6
**Confidence:** 4

**Summary:**

This paper claims that slight corruption of the condition is beneficial for conditional diffusion models. Experimentally, they conducted experiments with label flips or text swaps in the dataset and observed the performance improvement over zero noise at noise levels of 2.5% to 5%. Theoretically, they used a Gaussian mixture model to show that generation diversity and quality, as measured by entropy and Wasserstein distance, can be improved with a small amount of state corruption. Based on this, they proposed a method for injecting small amounts of Gaussian noise into the condition embeddings and experimentally demonstrated its superiority.

**Strengths:**

Their claim offers a new perspective that may not be intuitive, but has been partially mentioned in the existing literature. They provide a solid experimental and theoretical explanation for this view.

**Weaknesses:**

* From my understanding, in Theorem 2, $\gamma$ is $O(1/\sqrt{n})$, which is the scenario for small corruption. It would be beneficial to address this explicitly in the main text.

* It would be beneficial if the method that determines the noise level could be linked to the theory like using the $\gamma=O(1/\sqrt{n})$,.

* They briefly mentioned the various assumptions in their theory as limitations. It would be helpful to provide a detailed explanation of these assumptions and the scenarios in which they hold.

**Questions:**

Please see the Weaknesses part.

**Limitations:**

They mentioned in the last section.

---

> ### Author Rebuttal · Authors · 2024-08-06
>
> Thanks for your time reviewing this paper. We now address your questions as follows.
>
> ---
>
> > From my understanding, in Theorem 2...It would be beneficial to address this explicitly in the main text.
>
> Thank you for your suggestion. We have already mentioned that the order of $\gamma$ is  $O(\frac{1}{\sqrt{\max_k n_k}})$ in Theorem 2 and we  will further emphasize in the main text (page 7, lines 219-220) that the noise level needs to satisfy $\gamma = O(\frac{1}{\sqrt{\max_k n_k}})$.
>
>
> > It would be beneficial if the method that determines the noise level could be linked to the theory like using the [...]
>
>
> In the experimental section of Section 5.2, the (Gaussian) noise we used is approximately $0.04\epsilon$, where $\epsilon \sim N(0,I)$, while the theoretical part in Section 4 proves that the noise level is approximately $0.03 \epsilon$. The noise levels in the experiment and the theory are very close. We will further emphasize their link in the main text (page 8, line 247).
>
>
> > They briefly mentioned the various assumptions in their theory as limitations. It would be helpful to provide a detailed explanation of these assumptions and the scenarios in which they hold.
>
> Our theoretical part includes the main setup that the data distribution follows a mixture of Gaussian distributions, which is a common assumption in the theoretical analysis of diffusion models [1,2,3,4,5]. Follow the previous work [2,6], we accordingly considered a piece-wise linear function as the denoising neural network, which has been proven sufficient to handle the Gaussian mixtures. Based on these previous theoretical works, our theoretical framework, although seems simple, provides a paltform for understanding the improvements in quality and diversity introduced by noised label embeddings.
>
>    [1] Shah, Kulin, Sitan Chen, and Adam Klivans. "Learning mixtures of gaussians using the DDPM objective." *Advances in Neural Information Processing Systems* 36 (2023): 19636-19649.
>
>    [2] Chen, Sitan, Vasilis Kontonis, and Kulin Shah. "Learning general gaussian mixtures with efficient score matching." *arXiv preprint arXiv:2404.18893* (2024).
>
>    [3] Li, Puheng, et al. "On the generalization properties of diffusion models." *Advances in Neural Information Processing Systems* 36 (2024).
>
>    [4] Cui, H., Krzakala, F., Vanden-Eijnden, E., & Zdeborová, L. (2023). Analysis of learning a flow-based generative model from limited sample complexity. arXiv preprint arXiv:2310.03575.
>
>    [5] Li, Yangming, Boris van Breugel, and Mihaela van der Schaar. "Soft Mixture Denoising: Beyond the Expressive Bottleneck of Diffusion Models." *The Twelfth International Conference on Learning Representations*.
>
>    [6] Gatmiry K, Kelner J, Lee H. Learning mixtures of gaussians using diffusion models[J]. arXiv preprint arXiv:2404.18869, 2024.
>
> ---
>
> If you find our response helpful, please consider raising the score for better supporting our work.

---

> > ### Comment · Reviewer_mxHS · 2024-08-13
> >
> > Thank you for the author's response. After checking the author's response and other reviews, most of my concerns have been addressed, so I raise the score.

---

> > > ### Author Response · Authors · 2024-08-13
> > >
> > > Thank you for your feedback and for revisiting our paper. We're glad our response addressed your concerns, and we appreciate the updated score.

---

### Author Rebuttal · Authors · 2024-08-06

We first would like to thank all reviewer and AC's efforts and time reviewing this paper and suggestions for making it better.

According to the reviewers' feedback, we summarize several critical points and make a general response here.

---

> Motivation for corruption in pre-training.

- First of all, the motivation to study the problem of pre-training conditional corruption is well-supported by existing research showing that diffusion models are trained on massive (and potentially corrupted) pre-training data, which naturally motivates us to study *what* are the effects of the pre-training corruption, and *whether* and *why* the corruption in pre-training datasets can help.
- Second, the motivation of developing theories are also motivated by our experimental findings. We tie this advantage with the diversity of the learned distribution which is also closer to the ground truth distribution through a theoretical analysis on the mixture of Gaussian.
- Third, from the methodology level, finding an optimal noise ratio to corrupt the pre-training data is not our focus and it is also naturally impossible due to the cost of pre-training. One motivation to design our CEP approach is not to find the optimal noise ratio, but to mimic the pre-training corruption, to improve the diffusion pre-training. We show that CEP generally improves the performance and can also be applied in practical datasets with corruption in it already.

> Connection between theoretical and empirical findings. And Assumptions in our theoretical analysis.

While we consider a general form for noise in the theoretical analysis part, we demonstrated in the response to Reviewer XQzf that the fixed label noise can be included in this general form too.
We also showed in the response to Reviewer mxHS that the corruption level used in CEP is pretty close to our theoretical analysis.

In Section 3, we mainly consider a mixture of Gaussian model for our analysis.
This helps us derive clearer form to analyze the effect of noise.
The multi-centroids distribution of GMMs also well aligns with our insights in this paper, i.e., the slight corruption increases the diversity in the learned distribution.


> CEP functions as a regularization method.

Reviewer 6StB and Reviewer 45qi have raised questions regarding the regularization effects of the proposed CEP.

To show this, we have conducted additional ablation study of:

* FID and IS along training with slight corruption (response to Reviewer 6StB): we demonstrated that, with slight corruption, the noisy model becomes better than the clean one at early training.
* Comparison of CEP with Dropout and label smoothing on conditional embedding (response to Reviewer 45qi): we demonstrated that CEP is more effective than Dropout and label smoothing as a regularization method.
* Comparison of fixed data point CEP and random data corruption (response to Reviewer 45qi): we showed that fixed CEP is more effective than random data corruption and fixed data corruption. Also, random data corruption can be viewed as a CEP-variant, which is more effective than fixed data corruption.

Although CEP can also be viewed as a regularization method, through these experiments, we demonstrated its superiority and effectiveness over other methods.

Furthermore, we would like to highlight that while regularization in diffusion modeling is a broad research area, several works in the past have primarily focused on settings limited to only fine-tuning domain [1] and exclusively for inverse problems only [2,3]. We aim to emphasize the practicality of our approach in the context of large pre-training setups which differs from prior work.

[1] Tang, Wenpin. "Fine-tuning of diffusion models via stochastic control: entropy regularization and beyond." arXiv preprint arXiv:2403.06279 (2024).

[2] Chung, Hyungjin, Byeongsu Sim, Dohoon Ryu, and Jong Chul Ye. "Improving diffusion models for inverse problems using manifold constraints." Advances in Neural Information Processing Systems 35 (2022): 25683-25696.

[3] Mardani, Morteza, Jiaming Song, Jan Kautz, and Arash Vahdat. "A variational perspective on solving inverse problems with diffusion models." arXiv preprint arXiv:2305.04391 (2023).

---

### Decision · Program_Chairs · 2024-09-25

**Decision:**

Accept (spotlight)

**Comment:**

In this work, authors present an interesting finding that slightly corrupting the training data used to train diffusion models can actually lead to better performance - they demonstrated it on models trained using ImageNet and CC-3M and also proposed a new algorithm which can be integrated in any standard diffusion model training paradigm. Reviewers acknowledged the fact that the paper is well-written, experiments are robust and show the efficacy of the proposed algorithm. Compute budgets permitting,  I found the work to be rigorous and liked the emphasis on an under-explored aspect within the whole diffusion model training literature and can be beneficial for the field to further explore.

I recommend acceptance - as a suggestion, I'd like to see how this technique can be integrated into large-scale diffusion model training where it can be expensive to tune the noise scale.